# MedScope: Incentivizing "Think with Videos" for Clinical Reasoning via Coarse-to-Fine Tool Calling

**Wenjie Li** [* 1 2]  **Yujie Zhang** [* 2 3]  **Haoran Sun** [4]  **Xingqi He** [3]  **Hongcheng Gao** [5]  **Chenglong Ma** [2]  **Ming Hu** [4]
**Guankun Wang** [6]  **Shiyi Yao** [1]  **Renhao Yang** [1]  **Hongliang Ren** [6]  **Lei Wang** [1]  **Junjun He** [2 4]  **Yankai Jiang** [4]

## Abstract

Long-form clinical videos are central to visual evidence-based decision-making, with growing importance for applications such as surgical robotics and related settings. However, current multimodal large language models typically process videos with passive sampling or weakly grounded inspection, which limits their ability to iteratively locate, verify, and justify predictions with temporally targeted evidence. To close this gap, we propose **MedScope**, a tool-using clinical video reasoning model that performs coarse-to-fine evidence seeking over long-form procedures. By interleaving intermediate reasoning with targeted tool calls and verification on retrieved observations, MedScope produces more accurate and trustworthy predictions that are explicitly grounded in temporally localized visual evidence. To address the lack of high-fidelity supervision, we build **ClinVideo-Suite**, an evidence-centric, fine-grained clinical video suite. We then optimize **MedScope** with **G**rounding-**A**ware **G**roup **R**elative **P**olicy **O**ptimization (**GA-GRPO**), which directly reinforces tool use with grounding-aligned rewards and evidence-weighted advantages. On full and fine-grained video understanding benchmarks, **MedScope** achieves state-of-the-art performance in both in-domain and out-of-domain evaluations.

*Equal contribution  [1]Department of Orthopaedics, Ruijin Hospital, College of Health Science and Technology, Shanghai Jiao Tong University School of Medicine, Shanghai, China [2]Shanghai Innovation Institute, Shanghai, China [3]Shanghai Key Lab of Intelligent Information Processing, College of Computer Science and Artificial Intelligence, Fudan University, China [4]Shanghai Artificial Intelligence Laboratory, Shanghai, China [5]Tsinghua University, Beijing, China [6]The Chinese University of Hong Kong, Hong Kong, China. Correspondence to: Yankai Jiang <jiangyankai@pjlab.org.cn>, Junjun He <hejunjun@sjtu.edu.cn>, Lei Wang <ray_wangs@hotmail.com>.

*Proceedings of the 43rd International Conference on Machine Learning*, Seoul, South Korea. PMLR 306, 2026. Copyright 2026 by the author(s).

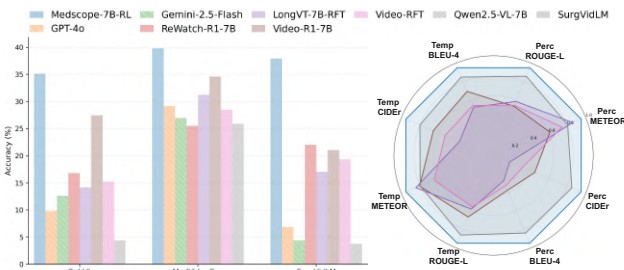

*Figure 1.* **Performance comparison on full and fine-grained video understanding and VQA benchmarks.** Left: grounded VQA accuracy on ClinVideo-Eval. Right: full and fine-grained video understanding quality on SVU-31K.

Our approach illuminates a path toward medical AI agents that can genuinely "think with videos" through tool-integrated reasoning. Code and resources are available at https://github.com/SII-WenjieLisjtu/MedScope.

## 1. Introduction

Understanding long videos that span tens of minutes remains challenging for multimodal intelligence, because decisive yet temporally sparse evidence is buried within complex event sequences across thousands of frames (Qian et al., 2024; Liu et al., 2025a; Ma et al., 2025; He et al., 2024a; Tang et al., 2025). In medicine, long surgical and endoscopic recordings support intraoperative decision making(Biffi et al., 2022), postoperative review(Al Abbas et al., 2024), and surgical training(Gastager et al., 2025), where conclusions often hinge on brief, fine-grained cues and must be grounded in concrete visual evidence(Kiyasseh et al., 2023). Clinicians therefore do not watch an entire procedure in one pass; they first skim the global course to form hypotheses, then revisit short temporal windows to retrieve and verify subtle signs(van der Leun et al., 2024).

Recent advances in multimodal chain-of-thought (CoT) reasoning have improved interpretability, and large multimodal models (LMMs) perform strongly on short-video understanding(Zhang et al., 2025c; Wang et al., 2025d). Yet most methods still follow an R1-style(Guo et al., 2025), text-first

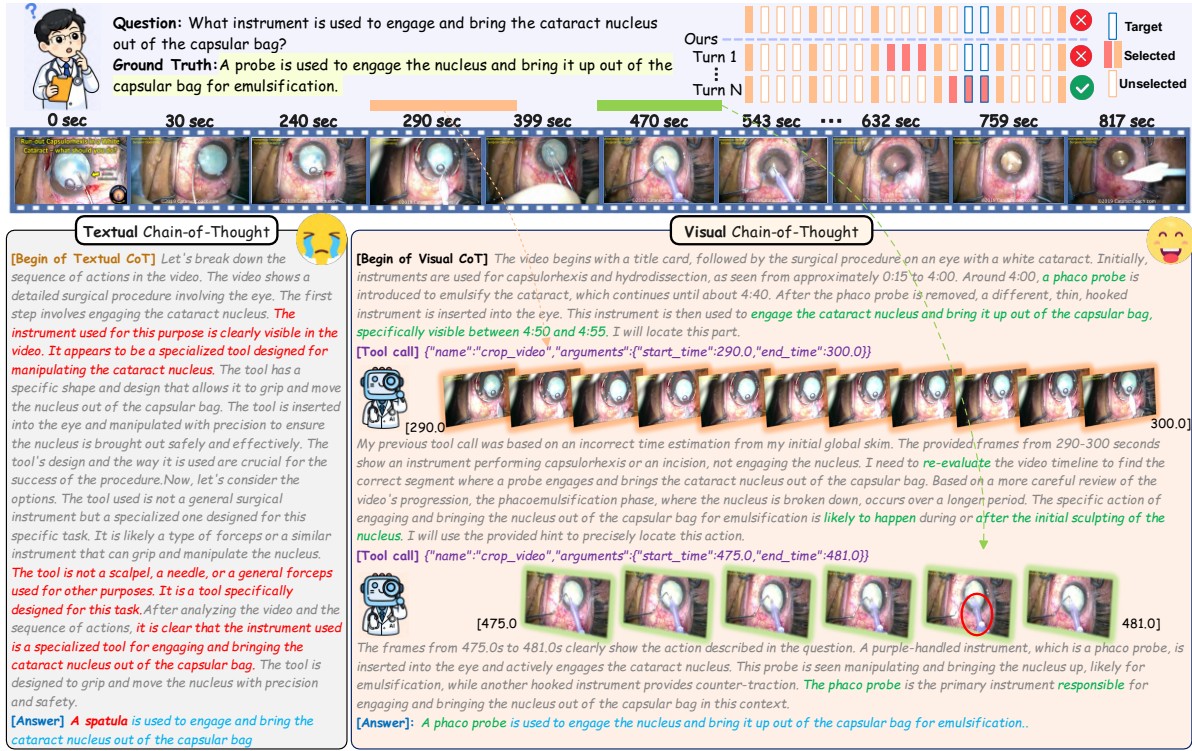

*Figure 2.* **Comparison between textual CoT and visual CoT for evidence-grounded clinical video reasoning.** Left: textual CoT shows overconfident hallucinations (red), inventing rationales and predicting the wrong instrument. Right: visual CoT iteratively retrieves and integrates dense visual evidence via tool calls, grounding reasoning in localized observations and producing the correct answer.

pipeline that treats the visual stream as static context and does not support explicit evidence seeking or iterative verification(Feng et al., 2025; Li et al., 2025; Liu et al., 2025b). This design is fragile for long videos, where sparse sampling can miss decisive moments and encourage plausible but unsupported rationales(Yao et al., 2025b). This raises a key question: can LMMs learn to search and verify evidence over time for long-form medical videos?

We answer this question with an interleaved visual CoT (VCoT) framework that moves beyond text-centric reasoning by enabling LMMs to couple hypothesis-driven reasoning with on-demand, coarse-to-fine temporal localization and iterative verification in long clinical videos(Yang et al., 2025; Zhang et al., 2025b). Inspired by how clinicians skim globally and then revisit locally, the model first forms a coarse understanding and then dynamically selects where to look next, repeatedly re-inspecting candidate moments and revising its hypothesis when needed. As shown in Figure 2, this behavior arises from the LMM's native temporal grounding ability and is realized through a lightweight visual toolbox that supports both clip-level densification and frame-level verification(Ge et al., 2025; He et al., 2025), enabling iterative evidence seeking rather than one-pass inference. In practice, this paradigm strengthens LMMs' long-video understanding, as evidenced in Figure 1.

In this paper, we present **MedScope**, a tool-using clinical video reasoning model that enables "think with videos" through iterative evidence seeking and verification over long-form procedures. However, existing medical video datasets still lack dense evidence-linked supervision and standardized evaluation for verification-driven reasoning. To bridge this gap, we introduce **ClinVideoSuite**, a large-scale evidence-centric suite with grounded QA and tool-augmented trajectories, including **ClinVideo-QA-254K** and **ClinVideo-VCoT-34K**, plus caption and CoT subsets. Building on ClinVideoSuite, we train MedScope in three stages to elicit doctor-like coarse-to-fine verification. We first perform clinical reasoning warm-up to learn medical semantics and long-horizon reasoning. We then conduct visual CoT cold-start supervised fine-tuning (SFT) to teach when additional evidence is needed and how to acquire it via native tools for clip-level densification and frame-level verification. Finally, we propose **GA-GRPO**, an agentic RL recipe with a grounding-aware reward and adaptive advantage reweighting to encourage temporally aligned tool use and stable learning under diverse video conditions.

Our contributions are summarized as follows:

- We propose a medical visual CoT paradigm for "thinking with videos", and instantiate it with **MedScope**, a tool-using video LMM trained via a three-stage

pipeline that elicits coarse-to-fine clinical reasoning.

- We build an evidence-centric data suite, **Clin-VideoSuite**, for training and evaluation, including **ClinVideo-Eval**, a fine-grained long-form medical video reasoning benchmark with traceable evidence.

- We introduce **GA-GRPO**, an agentic RL recipe with grounding-aware rewards and adaptive advantages to promote temporally aligned tool use.

- MedScope achieves **state-of-the-art** performance among **open-source** models on standard medical video benchmarks under both in-domain and out-of-domain evaluations.

## 2. Related Work

### 2.1. Long-Video Reasoning in MLLMs

Reasoning over long videos remains challenging for multimodal large language models (MLLMs) due to high computation and sparse task-relevant evidence (Zou et al., 2024; Tang et al., 2025). Iterative temporal selection methods such as MIST (Gao et al., 2023) and SeViLA (Yu et al., 2023) improve efficiency by localizing segments before answering, but largely decouple localization from reasoning. Long-context MLLMs including LLaMA-VID (Li et al., 2024b), Flash-VStream (Zhang et al., 2024a), LongVA (Zhang et al., 2024b), and LongVILA (Chen et al., 2024a) scale via token compression or long-context modeling, yet still rely on passive sampling and remain costly. Coarse-to-fine sampling strategies (Jeoung et al., 2024; Yao et al., 2025a) further improve efficiency by selecting informative frames on demand. Additionally, reinforcement learning has been explored for video reasoning (e.g., Video-R1 (Feng et al., 2025), VideoChat-R1 (Li et al., 2025), LongVILA-R1 (Chen et al., 2025)), but these methods often use outcome-level rewards under weak supervision, which can misalign intermediate reasoning with visual evidence. In contrast, our work integrates agentic interaction with RL to tightly couple coarse-to-fine visual exploration with reasoning for efficient, grounded medical long-video understanding.

### 2.2. Tool-Augmented Medical MLLMs

Augmenting medical MLLMs with external tools is an effective way to extend standalone reasoning by leveraging specialized perception modules and expert systems. Early works such as MMedAgent (Li et al., 2024a) and VILA-M3 (Nath et al., 2025) train models to invoke tools for tasks like segmentation and classification, while AURA (Fathi et al., 2025) unifies multiple domain-specific models, including MedSAM (Ma et al., 2024) and CheXAgent (Chen et al., 2024b), in a single tool-augmented framework. Despite strong task-level performance, these systems often rely on fragmented tool calls that limit coherent planning

and reflective reasoning. To better support multi-step clinical decision-making, MedAgent-Pro (Wang et al., 2025e) and SMR-Agents (Wang et al., 2026) adopt hierarchical or collaborative multi-agent designs, but their dependence on predefined workflows and prompt-level orchestration can hinder generalization and robustness to tool failures. More recently, inspired by the "thinking with images" paradigm, iterative tool-mediated visual reasoning has been shown to improve grounding and reduce hallucinations (Jiang et al., 2025). However, existing tool-augmented medical MLLMs mainly focus on static images and do not address the dynamic visual exploration and evidence accumulation required for long-horizon medical video reasoning (Jiang et al., 2026). In this work, we move beyond image-centric tool use and learn tool-mediated, temporally adaptive evidence gathering for clinical long-video reasoning.

## 3. Method

### 3.1. Overview

MedScope enables "thinking with videos" via coarse-to-fine clinical reasoning with a lightweight visual toolbox, alternating textual reasoning and tool-based evidence acquisition to verify predictions with temporally targeted dense observations (Section 3.2). To support this behavior, we build ClinVideoSuite with evidence-centric captions, QA pairs tied to localized supervision windows, and environment-interactive visual-CoT trajectories produced by native tools (Section 3.3). We then train the model in three stages: warmup, visual-CoT cold-start, and grounding-aware GRPO, so that tool use is both effective and evidence-grounded (Section 3.4).

### 3.2. Coarse-to-Fine Clinical Reasoning Framework

As shown in Figure 3(a), MedScope performs coarse-to-fine clinical video reasoning by alternating between hypothesis-driven reasoning and tool-based evidence retrieval, producing an explicit trajectory for verification.

**Textual and visual rationalization.** We decouple reasoning from evidence acquisition. At round $k$, the model outputs a textual rationale $t_{i,k}$ that states its current belief and the evidence it seeks, then invokes an action $a_{i,k}$ to retrieve an observation $o_{i,k}$ as visual evidence. This structured separation makes the decision process inspectable and supports iterative coarse-to-fine verification.

**Multi-round generation and trajectory.** Given a query $q_i$ and video $v_i$, the model generates a multi-round trajectory:

$$\tau_i = \{(t_{i,k}, a_{i,k}, o_{i,k})\}_{k=1}^{K_i}, \qquad (1)$$

where $t_{i,k}$ denotes the textual rationale, $a_{i,k}$ the selected tool action, and $o_{i,k}$ the returned visual evidence. At each round,

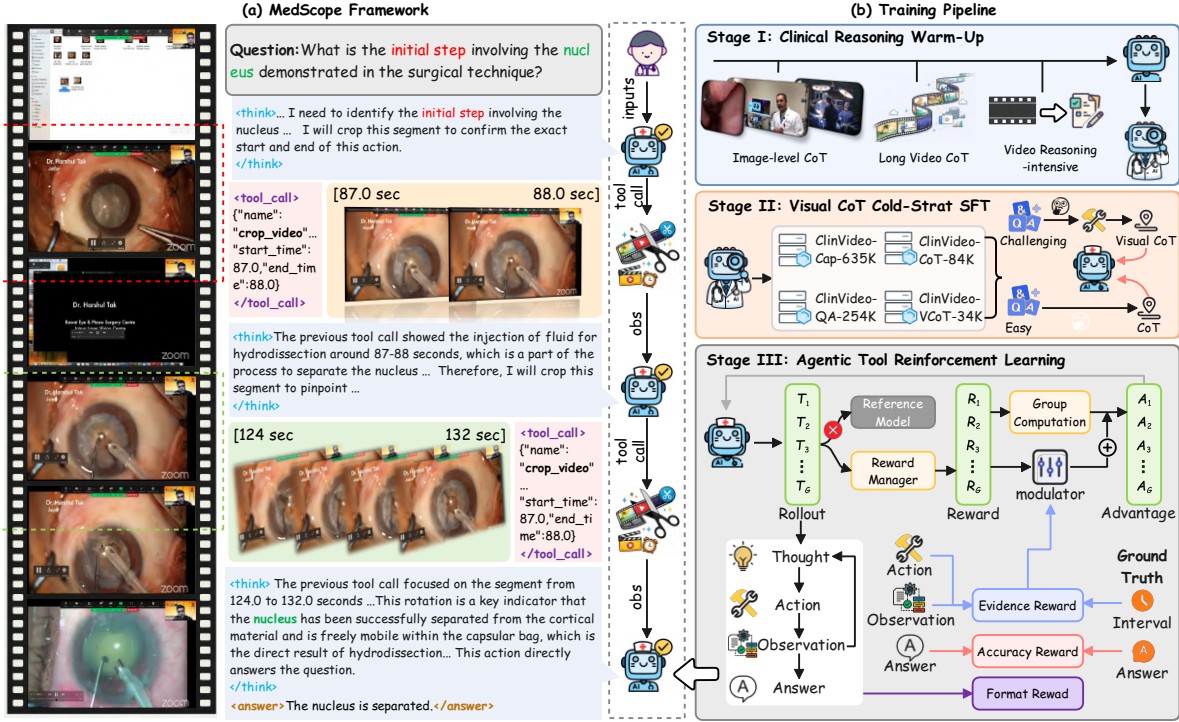

*Figure 3.* **Overview of MedScope.** (a) Coarse-to-fine clinical reasoning with explicit thought and tool actions that progressively retrieve temporally targeted dense evidence for verification. (b) Three-stage training pipeline: clinical reasoning warm-up, visual-CoT cold-start SFT on ClinVideoSuite, and agentic tool reinforcement learning with grounding-aware rewards and advantage shaping.

the policy predicts $(t_{i,k}, a_{i,k}) = \pi_\theta(c_{i,k})$ conditioned on the accumulated context $c_{i,k}$. If $a_{i,k}$ invokes a tool, the environment returns $o_{i,k} = \mathcal{E}(v_i, a_{i,k})$, and we update the context as $c_{i,k+1} = c_{i,k} \oplus (t_{i,k}, a_{i,k}, o_{i,k})$ for the next step. We enforce a structured interface within each trajectory: <think> for textual rationalization, <tool_call> for visual rationalization, <tool_response> for tool observations, and <answer> for the terminal prediction.

**Visual Toolbox** More complex and diverse examples of the model's reasoning process are provided in Appendix G. In this paper, the action space includes two native tools that support hierarchical inspection:

- crop_video: takes a video and a time window, and returns a clip-level densely sampled frame sequence within the window for localized evidence gathering.

- get_frame: takes a video and a timestamp, and returns the three frames nearest to that timestamp for fine-grained verification.

### 3.3. ClinVideoSuite: Evidence-Centric Data Synthesis

To address the data bottleneck for think-with-videos in clinical videos, we introduce **ClinVideoSuite**, a large-scale, high-fidelity, and video-grounded suite that emphasizes dense, temporally localized visual cues for videoQA and

temporal grounding (Figure 4). Built from MedVideo-Cap (Wang et al., 2025c), OphVL (Hu et al., 2025), and SurgVidLM (Wang et al., 2025a) via captioning, high-difficulty QA construction, and rationalization synthesis, ClinVideoSuite comprises **ClinVideo-Cap**, **ClinVideo-QA**, **ClinVideo-VCoT**, **ClinVideo-CoT**, and **ClinVideo-RL**, which together support training and evaluation of grounded clinical video reasoning.

**Stage 1: Evidence-centric captioning and global summarization.** Given a video $V$ with duration $T$, we construct timestamped dense captions $\mathcal{C}(V)$ as localized evidence and a coarse global summary $S(V)$ as context. We first sparsify $V$ by low-rate sampling $\tilde{V} = \mathcal{D}_\rho(V)$, where $\mathcal{D}_\rho$ samples at rate $\rho$. A segmenter $\Phi$ then predicts an entity-guided partition:

$$\mathcal{B}(V) = \Phi(\tilde{V}) = \{(e_m, s_m, u_m)\}_{m=1}^M, \qquad (2)$$

where $e_m$ is an entity cue and $[s_m, u_m] \subseteq [0, T]$ is the associated window. We densify each window to generate fine-grained descriptions, forming $\mathcal{C}(V) = \{(d, t)\}$ with $t \in [0, T]$, and summarize temporally ordered captions into $S(V) = \mathrm{Sum}(\mathrm{Merge}(\{d\}))$, which preserves global context while leaving localized cues to be retrieved from $\mathcal{C}(V)$ for downstream grounding.

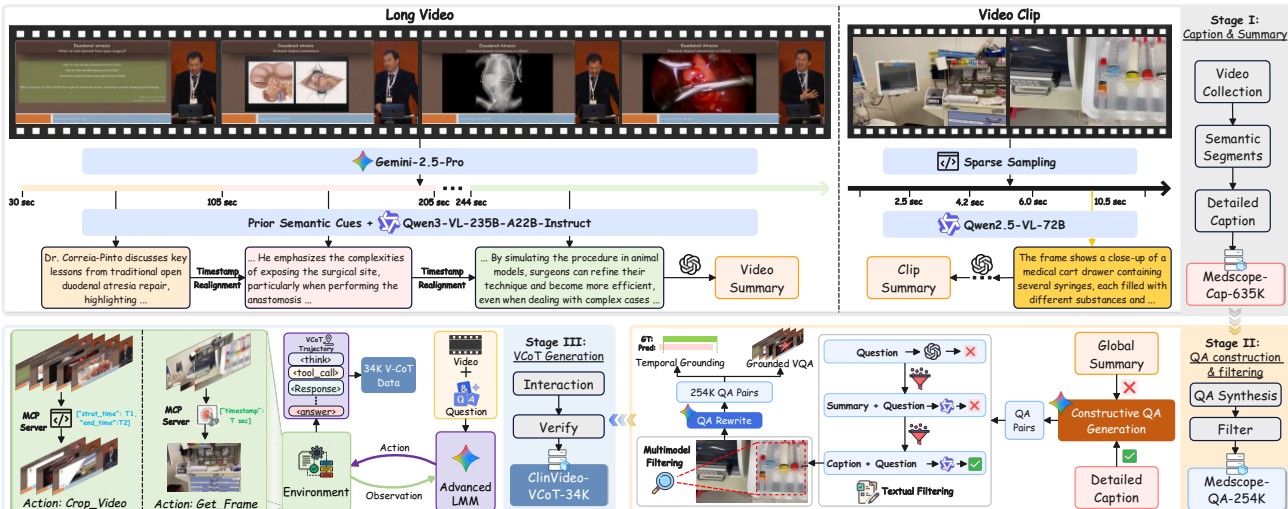

*Figure 4.* **ClinVideoSuite data synthesis pipeline.** Stage 1 builds evidence-centric dense captions and global summaries. Stage 2 generates and filters QA with text checks and multimodal verification to enforce localized evidence dependence. Stage 3 collects tool-augmented visual CoT trajectories via native tool interaction in a real video environment.

**Stage 2: High-quality QA construction with cross-model filtering.** In this stage, we construct open-ended QA pairs that hinge on temporally localized visual cues. For each window $w = [s, e]$, we derive a local view $\mathcal{C}(V; w)$ from dense captions and prompt a generator to produce candidate pairs $(q, a)$ that target details in $\mathcal{C}(V; w)$ but absent from the global summary $S(V)$, while inheriting the same supervision window $w$.

**Three-layer text filtering.** Raw candidates pass a voting-based filtering cascade designed to remove knowledge-driven questions, summary-solvable shortcuts, and internally inconsistent pairs. Let $\mathcal{M}$ be a pool of LLMs, and let $\mathrm{Judge}(\cdot, \cdot)$ judge whether two answers are equivalent. We define a consensus score under a context view $X$:

$$p_X(q, a) = \frac{1}{|\mathcal{M}|} \sum_{M \in \mathcal{M}} \mathbf{1}\left[\mathrm{Judge}(\mathrm{Ans}(M, q, X), a)\right], \quad (3)$$

where $\mathrm{Ans}(M, q, X)$ is model $M$ answering $q$ given $X$. Filter A removes knowledge-intensive or guessable items by requiring low text-only consensus, $p_\varnothing(q, a) < \theta_{\text{text}}$. Filter B removes global-summary shortcuts by requiring low consensus given only $S(V)$, $p_{S(V)}(q, a) < \theta_{\text{sum}}$. Filter C enforces local-evidence dependence by requiring high consensus under local dense captions $\mathcal{C}(V; w)$, $p_{\mathcal{C}(V;w)}(q, a) \geq \theta_{\text{loc}}$.

**Multimodal confirmation.** To eliminate caption hallucinations and ensure genuine video grounding, we perform a final multimodal check that answers $q$ directly from the video segment $\mathrm{Clip}(V, w)$ and retains only pairs with strong agreement against $a$. The surviving items form ClinVideo-QA-254K, and their associated dense evidence supports downstream visual rationalization synthesis.

**Stage 3: Environment-interactive visual-CoT synthesis.** We synthesize visual-CoT trajectories via real interaction with the video environment. A teacher model starts from sparse global context and decides whether to query native tools for temporally targeted observations, then continues reasoning with the observation; otherwise it answers directly. This yields both tool-augmented and tool-free trajectories, improving synthesis efficiency while retaining evidence-seeking behaviors. Prompting details are in Appendix H.

**Dataset Statistics.** We summarize statistics of **ClinVideo-Cap-635K** and **ClinVideo-QA-254K**, which form the core of ClinVideoSuite, while remaining subsets are derived from **ClinVideo-QA-254K** under different reasoning and interaction settings. As shown in Appendix Figures 7–9, **ClinVideo-Cap-635K** provides large-scale timestamped dense captions with diverse temporal densities across sources, and **ClinVideo-QA-254K** offers concise, evidence-grounded QA pairs with localized supervision windows. We provide detailed analyses and visualizations in Appendix A.

### 3.4. Training Strategy

As shown in Figure 3(b), we adopt a three-stage pipeline for grounded clinical video reasoning: (i) warm up to learn medical semantics and long-horizon reasoning, (ii) tool-augmented visual-CoT cold start to learn when and how to invoke tools, and (iii) GA-GRPO agentic RL to refine tool use via grounding-aware rewards and advantage modulation.

**Clinical Reasoning Warm-Up** We warm up the backbone with multi-source SFT to build general multimodal reasoning, long-video understanding, and surgical visual se-

mantics before introducing tool use. Specifically, we merge (i) Video-R1 video CoT for step-by-step multimodal inference (Feng et al., 2025), (ii) a curated subset of LongVideo-Reason for long-horizon temporal reasoning (Chen et al., 2025), and (iii) surgical image-level VQA from EndoVis-18-VQA (Bai et al., 2023), PitVQA (He et al., 2024b), and Surgical-VQA (Seenivasan et al., 2022). For the image-level data, we apply rejection sampling to retain only format-compliant, visually grounded rationales. Detailed training objectives are provided in Appendix B.1.

**Visual CoT Cold-Start Supervised Fine-Tuning** We then perform a cold-start SFT on our visual-CoT data **ClinVideo-VCoT-34K** and **ClinVideo-CoT-84K** to teach coarse-to-fine clinical verification in videos. This stage mixes tool-augmented trajectories that retrieve evidence from clips via `get_frame` and from long videos via `crop_video` with tool-free trajectories for cases that are already resolvable from sparse frames. Training on this mixture promotes adaptive reasoning, helping the model learn when additional evidence is necessary and how to incorporate observations into a faithful rationale.

**Grounding-Aware GRPO Training** We further apply agentic RL to extend coarse-to-fine clinical verification beyond supervised traces. Specifically, we instantiate GRPO with tool-aligned grounding rewards, reinforcing rollouts that are correct and temporally well-localized.

**Reward Design.** We use a composite reward $R = R_{\text{acc}} + R_{\text{format}} + R_{\text{evidence}}$, where $R_{\text{acc}}$ scores answer correctness via an LLM judge and $R_{\text{format}}$ checks the required structured interface. The evidence term $R_{\text{evidence}}$ evaluates grounding based on the final tool call, using temporal overlap for `crop_video` and timestamp alignment for `get_frame`. For `crop_video`, we additionally apply an IoU-based bonus schedule on top of the base overlap reward, which further encourages precise temporal alignment as IoU increases. Algorithm 1 summarizes the procedure, and the full mathematical definition is deferred to Appendix B.2.

**Objective** We optimize GA-GRPO under the GRPO objective without the KL term, directly maximizing grounding-aligned returns. We use a group-relative advantage modulated by the tool-grounding signal; the full objective and notation are provided in Appendix B.2.

**Trajectory-Level Fidelity-Weighted Advantage.** To encourage grounded tool use, we modulate each trajectory advantage by tool-outcome fidelity. We set $f_i = R_{\text{evidence}}$ and let $m_i \in \{0, 1\}$ indicate tool usage. We define:

$$\hat{A}_i = A_i \cdot h(\tau_i), \quad (4)$$

where $h(\tau_i)$ keeps tool-free trajectories unchanged and ad-

---

**Algorithm 1** Grounding-Aware Tool Reward

**Require:** Prediction $y$, GT $[g_s, g_e]$ or $g_f$, tool calls $T$
**Ensure:** $R = R_{\text{acc}} + R_{\text{format}} + R_{\text{evidence}}$
0: $R_{\text{acc}} \leftarrow \text{Judge}(y)$
0: $R_{\text{format}} \leftarrow \text{FormatOK}(y)$
0: $R_{\text{evidence}} \leftarrow 0$
0: **if** crop_video $\in T$ **then**
0: $\quad (p_s, p_e) \leftarrow \text{LastCrop}(T)$
0: $\quad r_c \leftarrow \text{CropReward}(p_s, p_e)$
0: $\quad R_{\text{evidence}} \leftarrow R_{\text{evidence}} + r_c$
0: **end if**
0: **if** get_frame $\in T$ **then**
0: $\quad t \leftarrow \text{LastFrame}(T)$
0: $\quad r_f \leftarrow \text{FrameReward}(t)$
0: $\quad R_{\text{evidence}} \leftarrow R_{\text{evidence}} + r_f$
0: **end if**=0

---

justs tool-using ones by fidelity and optimization direction:

$$\tilde{f}_i = \text{clip}(f_i, -c, c), \quad (5)$$

$$h(\tau_i) = \begin{cases} \max(1 + \alpha\tilde{f}_i, s_{\min}), & A_i \geq 0, m_i = 1, \\ \max(1 - \alpha\tilde{f}_i, s_{\min}), & A_i < 0, m_i = 1, \\ 1, & m_i = 0, \end{cases} \quad (6)$$

with $\alpha > 0$, clipping bound $c$, and floor $s_{\min} > 0$. Intuitively, higher fidelity boosts positive advantages and softens negative ones, avoiding over-penalizing grounded attempts while still discouraging ungrounded tool use.

## 4. Experiments

### 4.1. Experimental Setup

**Benchmarks** We evaluate our model on two medical video benchmarks covering multi-grained video understanding, visual reasoning, and grounded clinical question answering. The first benchmark, **SVU-31K** (Wang et al., 2025a), focuses on multi-grained video description and visual reasoning, including Full and Fine-grained Video Description as well as Fine-grained Temporal and Perception Visual Reasoning. The second benchmark, **ClinVideo-Eval**, evaluates grounded medical video understanding under both in-domain and out-of-domain settings, spanning OphVL, Med-VideoCap, and SurgVidLM. It supports Temporal Grounding and Grounded VQA tasks, enabling systematic evaluation of generalization from in-domain to out-of-domain scenarios. Additional details are provided in the Appendix D.

**Implementation Details.** We take Qwen2.5-VL-7B-Instruct as the base model for all experiments. For SFT, we use `LMMs-Engine` (LMMs-Lab, 2025), while RL is implemented with `verl` (Sheng et al., 2024) and rollouts are generated using `SGLang` (Zheng et al., 2024). For cold-start SFT, we jointly train on **ClinVideo-CoT** and **ClinVideo-**

*Table 1.* **Comparison of MedScope with zero-shot LMMs across multi-grained video understanding tasks on SVU-31K.** CI, DO, CU, and TU denote correctness of information, detail orientation, contextual understanding, and temporal understanding, respectively. Best results are in **bold** and second-best results are underlined.

| Model | Tool Use | Reasoning | Full Video Description | | | | Fine-grained Video Description | | | |
| --- | --- | --- | --- | --- | --- | --- | --- | --- | --- | --- |
| | | | CI | DO | CU | TU | CI | DO | CU | TU |
| Video-LLaVA-7B | ✗ | ✗ | 1.03 | 1.01 | 1.01 | 1.03 | 1.15 | 1.09 | 1.24 | 1.21 |
| GroundingGPT-7B | ✗ | ✗ | 1.05 | 1.06 | 1.15 | 1.08 | 1.05 | 1.16 | 1.26 | 1.16 |
| VideoLLaMA3-7B | ✗ | ✗ | 1.66 | 1.71 | 2.21 | 1.86 | 1.58 | 1.87 | 2.29 | 1.95 |
| Qwen2.5-VL-7B | ✗ | ✗ | 1.22 | 1.09 | 1.64 | 1.53 | 1.82 | 1.77 | 2.49 | 1.95 |
| InternVL3-8B | ✗ | ✗ | 1.34 | 1.11 | 1.52 | 1.63 | 1.76 | 1.67 | 2.31 | 2.02 |
| SurgVidLM | ✗ | ✗ | 2.40 | 2.11 | 2.69 | 2.22 | 2.27 | 2.62 | 3.06 | 2.44 |
| LongVT-7B-RFT | ✓ | ✓ | 3.96 | 4.10 | 3.90 | 3.87 | 3.77 | 4.10 | 3.78 | 3.76 |
| VideoR1-7B | ✗ | ✓ | 4.25 | 4.33 | 4.25 | 4.24 | 3.80 | 4.01 | 3.79 | 3.77 |
| VideoChat-R1-7B | ✓ | ✓ | 4.01 | 4.12 | 3.92 | 4.02 | 3.30 | 3.41 | 3.29 | 3.37 |
| VideoRFT-7B | ✓ | ✓ | 4.45 | 4.52 | 4.44 | 4.43 | 3.46 | 3.61 | 3.50 | 3.53 |
| MedScope | ✓ | ✓ | **4.76** | **4.77** | **4.77** | **4.75** | **4.00** | **4.36** | **4.15** | **4.08** |

| Model | Tool Use | Reasoning | Fine-grained Temporal Visual Reasoning | | | | Fine-grained Perception Visual Reasoning | | | |
| --- | --- | --- | --- | --- | --- | --- | --- | --- | --- | --- |
| | | | BLEU-4 | CIDEr | METEOR | ROUGE-L | BLEU-4 | CIDEr | METEOR | ROUGE-L |
| Video-LLaVA-7B | ✗ | ✗ | 5.65 | 2.17 | 11.59 | 19.73 | 10.76 | 5.48 | 13.52 | 25.93 |
| GroundingGPT-7B | ✗ | ✗ | 2.81 | 1.44 | 10.52 | 17.86 | 4.47 | 1.51 | 11.96 | 21.34 |
| VideoLLaMA3-7B | ✗ | ✗ | 7.68 | 1.92 | 11.53 | 23.19 | 9.57 | 3.97 | 12.63 | 24.21 |
| Qwen2.5-VL-7B | ✗ | ✗ | 4.41 | 1.55 | 13.09 | 16.27 | 3.19 | 2.18 | 15.04 | 25.55 |
| InternVL3-8B | ✗ | ✗ | 3.83 | 1.13 | 11.28 | 18.32 | 2.93 | 1.89 | 12.20 | 26.72 |
| SurgVidLM | ✗ | ✗ | 10.10 | 3.83 | 14.13 | 31.58 | 16.71 | 9.76 | 18.27 | 37.51 |
| LongVT-7B-RFT | ✓ | ✓ | 6.22 | 1.77 | 14.99 | 21.26 | 5.41 | 2.00 | 19.53 | 25.59 |
| VideoR1-7B | ✗ | ✓ | 8.25 | 3.13 | 14.23 | 24.41 | 7.45 | 5.13 | 13.91 | 23.34 |
| VideoChat-R1-7B | ✓ | ✓ | 4.30 | 1.56 | 11.89 | 19.83 | 5.90 | 1.93 | 17.62 | 24.54 |
| VideoRFT-7B | ✓ | ✓ | 6.45 | 2.52 | 11.44 | 20.43 | 6.46 | 3.61 | 16.90 | 23.78 |
| MedScope | ✓ | ✓ | **11.30** | **4.56** | **16.89** | **34.83** | **18.90** | **10.93** | **21.62** | **41.54** |

*Table 2.* **Performance comparison on ClinVideo-Eval.** We report in-domain results on OphVL and MedVideoCap and out-of-domain generalization on SurgVidLM for temporal grounding and grounded VQA. Agent baselines can invoke tools, while other models do not.

| Model | In-domain | | | | | | | | Out-of-domain | | | | | |
| --- | --- | --- | --- | --- | --- | --- | --- | --- | --- | --- | --- | --- | --- | --- |
| | OphVL ($\approx$ 371147 sec) | | | | | | MedVideoCap ($\approx$ 9575 sec) | | SurgVidLM ($\approx$ 1320694 sec) | | | | | |
| | Temporal Grounding | | | | Grounded VQA | | Grounded VQA | | Temporal Grounding | | | | Grounded VQA | |
| | R@0.3 | R@0.5 | R@0.7 | mIoU | mIoU | Acc | mIoU | Acc | R@0.3 | R@0.5 | R@0.7 | mIoU | mIoU | Acc |
| *Proprietary LMMs* | | | | | | | | | | | | | | |
| GPT-4o | 41.73 | 34.33 | 29.23 | 35.75 | 40.08 | 9.79 | 88.80 | 29.14 | 15.90 | 9.00 | 5.65 | 13.35 | 15.90 | 6.85 |
| Gemini-2.5-Flash | 59.33 | 48.77 | 13.50 | 68.31 | 43.38 | 12.59 | 76.74 | 26.96 | 43.51 | 33.89 | 23.64 | 33.95 | 12.94 | 4.36 |
| Gemini 3-Pro-Preview | 87.85 | 73.06 | 57.57 | 72.43 | 73.81 | 34.12 | 96.46 | 32.31 | 45.19 | 31.59 | 22.80 | 37.35 | 35.28 | 21.99 |
| *Open-Source LMMs* | | | | | | | | | | | | | | |
| Qwen2.5-VL-7B-Instruct | 43.45 | 41.16 | 21.94 | 30.18 | 32.91 | 4.37 | 56.87 | 25.86 | 27.61 | 21.94 | 15.31 | 19.15 | 11.98 | 3.73 |
| InternVL3-8B | 42.15 | 36.65 | 22.57 | 29.01 | 39.48 | 9.79 | 57.62 | 26.61 | 20.92 | 14.23 | 10.88 | 18.55 | 16.6 | 3.53 |
| VideoLLaMA3-7B | 37.36 | 25.01 | 14.39 | 23.10 | 20.78 | 1.34 | 42.14 | 15.89 | 12.67 | 10.08 | 5.21 | 9.37 | 8.55 | 1.93 |
| *Reasoning LMMs* | | | | | | | | | | | | | | |
| Video-R1-7B | 63.94 | 22.89 | 7.04 | 33.87 | 41.60 | 27.45 | 38.37 | 34.55 | 41.18 | 23.07 | 9.42 | 31.45 | 31.61 | 21.05 |
| VideoChat-R1-7B | 30.58 | 21.41 | 9.17 | 30.93 | 32.40 | 14.68 | 23.13 | 33.69 | 46.87 | 26.04 | 3.15 | 21.43 | 29.29 | 15.97 |
| Video-RFT | 26.41 | 19.37 | 7.04 | 27.06 | 28.7 | 15.24 | 31.08 | 28.47 | 60.67 | 23.01 | 4.18 | 43.26 | 29.21 | 19.3 |
| *Reasoning Agents* | | | | | | | | | | | | | | |
| LongVT-7B-RFT | 48.73 | 32.32 | 23.52 | 39.56 | 42.17 | 14.16 | 55.34 | 31.22 | 51.13 | 37.66 | 16.73 | 39.74 | 25.72 | 17.02 |
| ReWatch-R1-7B | 45.85 | 30.56 | 26.52 | 24.77 | 36.20 | 16.78 | 35.25 | 25.48 | **69.96** | 39.75 | 10.46 | **47.17** | 25.47 | 21.99 |
| MedScope-7B-SFT | 67.19 | 54.43 | 43.72 | 49.28 | 53.28 | 25.03 | 52.7 | 32.2 | 52.04 | 38.75 | **22.97** | 41.63 | 32.13 | 35.2 |
| MedScope-7B-RL | **81.92** | **68.87** | **54.35** | **64.42** | **65.58** | **35.10** | **61.20** | **39.80** | 55.60 | **40.10** | 22.35 | 44.30 | **34.10** | **37.90** |
| Δ (vs Qwen2.5-VL-7B) | +38.5 | +27.7 | +32.4 | +34.2 | +32.7 | +30.7 | +4.3 | +13.9 | +28.0 | +18.2 | +7.0 | +25.1 | +22.1 | +34.2 |

**VCoT** for one epoch using 32 H200 GPUs. For RL, we train on **ClinVideo-RL** for one epoch using 64 H200 GPUs. Additional optimization settings and hyperparameters are provided in the Appendix E.

**Baseline Methods & Evaluation Metrics** We compare MedScope with diverse baselines, including proprietary models, open-source video-language models, reasoning models, and tool-using agents. We follow standard medical video protocols to evaluate video description, visual reasoning, and grounded understanding in both in-domain and out-of-domain settings. Full details are provided in Appendix F.

## 4.2. Main Results

Table 1 reports results on SVU-31K across multi-grained video description and fine-grained temporal and perceptual visual reasoning. MedScope ranks first on both full-video and fine-grained description, achieving 4.77 DO on full-video description and 4.36 DO on fine-grained description, and consistently surpassing strong reasoning baselines. It also outperforms tool-using agents such as LongVT-7B-RFT as well as prior reasoning models including VideoRFT-7B and VideoR1-7B, showing that tool use is most effective when paired with verification-driven training. On fine-grained reasoning, MedScope sets a new state of the art, reaching 4.56 CIDEr on temporal reasoning and 10.93 CIDEr on perceptual reasoning, with clear gains over the medical baseline SurgVidLM. Overall, MedScope delivers robust improvements from global narrative fidelity to localized evidence reasoning on SVU-31K.

Table 2 reports results on the ClinVideo-Eval benchmark for temporal grounding and grounded VQA under in-domain and out-of-domain settings. MedScope-7B-RL achieves the strongest open-source performance overall, surpassing both reasoning models and tool-using agents. It reaches 64.42 mIoU for temporal grounding and 65.58 mIoU for grounded VQA on OphVL, and attains 61.20 mIoU and 39.80 accuracy on MedVideoCap, which contains shorter videos. On SurgVidLM, MedScope remains robust out of domain and competitive with closed-source models, while consistently improving over open-source baselines.

**MedScope also generalizes to established surgical VQA benchmarks.** To further assess whether MedScope generalizes beyond our own evaluation suite, we additionally evaluate it on two established short-clip surgical VQA benchmarks, Cholec80-VQA and EndoVis18-VQA. As shown in Table 4, MedScope achieves the best performance on both benchmarks, reaching 94.6 accuracy on Cholec80-VQA and 71.8 recall on EndoVis18-VQA. Although these benchmarks mainly contain short clips and therefore do not directly evaluate long-video evidence seeking, the results show that MedScope transfers well to existing medical video QA protocols and is not only effective on ClinVideo-Eval.

## 4.3. Ablation Studies

**Warm-up supplies basic clinical reasoning, SFT learns tool use, and RL optimizes verification decisions.** Table 3

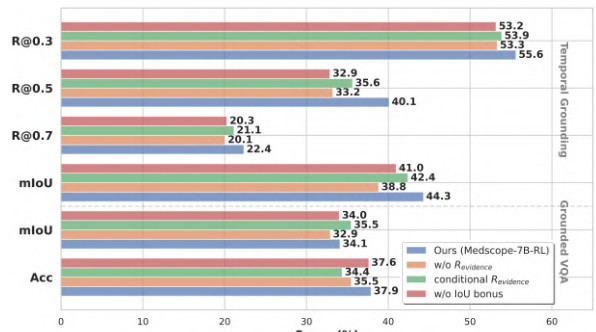

*Figure 5.* **Ablations on reward design.** We compare **Ours** with three variants: **w/o $R_{\text{evidence}}$** removes the evidence reward; **conditional $R_{\text{evidence}}$** applies it only when $R_{\text{acc}}=1$; **w/o IoU bonus** removes the continuous IoU bonus in $R_{\text{evidence}}$.

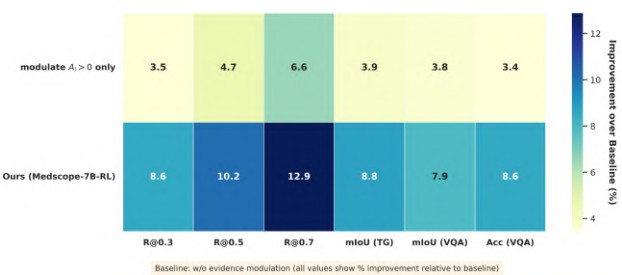

*Figure 6.* Comparison of advantage modulation strategies relative to the baseline. **modulate $A_i > 0$ only** applies modulation only to positive-advantage trajectories. The baseline is **w/o evidence modulation**.

(top) shows that Warm-up alone brings limited gains, while removing SFT severely degrades tool-conditioned verification, indicating that RL by itself is insufficient to induce reliable evidence seeking and often converges to shallow, instruction-following behaviors with fewer tool calls. Warm-up+SFT yields consistent improvements by imitating correct tool trajectories, but remains more vulnerable to distribution shift without decision-level refinement. Adding RL on top of SFT further strengthens evidence-grounded decision making, consolidating when to retrieve additional evidence and when to commit, and delivers the best overall results across both temporal and perceptual reasoning.

**High-quality VCoT cold-start trajectories are critical for effective RL.** Table 3 (bottom) shows that removing VCoT during SFT consistently hurts downstream reasoning. Without VCoT, the model learns weaker tool-conditioned evidence aggregation and fine-grained perception, leading to uniformly lower scores. The gap further widens after RL: starting RL from VCoT-free SFT yields only limited gains, suggesting unstable tool invocation and poorer credit assignment. In contrast, VCoT-enabled cold-start provides a reliable initialization for RL, producing the best overall performance across both temporal and perceptual reasoning.

*Table 3.* **Ablations on training stages and VCoT supervision. Top: Training stages.** We ablate Warm-up, SFT, and RL and evaluate combinations on SVU-31K. **Bottom: VCoT supervision.** We remove VCoT trajectories during SFT and compare SFT-only and SFT+RL.

| Setting | Training Components | | | Fine-grained Temporal Visual Reasoning | | | | | Fine-grained Perception Visual Reasoning | | | | |
|---|---|---|---|---|---|---|---|---|---|---|---|---|---|
| | Warm-up | SFT | RL | BLEU-4 | CIDEr | METEOR | ROUGE-L | Tool | BLEU-4 | CIDEr | METEOR | ROUGE-L | Tool |
| *Training Components* | | | | | | | | | | | | | |
| Qwen2.5-VL-7B-Instruct | ✗ | ✗ | ✗ | 4.41 | 1.55 | 13.09 | 16.27 | 7.8% | 3.19 | 2.18 | 15.04 | 25.55 | 6.4% |
| Warm-up only | ✓ | ✗ | ✗ | 5.13 | 2.16 | 12.98 | 17.36 | 8.6% | 7.01 | 4.54 | 16.40 | 26.72 | 6.9% |
| Warm-up + SFT | ✓ | ✓ | ✗ | 7.64 | 2.59 | 14.51 | 23.43 | 46.2% | 11.76 | 5.21 | 17.49 | 35.90 | 34.7% |
| Warm-up + RL | ✓ | ✗ | ✓ | 7.70 | 4.70 | **17.05** | 24.50 | 27.4% | 12.10 | 5.40 | 17.70 | 36.20 | 20.7% |
| Warm-up + SFT + RL (MedScope) | ✓ | ✓ | ✓ | **11.30** | **4.56** | 16.89 | **34.83** | **63.8%** | **18.90** | **10.93** | **21.62** | **41.54** | **51.8%** |
| *Data Recipe: Effect of VCoT* | | | | | | | | | | | | | |
| Warm-up + SFT (w/o VCoT) | ✓ | ✓ | ✗ | 6.92 | 2.31 | 14.06 | 22.35 | 8.9% | 10.88 | 4.93 | 17.12 | 34.70 | 11.0% |
| Warm-up + SFT (w/ VCoT) | ✓ | ✓ | ✗ | 7.64 | 2.59 | 14.51 | 23.43 | 46.2% | 11.76 | 5.21 | 17.49 | 35.90 | 34.7% |
| Warm-up + SFT + RL (w/o VCoT) | ✓ | ✓ | ✓ | 7.95 | 4.13 | 15.24 | 25.40 | 25.5% | 12.60 | 5.80 | 18.05 | 37.10 | 21.3% |
| Warm-up + SFT + RL (w/ VCoT) | ✓ | ✓ | ✓ | **11.30** | **4.56** | **16.89** | **34.83** | **63.8%** | **18.90** | **10.93** | **21.62** | **41.54** | **51.8%** |

*Table 4.* Additional evaluation on established short-clip surgical VQA benchmarks.

| Model | Cholec80-VQA Acc. | EndoVis18-VQA Recall |
|---|---|---|
| VisualBERT | 89.7 | 61.4 |
| Surgical-VQA | 89.8 | 63.2 |
| SurgicalGPT | 94.0 | 66.0 |
| Surgical-LLaVA | 92.2 | 68.7 |
| **MedScope** | **94.6** | **71.8** |

*Table 5.* Physician verification of ClinVideo-QA-254K. Correct denotes cases judged to be factually correct and supported by localized visual evidence.

| Dataset | Total | Correct | Accuracy |
|---|---|---|---|
| MedVideoCap | 500 | 489 | 97.8% |
| OphVL | 500 | 478 | 95.6% |
| SurgVidLM | 500 | 483 | 96.6% |
| Overall | 1,500 | 1,450 | 96.7% |

**Grounding-aware reward design is essential for evidence-faithful temporal reasoning.** Figure 5 reports reward ablations on temporal grounding and grounded VQA. Removing the evidence reward lowers localization quality substantially, with R@0.5 dropping from 40.1 to 33.2 and mIoU from 44.3 to 38.8, showing that answer-only supervision does not provide enough signal to learn reliable evidence selection. Conditioning the evidence reward on answer correctness improves over this variant, but still trails the full design, suggesting that decoupled grounding feedback is important even when intermediate attempts are imperfect. Removing the IoU bonus further weakens precise alignment, most clearly at stricter criteria where R@0.7 decreases from 22.4 to 20.3, highlighting the benefit of continuous, overlap-sensitive feedback for tightening temporal localization.

**Fidelity-weighted advantage modulation plays a key role in stabilizing grounded policy optimization.** Figure 6 compares advantage modulation strategies relative to the baseline without evidence modulation. Applying fidelity-weighted modulation to both positive and negative advantages (**Ours**) consistently outperforms positive-only modulation. In particular, full modulation yields larger gains in temporal grounding, improving R@0.7 by 12.9% compared to 6.6%, and achieves higher mIoU gains (8.8% vs. 3.9%). Similar improvements are observed on grounded VQA, with higher gains in both accuracy (8.6% vs. 3.4%) and localization quality.

## 4.4. Physician Verification

To further assess the quality of ClinVideo-QA-254K, we conducted physician verification on 1,500 randomly sampled cases. Three board-certified physicians independently reviewed 500 cases from one source dataset each. As shown in Table 5, the verified accuracy reaches 97.8% on Med-VideoCap, 95.6% on OphVL, and 96.6% on SurgVidLM, with an overall accuracy of 96.7%. This physician audit provides additional evidence that the generated QA pairs are factually reliable and visually grounded, supporting the use of ClinVideoSuite as a high-quality supervision and evaluation resource.

## 5. Conclusion

We present **MedScope**, a tool-using clinical LMM that enables "think with videos" via coarse-to-fine temporal evidence seeking and frame-level verification in long-form procedures. We introduce **ClinVideoSuite**, a large-scale suite with localized supervision, grounded QA, and environment-interactive visual-CoT trajectories, and propose **GA-GRPO** to reinforce temporally aligned tool use with grounding-aware rewards and evidence-modulated advantages. Experiments on SVU-31K and ClinVideo-Eval show state-of-the-art results on multi-grained video understanding, fine-grained reasoning, and grounded VQA across diverse benchmarks. We hope this work encourages more reliable verification and evidence attribution for clinical video intelligence.

## Acknowledgement

We gratefully acknowledge the LeapQuest AI team at Shanghai Innovation Institute for their support of this research. We thank our collaborators for their valuable discussions and support throughout the project. We also sincerely thank the anonymous reviewers and the Area Chair for their constructive feedback, which helped us improve the clarity, evaluation, and responsible-release discussion of this work.

## Impact Statement

This work aims to advance evidence-grounded clinical video reasoning for research and educational purposes. ClinVideo-Suite is constructed from existing publicly available research corpora derived from open surgical video platforms, rather than from newly collected hospital recordings. We use these datasets as downstream academic users and document their provenance, while respecting the terms, licenses, copyright constraints, and research-use conditions specified by the original dataset custodians.

To support responsible release, we make available only artifacts that can be legally and ethically shared, including code, training recipes, prompts, evaluation scripts, and permitted derived annotations. Raw surgical videos will not be redistributed; users must obtain them directly from the original platforms or dataset custodians under their own applicable approvals, licenses, and terms of use. MedScope is not intended for standalone clinical use or direct patient-care decision-making. Any future clinical deployment would require prospective validation, appropriate governance review, and oversight by qualified clinicians.

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

## A. Dataset Statistics

**ClinVideo-Cap-635K.** ClinVideo-Cap-635K contains 634.8K timestamped dense captions sourced from MedVideo-Cap (Wang et al., 2025c), OphVL (Hu et al., 2025), and SurgVidLM (Wang et al., 2025a). Figure 7 summarizes the key distributional characteristics across sources. SurgVidLM provides the densest temporal supervision, with an average of approximately 22 captions per video and a long-tailed duration distribution whose 95th percentile extends to several hundred seconds, reflecting the prolonged and continuous nature of surgical workflows. In contrast, MedVideoCap videos are shorter and more uniform, typically containing fewer than 9 captions per video with tightly distributed anchor times. OphVL exhibits moderate caption density but substantially longer captions, with average word counts often exceeding 185 words. These linguistic characteristics are further illustrated in Figure 8.

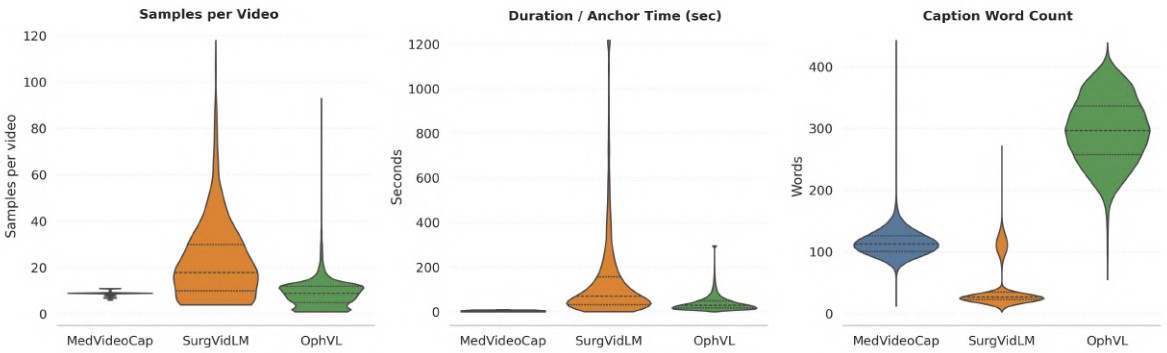

*Figure 7.* Overview of data characteristic distributions across different data sources used in constructing ClinVideo-Cap-635K.

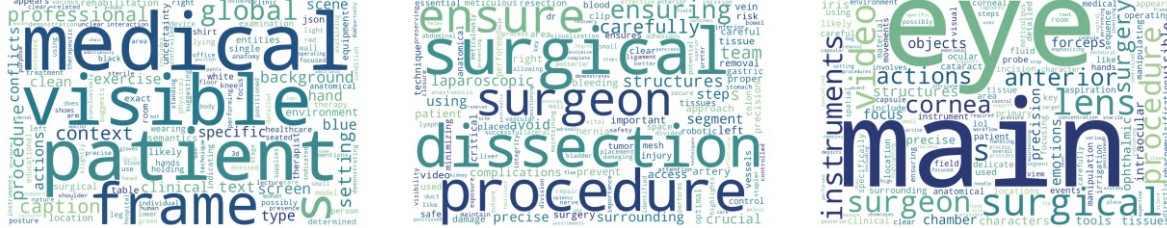

*Figure 8.* Word Clouds of ClinVideo-Cap-635K (from left to right: MedVideoCap, SurgVidLM, and OphVL).

**ClinVideo-QA-254K.** ClinVideo-QA-254K consists of 253.8K open-ended and evidence-grounded question answering pairs with localized supervision windows. As shown in Figure 9, questions remain concise across all sources, with average lengths of roughly 6 to 11 words, while answers are short and precise at about 4 to 5 words. We also observe source-specific differences in evidence granularity: SurgVidLM QA pairs are associated with longer clip durations and higher clip ratios, whereas OphVL questions more frequently rely on shorter and more localized visual evidence. Complementary word cloud visualizations for questions and answers are provided in Figure 10.

**Professional categorization.** To characterize clinical coverage beyond basic statistics, we perform a fine-grained professional categorization of ClinVideo-QA-254K using a two-level hierarchy, where each item is jointly interpreted from its question and answer by `gemini-2.5-flash` (Comanici et al., 2025). Figure 15 reports the hierarchical category distribution for MedVideoCap, while Figures 16 and 17 present the corresponding distributions for SurgVidLM and OphVL, respectively. At the top level, the dataset covers 21 distinct clinical areas across the three sources, and dominant categories closely align with real-world practice. Clinical Practice accounts for approximately 40% of MedVideoCap QA pairs, while procedure-centric topics dominate surgical domains, including Lens and Vitreous Procedures in OphVL at about 40% and Robotic Tissue Manipulation in SurgVidLM at about 30%. At the second level, the majority of QA pairs concentrate on fine-grained procedural reasoning, such as treatment and operative steps accounting for over 60% within Clinical Practice, lens fragmentation and removal at about 59% within OphVL, and robotic tissue dissection at about 70% within SurgVidLM. These results indicate that **ClinVideo-QA-254K** spans broad clinical coverage while placing strong emphasis on action-centric and temporally grounded visual understanding.

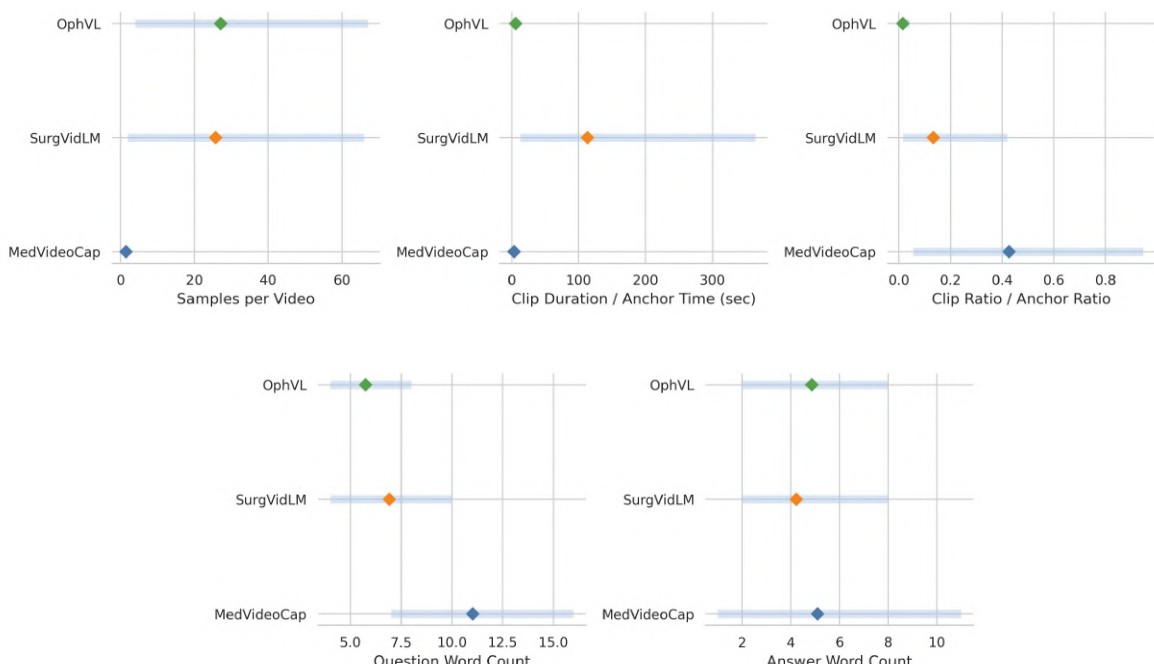

*Figure 9.* Overview of data characteristic distributions across different data sources used in constructing ClinVideo-QA-254K. Diamond markers indicate the mean, and the light-blue bars represent the 5th–95th percentile range.

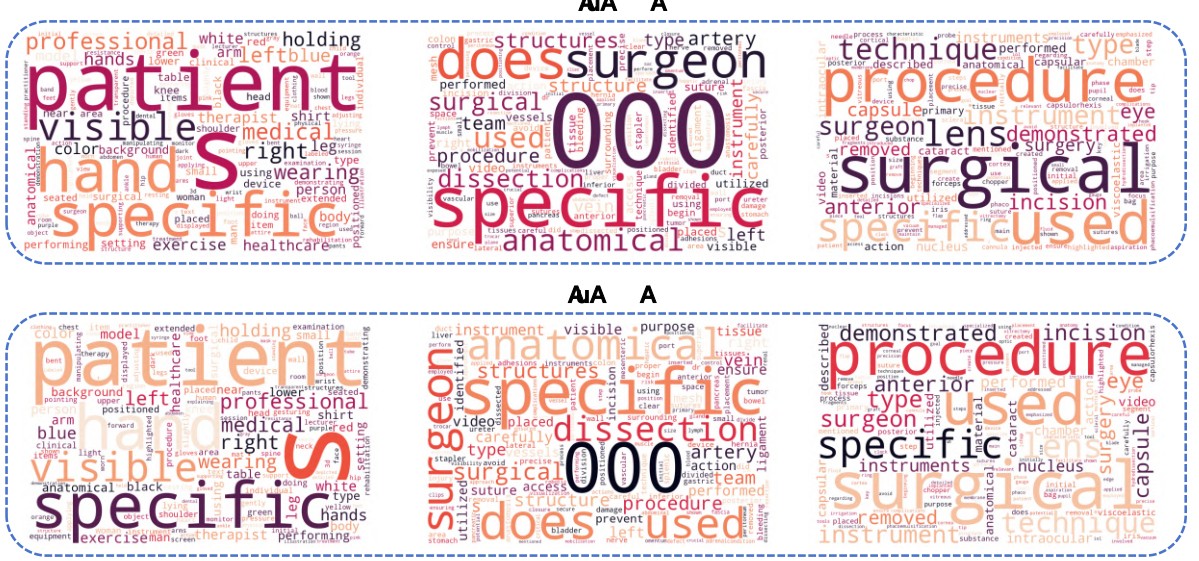

*Figure 10.* Word Clouds of ClinVideo-QA-254K for Questions and Answers (from left to right: MedVideoCap, SurgVidLM, and OphVL).

## B. Details of Algorithm

### B.1. Warm-Up SFT Objective

Let $\mathcal{D}_{\text{vid-cot}}$, $\mathcal{D}_{\text{long-cot}}$, and $\mathcal{D}_{\text{img-vqa}}$ denote the three warm-up corpora (Video-R1 CoT, LongVideo-Reason subset, and surgical image-level VQA, respectively). We form a single merged training set $\mathcal{D}_{\text{warm}} = \mathcal{D}_{\text{vid-cot}} \cup \mathcal{D}_{\text{long-cot}} \cup \mathcal{D}_{\text{img-vqa}}$ by concatenating all examples and shuffling, while preserving each source's supervision format. Each example is represented as a token sequence $y = (y_1, \ldots, y_{|y|})$ conditioned on context $x$ (prompts and multimodal inputs). We train the policy $\pi_\theta$

with the standard teacher-forcing objective, minimizing the token-level negative log-likelihood over the merged corpus:

$$\mathcal{L}_{\text{warm}}(\theta) = \mathbb{E}_{(x,y) \sim \mathcal{D}_{\text{warm}}} \left[ -\sum_{t=1}^{|y|} \log \pi_\theta(y_t \mid x, y_{<t}) \right]. \tag{7}$$

We implement this objective by sampling mini-batches from $\mathcal{D}_{\text{warm}}$ after global shuffling, ensuring that the model is jointly exposed to video CoT, long-video CoT, and image-level VQA signals during warm-up.

## B.2. GA-GRPO Reward Details

**Composite reward.** For each rollout trajectory $\tau$ with terminal prediction $y$ and tool-call trace $T$, we define a dense composite reward

$$R(\tau) = R_{\text{acc}}(y) + R_{\text{format}}(\tau) + R_{\text{evidence}}(T). \tag{8}$$

Here $R_{\text{acc}}$ measures answer correctness, $R_{\text{format}}$ enforces the structured interface, and $R_{\text{evidence}}$ evaluates tool-grounding quality based on the last tool call.

**Answer reward $R_{\text{acc}}$.** We use an LLM-as-a-judge to score the final answer with a three-level grading scheme to reduce reward sparsity:

$$R_{\text{acc}}(y) = \begin{cases} 1, & \text{correct,} \\ 0.5, & \text{partially correct,} \\ 0, & \text{incorrect or overlong.} \end{cases} \tag{9}$$

**Format reward $R_{\text{format}}$.** We apply a deterministic validator for the required trajectory interface. The format reward is binary:

$$R_{\text{format}}(\tau) = \mathbf{1}\big[\text{FORMATOK}(\tau)\big], \tag{10}$$

where FORMATOK checks the presence and ordering constraints of `<think>`, `<tool_call>` (optional), `<tool_response>` (if a tool is called), and `<answer>`.

**Tool-grounding reward $R_{\text{evidence}}$.** We compute $R_{\text{evidence}}$ from the last tool call in $T$, aligning the reward with the native tool interface. If no tool is invoked, we set $R_{\text{evidence}} = 0$. If the last tool call is `crop_video`, we score interval alignment; if it is `get_frame`, we score timestamp alignment. When both tools appear in a rollout, we still use the last call to define the primary grounding signal (consistent with the execution trace used during rollout).

**`crop_video` reward.** Let the last `crop_video` call predict an interval $[p_s, p_e]$. When interval supervision $[g_s, g_e]$ is available, we compute the temporal intersection-over-union $\text{IoU}([p_s, p_e], [g_s, g_e]) = \frac{|[p_s,p_e] \cap [g_s,g_e]|}{|[p_s,p_e] \cup [g_s,g_e]|}$ and map it to a monotonic bonus schedule $R_{\text{crop}} = \begin{cases} \alpha \cdot \text{sign}(\text{IoU} - h_0) + \eta \left\lfloor \frac{\text{IoU}-h_0}{\Delta} \right\rfloor, & \text{IoU} > 0, \\ 0, & \text{IoU} = 0, \end{cases}$ where $h_0$ is a base overlap threshold, $\Delta$ is the step size, and $\alpha, \eta$ control the base and incremental bonus magnitudes. When only frame supervision $g_f$ is provided, we instead use a coverage indicator $R_{\text{crop}} = \mathbf{1}[p_s \leq g_f \leq p_e]$.

**`get_frame`: interval or frame supervision.** Let the last `get_frame` call query timestamp $t$. Under interval supervision $[g_s, g_e]$, we use an in-segment indicator:

$$R_{\text{frame}} = \mathbf{1}\big[g_s \leq t \leq g_e\big]. \tag{11}$$

Under frame supervision $g_f$, we use a tolerance-based proximity score:

$$R_{\text{frame}} = \max\left(0, 1 - \frac{|t - g_f|}{w}\right), \tag{12}$$

where $w$ is a tolerance window controlling how quickly the reward decays as the query moves away from $g_f$.

**Final grounding term.** The tool-grounding reward sums the interval- and timestamp-level components:

$$R_{\text{evidence}}(T) = \mathbf{1}[\exists\, \texttt{crop\_video} \in T] \cdot R_{\text{crop}} + \mathbf{1}[\exists\, \texttt{get\_frame} \in T] \cdot R_{\text{frame}}, \tag{13}$$

where $R_{\text{crop}}$ is computed from the last `crop_video` call (if any) and $R_{\text{frame}}$ from the last `get_frame` call (if any). If no tool call is made, both indicators are zero and $R_{\text{evidence}}(T) = 0$. We use this grounding term together with $R_{\text{acc}}$ and $R_{\text{format}}$ to encourage rollouts that are both correct and temporally grounded by tool-based visual rationalization.

**GA-GRPO Objective** We follow the official GRPO objective and remove the KL term, directly optimizing grounding-aligned returns. Following the variance-reduction intuition of DAPO (Yu et al., 2025), we adopt a group-relative, evidence-weighted advantage and omit variance normalization. Let $G(q) = \{\tau_k\}_{k=1}^G$ denote the group of sampled trajectories for a query $q$. The objective is

$$\mathcal{J}(\theta) = \mathbb{E}_{\substack{q \sim P(Q) \\ \{\tau_k\} \sim \pi_{\theta_{\text{old}}}(\cdot|q)}} \left[ \frac{1}{G} \sum_{k=1}^G \frac{\pi_\theta(\tau_k \mid q)}{\pi_{\theta_{\text{old}}}(\tau_k \mid q)} \hat{A}_k \right]. \tag{14}$$

### B.3. GA-GRPO Group-Level Advantage.

Following the variance-reduction intuition of DAPO (Yu et al., 2025), we compute a group-relative advantage to reduce instability from overly easy queries with saturated rewards and overly hard queries with uniformly low rewards. Concretely, we summarize token-level rewards into a trajectory score and mean-center it within the response group $G$ for the same query:

$$S_i = \sum_{t=1}^{T_i} r_{i,t}, \qquad A_i = S_i - \frac{1}{|G|} \sum_{j \in G} S_j. \tag{15}$$

**Why Advantage Modulation is Necessary?** Group-centering stabilizes GA-GRPO by removing query-specific bias, but it does not solve the fundamental credit-assignment issue: a trajectory-level advantage $A_i$ is a single scalar applied to all actions in the response. As a result, all steps are rewarded or penalized equally, regardless of whether a particular visual rationale is faithful or spurious. In tool-augmented reasoning, this is especially problematic: a trajectory can achieve high reward due to a correct final answer while still using incorrect visual evidence, or fail due to language errors despite correctly localized evidence. Without modulation, the policy is incentivized to reproduce any visual rationale that co-occurs with success, even if it is misaligned with the evidence.

**Evidence-Aware Modulation as Targeted Credit Assignment.** We therefore modulate the trajectory advantage by an evidence-quality factor computed from the same grounding signals used in the reward (crop IoU and frame alignment). This transforms the update from a uniform scalar into a selectively scaled signal that reflects whether visual evidence is reliable. In advantageous trajectories, high-fidelity evidence receives amplified credit while low-fidelity evidence is down-weighted; in disadvantageous trajectories, the opposite weighting increases blame for poor evidence while protecting good evidence from being over-penalized. This mechanism preserves the variance-reduction benefits of group-level baselines while directly addressing the core mismatch between trajectory-level rewards and step-level visual correctness, thereby aligning policy updates with grounded visual reasoning.

## C. Additional Evaluation Results

In this section, we provide additional results to further contextualize our main findings. First, we report a toolbox-enabled comparison on SVU-31K under a unified inference interface, where all baselines are permitted to use the same native tools (`crop_video`, `get_frame`) to enable a more controlled assessment of tool access versus tool-conditioned reasoning (Table 6). Second, we present extra ablations on fidelity-weighted advantage modulation, isolating how different modulation designs affect the stability and effectiveness of grounded policy optimization (Figure 6).

### C.1. Toolbox-enabled comparison under a unified inference interface.

To ensure a fairer comparison, we additionally evaluate all baselines under the same toolbox-enabled setting, where every LMM is allowed to invoke the native `crop_video` and `get_frame` tools at test time in Table 6. Under this unified interface, most zero-shot LMMs show only small changes in scores, indicating that simply granting tool access is insufficient: these models typically do not learn when to call tools, which evidence window to retrieve, or how to integrate retrieved observations into a verifiable decision. In contrast, MEDSCOPE consistently achieves the best performance across all task families, obtaining 4.76–4.77 on full-video description and 4.00/4.36/4.15/4.08 on fine-grained description (CI/DO/CU/TU), and outperforming strong reasoning baselines such as VideoRFT and VideoR1 even when they can use the same tools. The

*Table 6.* **Toolbox-enabled comparison across multi-grained video understanding tasks on SVU-31K.** All models are allowed to call the same native tools (`crop_video`, `get_frame`) during inference. CI, DO, CU, and TU denote correctness of information, detail orientation, contextual understanding, and temporal understanding, respectively. Best results are in **bold** and second-best results are underlined.

| Model | Tool Use | Reasoning | Full Video Description | | | | Fine-grained Video Description | | | |
|---|---|---|---|---|---|---|---|---|---|---|
| | | | CI | DO | CU | TU | CI | DO | CU | TU |
| Video-LLaVA-7B | ✓ | ✗ | 1.05 | 1.00 | 1.03 | 1.02 | 1.14 | 1.11 | 1.23 | 1.22 |
| GroundingGPT-7B | ✓ | ✗ | 1.06 | 1.07 | 1.13 | 1.09 | 1.04 | 1.17 | 1.25 | 1.15 |
| VideoLLaMA3-7B | ✓ | ✗ | 1.67 | 1.69 | 2.24 | 1.84 | 1.60 | 1.85 | 2.31 | 1.93 |
| Qwen2.5-VL-7B | ✓ | ✗ | 1.24 | 1.10 | 1.62 | 1.55 | 1.80 | 1.79 | 2.47 | 1.97 |
| InternVL3-8B | ✓ | ✗ | 1.33 | 1.13 | 1.54 | 1.61 | 1.77 | 1.66 | 2.30 | 2.04 |
| SurgVidLM | ✓ | ✗ | 2.38 | 2.13 | 2.70 | 2.21 | 2.26 | 2.60 | 3.08 | 2.43 |
| LongVT-7B-RFT | ✓ | ✓ | 3.95 | 4.08 | 3.92 | 3.86 | 3.78 | 4.09 | 3.77 | 3.77 |
| VideoR1-7B | ✓ | ✓ | 4.24 | 4.35 | 4.23 | 4.25 | 3.79 | 4.03 | 3.80 | 3.76 |
| VideoChat-R1-7B | ✓ | ✓ | 4.03 | 4.10 | 3.93 | 4.00 | 3.31 | 3.39 | 3.30 | 3.38 |
| VideoRFT-7B | ✓ | ✓ | 4.44 | 4.53 | 4.43 | 4.41 | 3.47 | 3.60 | 3.52 | 3.52 |
| MedScope | ✓ | ✓ | **4.76** | **4.77** | **4.77** | **4.75** | **4.00** | **4.36** | **4.15** | **4.08** |

| Model | Tool Use | Reasoning | Fine-grained Temporal Visual Reasoning | | | | Fine-grained Perception Visual Reasoning | | | |
|---|---|---|---|---|---|---|---|---|---|---|
| | | | BLEU-4 | CIDEr | METEOR | ROUGE-L | BLEU-4 | CIDEr | METEOR | ROUGE-L |
| Video-LLaVA-7B | ✓ | ✗ | 5.72 | 2.12 | 11.62 | 19.60 | 10.80 | 5.45 | 13.58 | 25.85 |
| GroundingGPT-7B | ✓ | ✗ | 2.78 | 1.48 | 10.50 | 18.05 | 4.52 | 1.47 | 11.92 | 21.40 |
| VideoLLaMA3-7B | ✓ | ✗ | 7.60 | 1.95 | 11.60 | 23.05 | 9.62 | 3.90 | 12.70 | 24.30 |
| Qwen2.5-VL-7B | ✓ | ✗ | 4.48 | 1.52 | 13.15 | 16.40 | 3.16 | 2.20 | 14.98 | 25.70 |
| InternVL3-8B | ✓ | ✗ | 3.80 | 1.18 | 11.20 | 18.45 | 2.98 | 1.85 | 12.30 | 26.60 |
| SurgVidLM | ✓ | ✗ | 10.05 | 3.90 | 14.05 | 31.70 | 16.75 | 9.70 | 18.35 | 37.40 |
| LongVT-7B-RFT | ✓ | ✓ | 6.30 | 1.72 | 14.95 | 21.10 | 5.45 | 1.98 | 19.55 | 25.45 |
| VideoR1-7B | ✓ | ✓ | 8.20 | 3.18 | 14.30 | 24.20 | 7.50 | 5.05 | 13.88 | 23.50 |
| VideoChat-R1-7B | ✓ | ✓ | 4.25 | 1.60 | 11.85 | 20.00 | 5.95 | 1.90 | 17.70 | 24.40 |
| VideoRFT-7B | ✓ | ✓ | 6.52 | 2.48 | 11.38 | 20.60 | 6.40 | 3.65 | 16.85 | 23.90 |
| MedScope | ✓ | ✓ | **11.30** | **4.56** | **16.89** | **34.83** | **18.90** | **10.93** | **21.62** | **41.54** |

gap is larger on fine-grained visual reasoning, where MEDSCOPE reaches 11.30 BLEU-4 and 4.56 CIDEr for temporal reasoning and 18.90 BLEU-4 and 10.93 CIDEr for perception reasoning, demonstrating more reliable tool-conditioned evidence localization and verification. Overall, the toolbox-enabled results show that effective long-video grounding requires learned, verification-driven tool use rather than tool availability alone.

### C.2. Training Dynamics

Figure 11 visualizes key training signals during GA-GRPO. The total reward shows a steady upward trend with moderate oscillations, indicating stable optimization without late-stage collapse. The accuracy-related reward increases rapidly in the early phase and then gradually plateaus, suggesting that the policy first learns to satisfy answer-level objectives before focusing on finer improvements. Meanwhile, the response length rises at the beginning and then decreases over training, consistent with the model learning more concise reasoning once tool-based evidence becomes reliable. Finally, the tool-call ratio exhibits a mild downward drift with intermittent spikes, implying that the policy becomes more selective in invoking tools while still calling them when evidence is needed. Overall, these dynamics support that GA-GRPO improves both correctness and evidence efficiency, while keeping the rollout behavior stable.

### D. Additional Benchmarks

**SVU-31K** SVU-31K (Wang et al., 2025a) is a large-scale benchmark designed for multi-grained medical video understanding. It consists of two major task families. (1) **Multi-grained Video Description**, which includes Full Video Description and Fine-grained Video Description. These tasks evaluate the model's ability to generate comprehensive and detailed clinical video summaries across four clinical dimensions, covering content correctness, detail richness, contextual understanding, and temporal coherence. (2) **Visual Reasoning**, which comprises Fine-grained Temporal Visual Reasoning and Fine-grained Perception Visual Reasoning. These tasks assess the model's capability to reason over temporal event sequences and fine-grained visual cues within medical procedures.

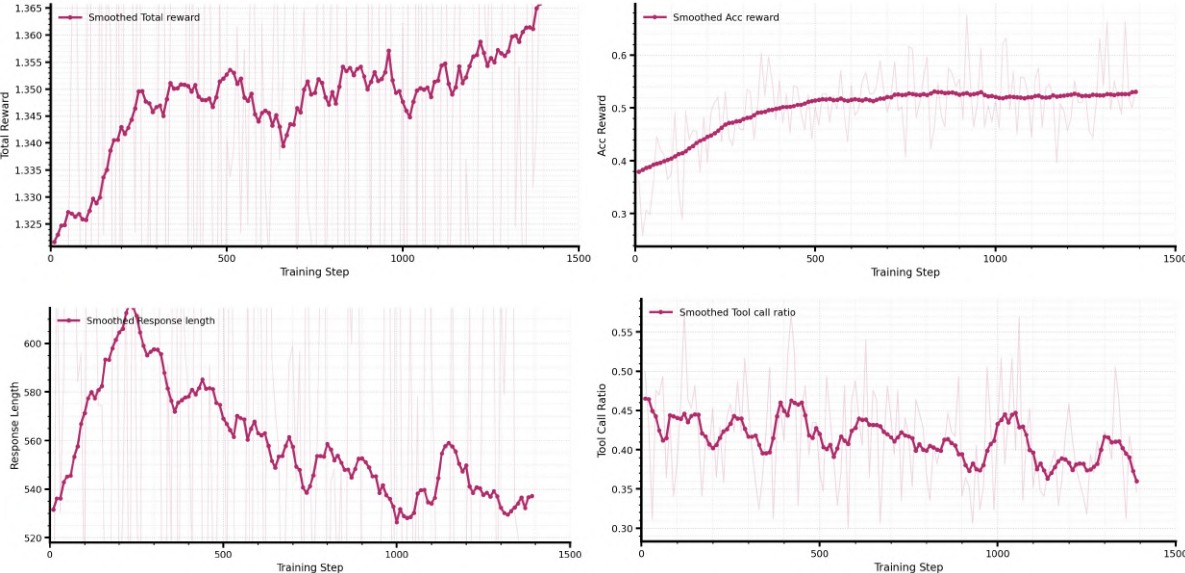

*Figure 11.* Training dynamics of GA-GRPO. We plot the smoothed total reward, accuracy reward, response length, and tool-call ratio over training steps. Light curves show per-step values, while dark curves show smoothed trends, highlighting stable reward growth, early accuracy saturation, and a gradual shift toward more concise and selective tool use.

**ClinVideo-Eval** ClinVideo-Eval is a grounded medical video benchmark constructed to evaluate both in-domain and out-of-domain generalization. It includes three datasets: OphVL, MedVideoCap, and SurgVidLM. OphVL and MedVideoCap are treated as in-domain benchmarks, while SurgVidLM serves as an out-of-domain test set. The former two support both Temporal Grounding and Grounded VQA tasks, whereas SurgVidLM focuses exclusively on Grounded VQA. Temporal Grounding evaluates the model's ability to localize clinically relevant temporal segments, while Grounded VQA assesses joint visual grounding and answer correctness. This benchmark enables a comprehensive analysis of model robustness and generalization across diverse medical video domains.

## E. Additional Implementation Details

**Model settings for data synthesis.** We specify the models and selection hyperparameters used in our three-stage data synthesis pipeline. In Stage 1, the segmenter $\Phi$ is instantiated with `gemini-2.5-pro` to predict entity-guided windows, clip-level dense captions are produced by `Qwen3-VL-235B-A22B-Instruct`, and caption merging plus global summarization are performed by `gpt-4.1`. For clip-level segments, we additionally adopt a lightweight captioning branch that sparsely samples frames within each window and queries `Qwen2.5-VL-72B` to obtain auxiliary captions. In Stage 2, the cross-model filtering pool $\mathcal{M}$ consists of `gemini-2.5-flash`, `grok-4-1-fast`, and `Qwen3-235B-A22B-Instruct`; we set $\theta_{\text{text}} = 1$ and $\theta_{\text{sum}} = 1$ to remove candidates answerable without localized evidence, and $\theta_{\text{loc}} = 2$ to retain candidates with majority agreement under the local caption view $\mathcal{C}(V; w)$. After text filtering, we perform multimodal confirmation with `gemini-2.5-flash` by answering from the corresponding video segment $\text{Clip}(V, w)$ and keeping only pairs that match the target answer. In Stage 3, we use `gemini-2.5-pro` as the teacher to construct environment-interactive VCoT trajectories by iteratively invoking native tools for temporally targeted observations before generating the final prediction.

**SFT.** We implement SFT with lmms-engine using an FSDP2 trainer (Table 7). We train in BF16 and enable FlashAttention-2 for efficient long-context attention. Input sequences are packed to a maximum length of 51,200 tokens with a first-fit strategy, overlong samples are filtered, and dynamic batch sizing is enabled to maximize throughput under variable video and text lengths. To reduce memory usage, we enable gradient checkpointing and remove padding, and we further enable the Liger kernel. Videos are loaded through the qwen_vl_utils backend with an FPS-based sampling strategy, capped at 512 frames, and constrained by a maximum pixel budget of 50,176. Optimization uses AdamW with a learning rate of $5 \times 10^{-5}$, weight decay 0.0, a cosine scheduler, and 120 warmup steps. We use a per-device batch size of 2 with gradient accumulation 1, train for 1,200 steps, and run on 32 H200 GPUs. Checkpoints are saved every 200 steps with a maximum of 5 retained. This SFT stage takes approximately 22.5 hours on 32 GPUs.

| Component | SFT | RL |
|---|---|---|
| Optimizer | AdamW | AdamW |
| Learning Rate (LR) | 5e-5 | 1e-6 |
| LR Scheduler | cosine | constant |
| Weight Decay | 0.0 | 1e-2 |
| No. of Training Steps | 1200 | 1327 |
| No. of Warmup Steps | 120 | 0 |
| Max Length | 51200 | 52384 |
| Dynamic Batch Size | True | False |
| Remove Padding | True | True |
| Liger Kernel | True | False |
| Per-device train batch size | 2 | - |
| Gradient accumulation steps | 1 | - |
| Rollout engine | - | SGLang |
| Rollouts per prompt ($n$) | - | 16 |
| No. of GPUs | 32 | 64 |
| No. of Frames | 512 | 512 |

*Table 7.* Key hyperparameters for supervised fine-tuning and reinforcement learning.

**RL.** We implement RL with verl and generate rollouts using SGLang (Table 7). We disable entropy regularization by setting the actor entropy coefficient to 0 and optimize with AdamW at a learning rate of $10^{-6}$ under a constant learning-rate schedule with weight decay $10^{-2}$. Training runs for 1,327 steps with no warmup and a maximum sequence length of 52,384 tokens. We enable gradient checkpointing for the actor model and use BF16 mixed precision throughout, explicitly setting reduce and buffer dtypes to BF16. Both the actor and reference models are trained under FSDP with parameter and optimizer offloading disabled to avoid host-device transfer overhead. For rollouts, we set the rollout engine to sglang with tensor model parallel size 1, log-prob micro-batch size per GPU 1 for both actor and reference, and we cap rollout GPU memory utilization at 0.4 to maintain stable serving during sampling. We sample $n = 16$ rollouts per prompt and use Ulysses sequence parallelism with size 4 for the actor. We further remove KL from the reward matching our GA-GRPO training setup. RL is trained on 64 H200 GPUs.

## F. Additional Baselines & Metrics

### F.1. Baseline Models

We compare MedScope with several baseline methods, which are divided into proprietary models, open-source models, reasoning models, and reasoning agents.

**Proprietary Models**   This category includes state-of-the-art closed-source models such as GPT-4o (Hurst et al., 2024), Gemini 3-Pro-Preview (Comanici et al., 2025), and Gemini-2.5-Flash (Comanici et al., 2025).

**Open-source Models**   The open-source baselines include video-language models such as Qwen2.5-VL-7B (Bai et al., 2025), InternVL3-8B (Zhu et al., 2025), VideoLLaMA3-7B (Zhang et al., 2025a), Video-LLaVA-7B (Lin et al., 2024), GroundingGPT-7B (Lin et al., 2024), and SurgVidLM (Wang et al., 2025a).

**Reasoning Models**   Reasoning-oriented baselines include VideoR1-7B (Feng et al., 2025), VideoChat-R1-7B (Li et al., 2025), and VideoRFT-7B (Wang et al., 2025b).

**Reasoning Agents**   Reasoning agents incorporate structured tool use and multi-step inference, including MedScope-7B-SFT, MedScope-7B-RL, ReWatch-R1-7B (Zhang et al., 2025b), and LongVT-7B-RFT (Yang et al., 2025).

### F.2. Evaluation Metrics

For video description tasks, we follow the evaluation pipeline in SurgVidLM (Wang et al., 2025a), employing a GPT-based evaluator to score model outputs on a 1–5 scale across four clinical aspects: correctness of information (CI), detail orientation (DO), contextual understanding (CU), and temporal understanding (TU). For SVU-31K, we additionally report standard

captioning and reasoning metrics, including BLEU-4 (Papineni et al., 2002), CIDEr (Vedantam et al., 2015), METEOR (Banerjee & Lavie, 2005), and ROUGE-L (Lin, 2004). For grounded medical video understanding, Temporal Grounding is evaluated using Recall at multiple IoU thresholds and mIoU, while Grounded VQA is evaluated using mIoU and accuracy.

## G. Case Analysis

This section provides brief case studies to visualize test-time tool use. We highlight recurring behaviors learned after VCoT cold-start and GA-GRPO, showing how the model localizes evidence with `crop_video`, probes details with `get_frame`, and composes them for verification. We also include representative failure cases, revealing residual phenomena where negative evidence does not always trigger re-localization and where textual priors can occasionally dominate fine-grained visual binding.

### G.1. Good Cases Analysis

**Hypothesis-Driven Temporal Localization.**    Figure 18 illustrates an emergent test-time behavior after VCoT cold-start and GA-GRPO: the model separates hypothesis formation from evidence acquisition, and uses temporally targeted tool calls as a verification step rather than an optional add-on. Instead of answering immediately from coarse context, it first proposes a plausible surgical rationale with explicit temporal anchoring, then commits to a `crop_video` call that concentrates computation on the suspected decision window. The second turn shows a stable evidence integration loop where the retrieved clip is treated as the decisive signal to confirm or revise the hypothesis before producing the final answer. This suggests the model has learned a lightweight "plan then act then verify" policy with calibrated stopping, which is difficult to elicit from tool access alone and aligns with the intended training objective of coupling reasoning with temporally grounded verification.

**Framewise Detail Probing.**    Figure 19 highlights a characteristic behavior learned after tool-augmented training: the model treats `get_frame` as a targeted visual probe for fine-grained attributes, using minimal queries to resolve a localized detail question instead of over-retrieving. A notable trait is its failure-aware control policy: when a probe violates the clip boundary and the tool returns an execution error, the model does not collapse into speculation, but reuses already validated temporal evidence and preserves the queried action context to complete the answer. This indicates improved robustness to tool noise and boundary conditions, as well as an internalized notion of evidence sufficiency for short-clip decision making.

**Fine-to-Coarse Evidence Search.**    Figure 20 illustrates an emergent control behavior after tool-use training: the model treats temporal retrieval as an iterative search problem and adjusts the evidence window based on observed insufficiency. Instead of committing to the first hypothesis-driven crop, it diagnoses that the initial narrow interval lacks decisive evidence, then deliberately broadens the crop to capture the complete procedural sequence and any on-screen cues. This reflects a learned notion of evidence completeness and a self-correcting retrieval policy, improving reliability when the first tool call under-covers the critical moment.

**Parallel Time-Jump Self-Correction.**    Figure 21 reflects an emergent failure-aware retrieval strategy learned through tool-use training. The first crop retrieves a segment that is visually well formed but semantically misaligned with the question because it remains in the speaker presentation rather than the operative field. The model explicitly recognizes this mismatch and treats it as an evidence acquisition error instead of forcing an answer from irrelevant frames. It then revises its temporal hypothesis and performs a parallel jump to a distant time window that is more likely to contain the suturing step. After retrieving the corrected interval, the model grounds the answer in a concrete spatial cue by locating the needle entry at the junction where the two wound edges meet. This behavior indicates that the model has learned to decouple retrieval from decision making, using tool feedback to correct localization and improve reliability when early evidence windows are wrong.

**Coarse-to-Fine Tool Chaining.**    Figure 22 demonstrates an emergent coarse-to-fine verification behavior where the model composes multiple tools into a single evidence-seeking routine rather than treating each call independently. The model begins with a deliberately wide crop to maximize recall because the target event is temporally sparse and easy to miss under motion, haze, and intermittent occlusion. After detecting a plausible two-tool overlap, it reduces the temporal extent to suppress distractors and stabilize the visual context, effectively trading coverage for precision once a candidate window has been identified. It then switches tools from temporal localization to frame-level inspection, selecting a representative timestamp inside the overlap to make the count checkable from a single image and therefore less sensitive to narrative drift. This chaining behavior suggests the model has learned an internal division of labor across tools, using cropping for search

and get_frame for confirmation, which improves robustness for brief concurrency queries that would otherwise be unreliable from global summarization alone.

## G.2. Failure Cases Analysis

**Evidence-Overrule Failure.** Figure 23 shows that the model can invoke a tool for verification and explicitly report that the retrieved frames do not contain the queried action. The trajectory then transitions to an answer without launching a follow-up localization step, suggesting a weak linkage between negative evidence and re-planning. One plausible contributing factor is that the policy has learned a common response template where a single verification attempt is often sufficient, while cases requiring iterative recovery from uninformative crops are less emphasized. This leads to occasional reliance on procedural priors when tool feedback is inconclusive.

**Text-Anchored Entity Binding Drift.** Figure 24 shows that the model can use a tool to retrieve a broad segment that mixes case text with the operative view, and it then commits to an anatomical label that is strongly conditioned on the textual anchor. The trajectory suggests that the case description is treated as a high-confidence prior, while the fine-grained visual binding between the instrument and the grasped tissue is not explicitly stress-tested. This can yield occasional confusion between the true target structure and a nearby landmark when the target is small or visually ambiguous.

**Repetitive Temporal Loop Failure.** Figure 25 shows a failure mode in which the model repeatedly queries nearly the same temporal region despite receiving uninformative visual feedback. The first two crop_video calls are both anchored near the beginning of the video and mainly retrieve introductory title frames, while the decisive evidence appears more than 100 seconds later. Rather than using the negative observation to re-hypothesize an unexplored temporal region, the model only marginally extends the end time of the same early window. This suggests a weak exploration mechanism after non-discriminative crops: the policy locally adjusts a failed query instead of performing a broader temporal reset. As a result, the final answer is inferred from general medical knowledge rather than grounded video evidence, leading to an incorrect prediction.

**Premature Scan Termination Failure.** Figure 26 illustrates a case where the model begins a reasonable sequential scan but stops before reaching the decisive terminal action. The retrieved crops contain suture-related content, including the needle passing through the sclera and the thread being pulled, and the model correctly recognises that these frames do not cover the very end of the video. However, despite explicitly noting that the final knot action has not been observed, it does not issue an additional tool call to inspect the remaining segment. The trajectory therefore reflects a premature commitment failure: partial evidence is treated as sufficient for answering, even when the model's own reasoning indicates that the critical endpoint remains unseen. This leads the model to produce a generic answer about the knot's fate while missing the specific action in the ground-truth evidence window.

# H. Prompts

## H.1. Training Prompts

To operationalize the coarse-to-fine clinical reasoning framework (Section 3.2), MedScope employs a lightweight yet *strictly structured* prompting interface that couples hypothesis-driven textual reasoning with temporally targeted evidence acquisition. As shown in Figure 12 and Figure 13, our prompts are designed to (i) expose a compact visual toolbox through an explicit function contract, (ii) enforce an unambiguous interaction protocol for multi-round trajectories, and (iii) standardize output boundaries to make tool use and final predictions reliably parsable and verifiable.

**System prompt (tool contract and executable interface).** Figure 12 presents the system prompt, which defines the assistant role and specifies the available tools via machine-readable function signatures enclosed in <tools>...</tools>. This contract-first design makes tool invocation an explicit, executable action rather than an implicit textual suggestion, thereby supporting inspectable trajectories of the form $\tau_i = \{(t_{i,k}, a_{i,k}, o_{i,k})\}_{k=1}^{K_i}$ (Section 3.2). Concretely, the action space contains two native functions that implement hierarchical temporal inspection: (1) crop_video, which takes a video path and a coarse time window and returns a clip for localized, denser evidence gathering; and (2) get_frame, which takes a video path and a fine-grained timestamp to extract frames around a specific moment for verification. This coarse-to-fine evidence retrieval pattern aligns with prior tool-augmented video reasoning paradigms that dynamically sample additional frames on demand to reduce hallucination and improve long-video grounding.

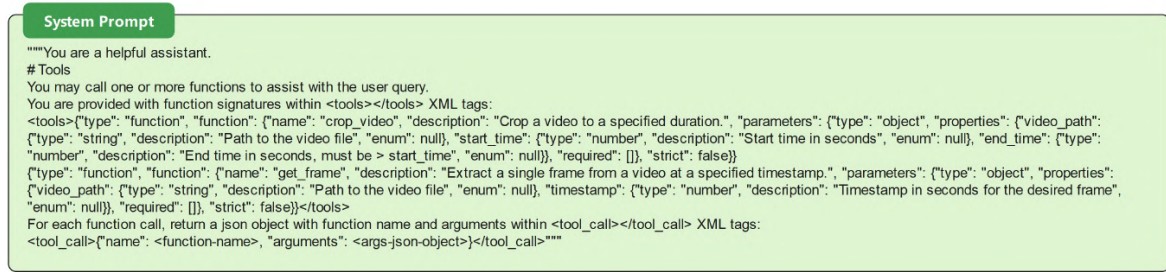

*Figure 12.* System prompt of MedScope.

**User Prompt**

<video>{question} Think first, call **crop_video** or **get_frame** if needed, then answer. Format strictly as: <think>...</think> <tool_call>...</tool_call> (if tools needed) <answer>...</answer>. The Video path for this video is:{video_path}

*Figure 13.* User prompt of MedScope.

**User prompt (goal specification and protocol constraints).** Figure 13 shows the user prompt template, which injects the instance query and the concrete video path while enforcing a strict output protocol: the model must first produce textual rationalization in <think>...</think>, optionally emit one or more tool calls in <tool_call>...</tool_call> (when evidence is needed), and finally output the terminal prediction in <answer>...</answer>. By explicitly separating *reasoning* (<think>) from *evidence acquisition* (<tool_call>) and *final decision* (<answer>), the prompt makes the decision process externally auditable and naturally supports iterative verification, where observations returned by the environment are appended back into context for the next round (Section 3.2). Moreover, the prompt encourages *evidence-seeking* behavior (invoke crop_video/get_frame "if needed"), which is critical for clinical scenarios where temporally localized cues must be checked rather than assumed.

**LLM-as-a-Judge prompt.** As shown in Figure 14, we adopt a lightweight LLM-as-a-judge template to measure the semantic consistency between two answers: the reference and the extracted model output given the same Question. The judge is instructed to output only one scalar from $\{1, 0.5, 0\}$, corresponding to fully consistent, partially consistent, and inconsistent. This discrete, format-constrained scoring avoids verbose rationales and reduces parsing ambiguity, enabling robust aggregation (e.g., voting or consensus) when computing agreement signals in our filtering pipeline.

### H.2. Prompting Details for ClinVideoSuite Data Synthesis

**ClinVideoSuite** is constructed to be evidence-centric and temporally localized: captions, QA pairs, and VCoT trajectories are all tied to explicit supervision windows and are constrained to rely on visually observable cues. To make this scalable and automatically verifiable, we use a set of strict, interface-oriented prompts that standardize (i) temporal segmentation, (ii) dense clip captioning and global summarization, (iii) window-grounded QA construction, and (iv) environment-interactive VCoT trajectory synthesis.

#### H.2.1. STAGE 1: EVIDENCE-CENTRIC CAPTIONING AND GLOBAL SUMMARIZATION.

Stage 1 builds timestamped dense captions as localized evidence and a compact global summary as context. This stage is implemented by three prompts that form a coarse-to-fine summarization stack: semantic boundary detection → clip-level dense captioning → chronological caption merging.

**Original Clip Prompt: semantic boundary detection for windowing.** As shown in Figure 27, given sparse, chronological frames from a video interval (each frame annotated with an absolute timestamp in seconds), the prompt asks the model to propose semantic change points that split the interval into coherent phases or steps. Crucially, it enforces a JSON-only output with a deterministic schema: {"cut_points":[...], "segment_summaries":[...]}, where the summaries are *single concise phrases* and the segments must be continuous and cover the full interval. The prompt further imposes structural constraints that stabilize window generation at scale: cut points must be strictly increasing and lie inside the open interval (excluding boundaries), a minimum number of segments is required for sufficiently long clips (unless truly no semantic change exists), and very short segments are discouraged via an explicit minimum-length constraint. These rules make $\mathcal{B}(V)$ (Eq. (2)) *well-formed* and prevent degenerate partitioning (e.g., missing coverage or noisy micro-segments),

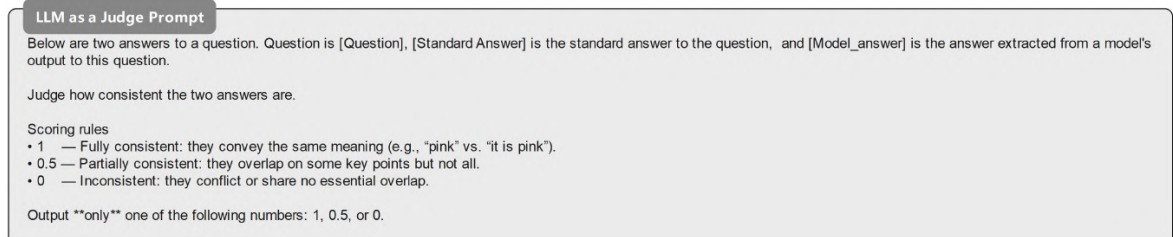

*Figure 14.* Prompt for LLM-as-a-Judge.

yielding reliable supervision windows for downstream dense captioning and QA grounding.

**Clip Caption Prompt: dense, clinically grounded, *visual-only* descriptions.**    For each predicted window $w = [s, e]$, the model generates a dense caption that describes only what is visually supported within that temporal span. The prompt in Figure 28 explicitly prioritizes medical and surgical semantics—anatomy or target tissue, instruments, actions, and workflow steps—and encourages spatial relations when visible (e.g., left/right, proximal/distal, superficial/deep, anterior/posterior). At the same time, it includes a strong anti-hallucination constraint that forbids inventing patient information, diagnoses, or outcomes not directly observable. This balances two needs: (i) high-fidelity evidence that captures fine-grained visual cues for grounding, and (ii) conservative phrasing that reduces caption-induced shortcuts in later QA generation and verification.

**Caption Merge Prompt: compressing chronological caption blocks into a global summary.**    To obtain a compact global context $S(V)$ without leaking localized cues, we apply the prompt in Figure 29 to merge two chronological caption blocks into one shorter block while preserving timestamp ranges at the beginning of each output line. The prompt enforces plain-text lines only (no headings, bullets, or markdown), strict chronological order, and aggressive compression that keeps only the surgical/clinical storyline (major steps, key anatomy, and essential instruments). It explicitly drops generic filler (e.g., emotions, professionalism, operating-room environment narration, long instrument inventories), merges adjacent lines describing the same step, caps the output length (e.g., $\leq 6$ lines), and forbids introducing new facts. As a result, $S(V)$ remains a coarse overview that supports high-level reasoning, while preserving the necessity of retrieving temporally localized evidence from $\mathcal{C}(V)$ for fine-grained grounding.

### H.2.2. STAGE 2: HIGH-QUALITY QA CONSTRUCTION WITH LOCALIZED EVIDENCE DEPENDENCE.

Stage 2 constructs QA pairs that hinge on temporally localized visual cues and are not answerable from coarse context alone. As shown in Figure 30, we operationalize this objective with a single, strongly constrained prompt that explicitly separates global context from local evidence.

**QA Pair Generation Prompt: clip-answerable but global-not-answerable QA.**    The QA pair generation prompt takes as input: (A) a *Global Summary* (coarse overview of the full video), (B) a *Clip Caption* (local window description with fine-grained details), (C) the *Clip Timecodes* $[s, e]$ in seconds, and (D) an upper bound $K$ on the number of QA pairs. It then generates between 1 and $K$ QA items under eight hard rules: **(i)** one single-part English question (length-limited; avoid double-barreled forms), **(ii)** *visual-only* evidence (no domain knowledge, guessing, or narration-only facts), **(iii)** answerable using the clip caption but not confidently answerable using only the global summary (prefer details present in the clip caption but absent/vague globally), **(iv)** concrete and fine-grained (objects/instruments, colors, actions, temporal order, spatial relations; avoid subjective/evaluative questions), **(v)** short, definite answers (length-limited; disallow hedging), **(vi)** no answer leakage in the question, **(vii)** no explicit mentions of "clip/segment" or timestamps in the question, while forcing each output item to carry `start_time` and `end_time` exactly equal to the provided timecodes, and **(viii)** diversity across multiple QA items (no paraphrases; cover different visual details). The output is JSON-only as a list of objects, making it easy to validate automatically and to align each QA with a supervision window for later multimodal confirmation and trajectory synthesis.

### H.2.3. STAGE 3: ENVIRONMENT-INTERACTIVE VISUAL-COT SYNTHESIS.

Stage 3 synthesizes visual-CoT trajectories by having a teacher model alternate between hypothesis-driven reasoning and tool-based evidence retrieval in a real video environment. We implement this with a two-phase VCoT prompting protocol in Figure 31 that explicitly encodes coarse-to-fine verification and yields inspectable tuples $(t_{i,k}, a_{i,k}, o_{i,k})$.

**VCoT Generation Prompts: Phase 1 global skim and planning (Round 1).** The **Round 1 VCoT prompt** casts the model as a long-video reasoning assistant and exposes a native tool interface (e.g., `crop_video`) via `<tools>...</tools>` signatures. To make tool use executable and parsable, it requires each tool invocation to be wrapped in `<tool_call>...</tool_call>` and prohibits emitting `<tool_response>` directly (the environment injects observations after execution). The prompt further enforces a strict decision structure: in each round, the model must first write a non-empty, evidence-integrating `Thinking` section (3–6 sentences, with natural time anchors), then output either exactly one tool call and stop, or a final answer and stop (never mixing both). This "plan-then-act" constraint encourages the teacher to articulate what evidence is missing before querying the environment, aligning with our coarse-to-fine verification objective.

**VCoT Generation Prompts: Phase 2 fine-grained inspection (Round 2 to Round n).** For subsequent rounds, the **fine inspection template** instructs the model to continue the trajectory without repeating earlier content, produce a short reflective `Thinking` section grounded in the newly retrieved frames (typically low-resolution segment frames), and again choose exactly one action: either request another tool call (one per round) or output the terminal answer. The prompt explicitly reminds the model not to expose privileged metadata (e.g., time-window hints or reference answers) in its reasoning text, which prevents leakage while still supporting consistent supervision during synthesis. Overall, the two-phase prompting design induces a natural hierarchy: global skim to localize candidate evidence regions, followed by iterative fine inspection to confirm or revise hypotheses with temporally targeted observations, producing high-fidelity environment-interactive VCoT trajectories suitable for cold-start and RL stages.

# I. Limitations

While MedScope demonstrates strong evidence-grounded reasoning on long-form clinical videos, several limitations remain. First, our study focuses on 2D endoscopic and surgical video streams with time-localized evidence and does not yet extend to volumetric or spatially registered modalities such as 3D endoscopy, CT, MRI, ultrasound, or multi-view operating-room recordings. As a result, MedScope may not fully capture clinical workflows that require reasoning over three-dimensional anatomy, cross-modal alignment, or longitudinal fusion of imaging with perioperative records. In addition, ClinVideoSuite is built from existing publicly available research corpora rather than prospectively collected clinical data, and may therefore inherit biases from the source videos, including procedure selection, recording quality, narration style, institution or surgeon visibility, and platform-specific availability. Although our evaluation suite includes physician verification for sampled cases and evidence traceability, part of the annotation pipeline relies on automatically generated captions, questions, answers, and reasoning traces. Thus, residual annotation noise, incomplete evidence windows, or source-dataset biases may remain, and broader validation across more institutions, procedures, and annotation protocols would further strengthen conclusions about generalizability.

Second, the current tool interface is intentionally lightweight, centered on temporal densification and frame-level verification. This design improves reliability and efficiency, but it does not exhaust the range of tools that could benefit clinical decision-making, such as instrument and anatomy segmentation, action phase recognition, OCR for on-screen metadata, structured report generation, or integration with external knowledge bases and patient context. Extending the toolbox to richer modalities and more diverse perception primitives, as well as scaling training and rollouts under larger compute budgets, may further improve robustness, reduce failure cases under ambiguous visual evidence, and enable more comprehensive clinical reasoning beyond video-only settings. Finally, evidence localization should not be interpreted as a guarantee of correctness, robustness, or clinical safety. MedScope is intended for research and educational use only, not for standalone clinical deployment or direct patient-care decision-making. The release of code, recipes, prompts, evaluation scripts, and derived annotations will follow the licenses, terms of use, and redistribution permissions of the corresponding source datasets; raw surgical videos will not be redistributed.

# J. Future Work

A natural next step is to broaden MedScope from video-only reasoning to holistic multimodal clinical intelligence by incorporating volumetric and multi-view signals such as 3D endoscopy, CT, MRI, and ultrasound, and by aligning video evidence with structured patient context across the perioperative timeline. On the modeling side, extending the action space beyond temporal cropping and frame querying to include richer perception and analysis tools, such as segmentation, tracking, phase recognition, and structured summarization, may enable more precise evidence attribution and stronger robustness under challenging or rare visual patterns. Finally, scaling physician-verified evaluation to a wider range of procedures and

institutions, and exploring more principled reward shaping and uncertainty-aware stopping criteria for agentic rollouts, are promising directions to further improve reliability, safety, and generalization for real-world deployment.

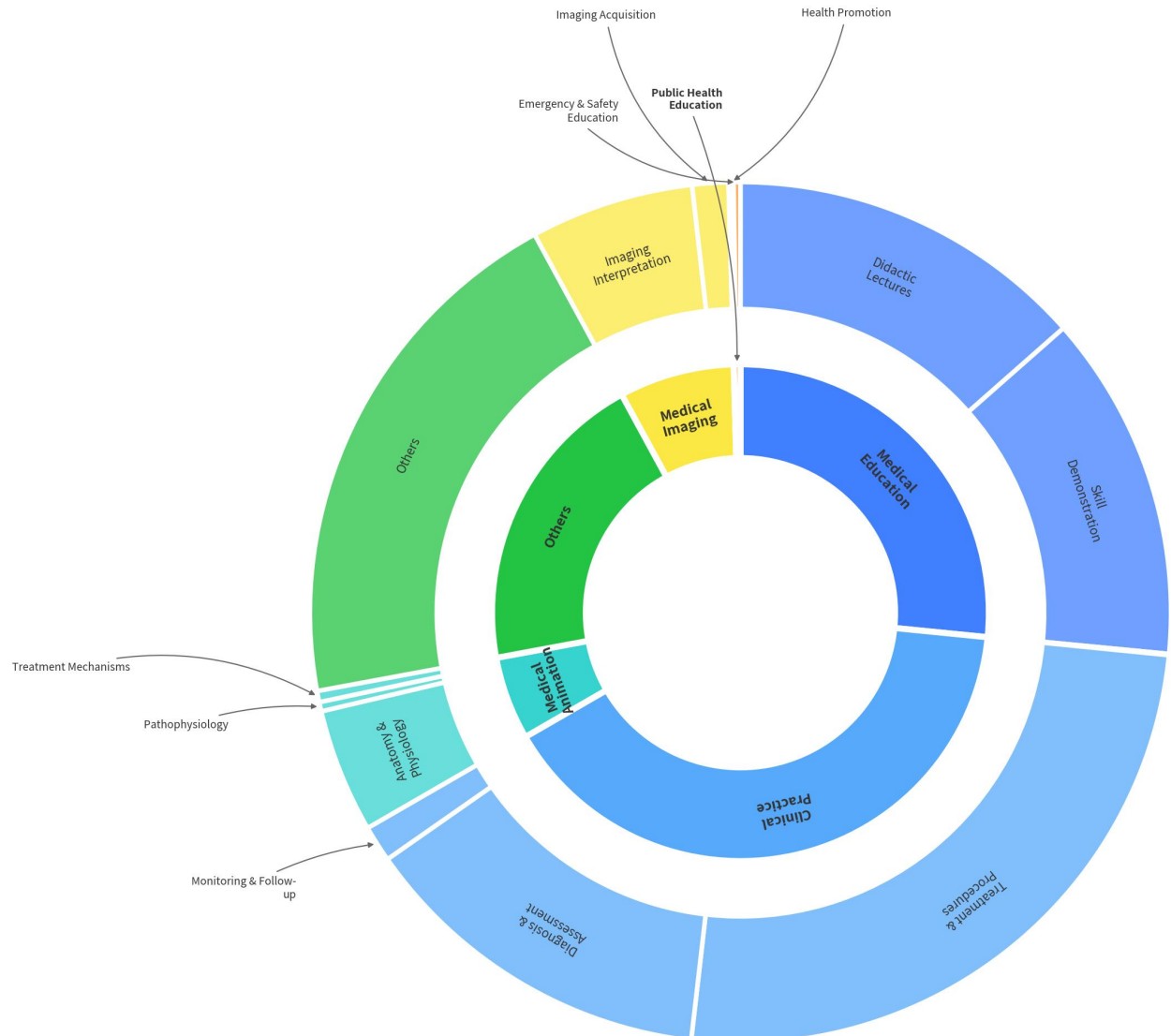

*Figure 15.* Hierarchical Category Distribution of ClinVideo-QA-254K (MedVideoCap): Inner Ring for Top-Level and Outer Ring for Second-Level Categories.

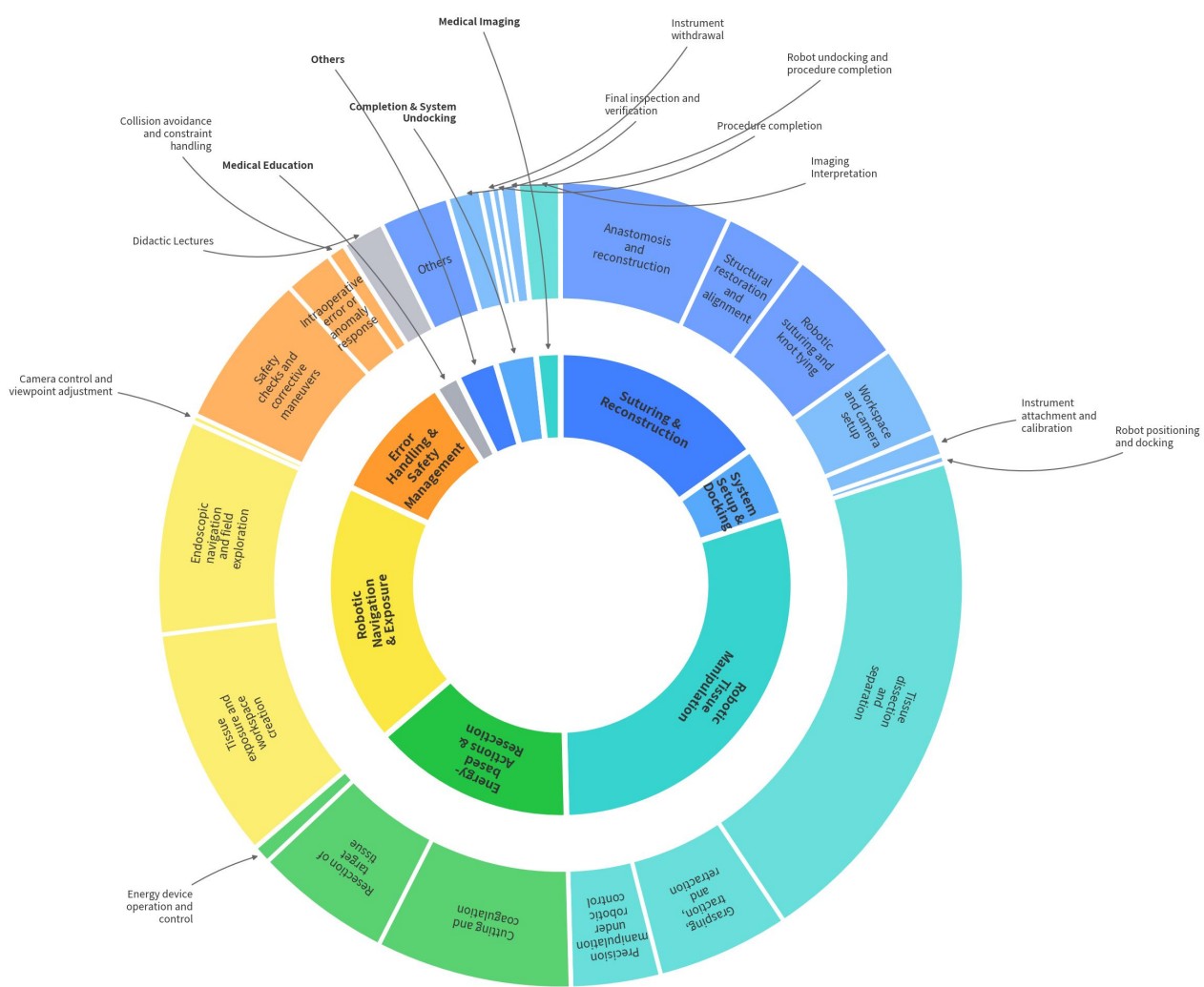

*Figure 16.* Hierarchical Category Distribution of ClinVideo-QA-254K (SurgVidLM): Inner Ring for Top-Level and Outer Ring for Second-Level Categories.

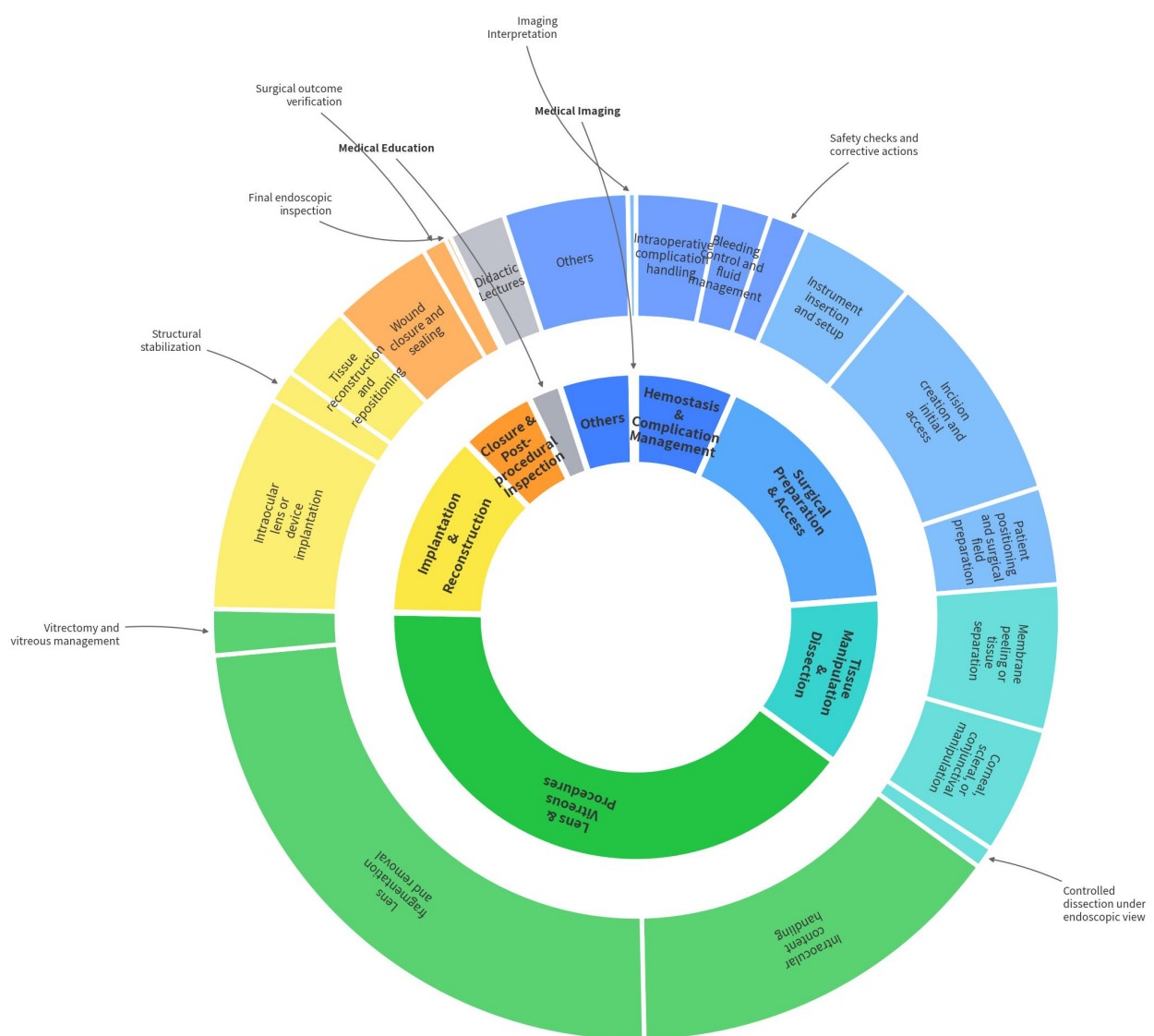

*Figure 17.* Hierarchical Category Distribution of ClinVideo-QA-254K (OphVL): Inner Ring for Top-Level and Outer Ring for Second-Level Categories.

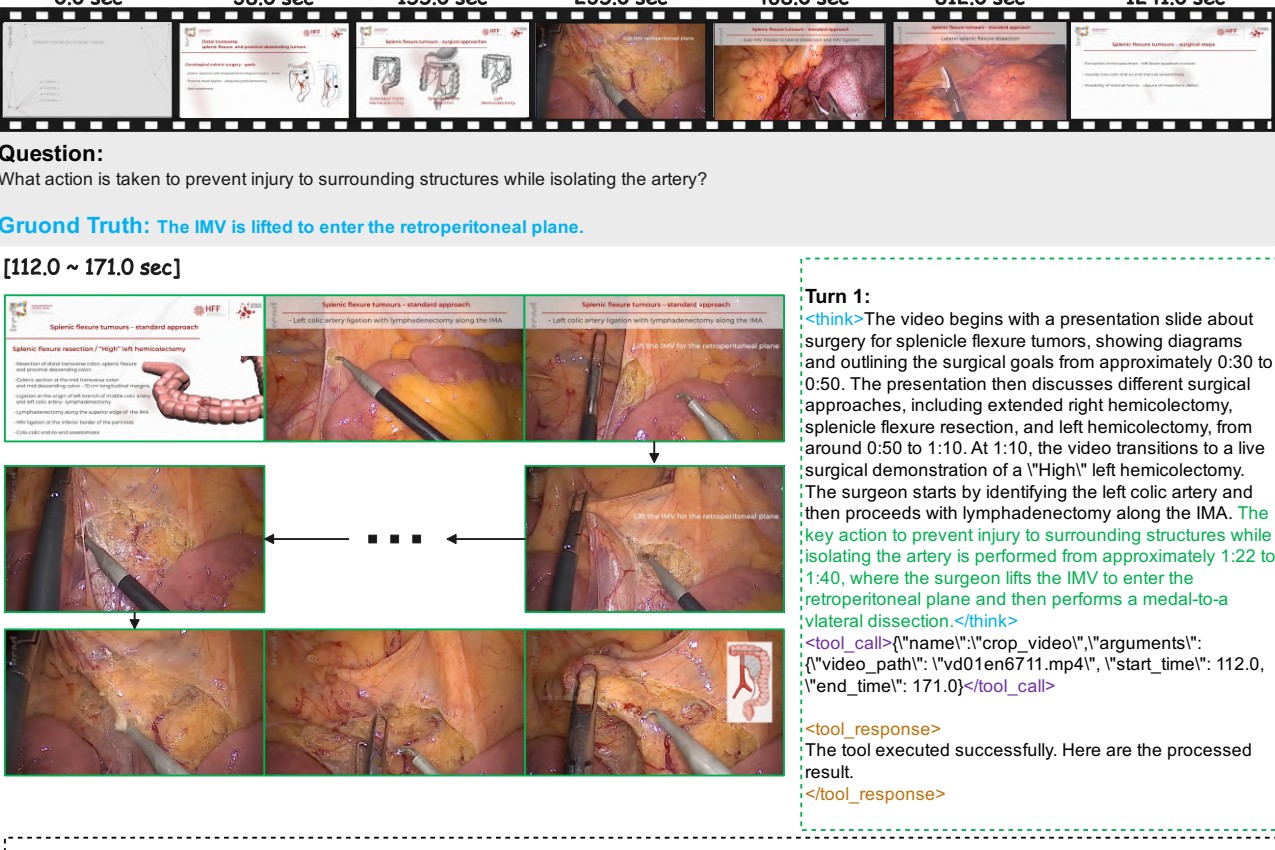

**Question:**
What action is taken to prevent injury to surrounding structures while isolating the artery?

**Ground Truth:** The IMV is lifted to enter the retroperitoneal plane.

**[112.0 ~ 171.0 sec]**

**Turn 1:**
<think>The video begins with a presentation slide about surgery for splenicle flexure tumors, showing diagrams and outlining the surgical goals from approximately 0:30 to 0:50. The presentation then discusses different surgical approaches, including extended right hemicolectomy, splenicle flexure resection, and left hemicolectomy, from around 0:50 to 1:10. At 1:10, the video transitions to a live surgical demonstration of a \"High\" left hemicolectomy. The surgeon starts by identifying the left colic artery and then proceeds with lymphadenectomy along the IMA. The key action to prevent injury to surrounding structures while isolating the artery is performed from approximately 1:22 to 1:40, where the surgeon lifts the IMV to enter the retroperitoneal plane and then performs a medal-to-a vlateral dissection.</think>
<tool_call>{\"name\":\"crop_video\",\"arguments\": {\"video_path\": \"vd01en6711.mp4\", \"start_time\": 112.0, \"end_time\": 171.0}</tool_call>

<tool_response>
The tool executed successfully. Here are the processed result.
</tool_response>

**Turn 2:**
<think>This video segment, from 1:12 to 1:51, clearly illustrates the surgical steps to isolate the artery while preventing injury to surrounding structures. It begins with the identification of the left colic artery and the start of lymphadenectomy along the IMA. Around 1:22, the procedure explicitly shows the surgeon lifting the inferior mesenteric vein (IMV) to gain access to the retroperitoneal plane. This action of lifting the IMV is crucial for creating a safe dissection plane and preventing damage to the vein and other nearby vessels. Subsequently, from approximately 1:30, the surgeon performs a medial-to-lateral dissection in this protected plane, further ensuring careful isolation of the artery. This confirming and elaborating on my initial hypothesis that the action of lifting the IMV for the retroperitoneal plane is the key step. The current window is well-localized and provides sufficient evidence to answer the question.</think>

<answer>The IMV is lifted to enter the retroperitoneal plane.</answer> ✅

*Figure 18.* **Hypothesis-Driven Temporal Localization Example.** Given an instrument-safety question during arterial isolation, the model first performs a global skim to hypothesize the critical maneuver, then calls `crop_video` to zoom into the suspected window (around 1:22–1:40). The retrieved segment verifies that the surgeon lifts the IMV to enter the retroperitoneal plane, enabling safe medial-to-lateral dissection and reducing the risk of injury to surrounding structures. Green highlights indicate key evidence-seeking and verification statements in the VCoT trajectory.

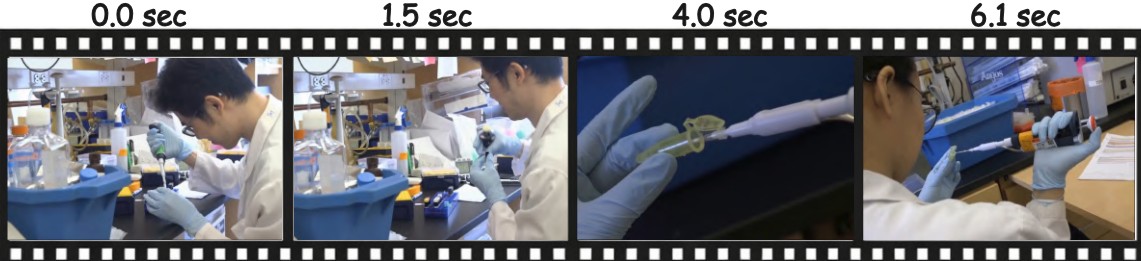

0.0 sec          1.5 sec          4.0 sec          6.1 sec

**Question:**
What color is the storage box visible in the background when the researcher is holding a pipette and a small, green, transparent component?

**Gruond Truth:** The storage box is blue.

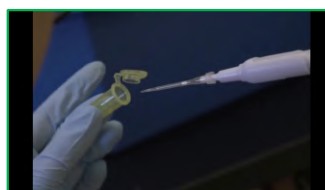

5.0 sec

**Turn 1:**
<think>The video begins with a researcher in a lab setting, looking into a microscope and holding a pipette. In the initial frames, specifically around 0-2 seconds, a small, green, transparent component is visible on the desk, and the researcher is holding a pipette. However, I do not observe a storage box in the background during these initial moments. The background primarily shows a computer monitor and general lab equipment. To answer the question, I need to find a segment where a storage box is clearly visible in the background while the researcher is holding a pipette and the green component is present. I will check a later part of the video, around 5 seconds, to see if the scene changes to reveal such an object.</think>

<tool_call>{\"name\":\"get_frame\",\"arguments\":{\"video_path\": \"../59309.mp4\", \"timestamp\": 5.0}}</tool_call>

<tool_response>
The tool executed successfully. Here are the processed result.
</tool_response>

**Turn 2:**
<think>The frames around 5 seconds show the researcher holding a small, green, transparent component and a pipette, similar to the beginning of the video. However, there is still no storage box visible in the background. The background is largely dark and out of focus, with only a blue surface and some yellow labware discernible. Since the question specifically asks about the color of the storage box when these items are present, and it hasn't appeared yet, I will check the very end of the video to see if a storage box becomes visible in the background during this action.</think>

<tool_call>{\"name\":\"get_frame\",\"arguments\":{\"video_path\": \"../59309.mp4\", \"timestamp\": 7.5}}</tool_call>

<tool_response>Tool execution failed: Error executing tool get_frame: timestamp (7.5s) exceeds video duration (7.00s)</tool_response>

**Turn 3:**
<think>The frames at the beginning of the video, specifically around 0 to 2 seconds, clearly show the researcher holding a pipette and a small, green, transparent component. In these initial shots, a storage box is visible in the background to the left of the computer monitor. This storage box has a distinct blue color. Although my tool call for 7.5 seconds failed, the question asks for the color of the storage box when the researcher is performing a specific action (holding a pipette and the component), which is consistently shown in the early part of the video. Therefore, the color of the storage box at that time is blue.</think>

<answer>Blue.</answer> 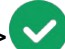

*Figure 19.* **Framewise Detail Probing Example.** The model uses `get_frame` to probe fine-grained visual attributes in a short clip, first querying a plausible timestamp and then adaptively shifting its probe. When a boundary-violating call fails (timestamp exceeds video duration), it remains robust by falling back to earlier verified frames that satisfy the queried action context, and correctly identifies the storage box color as blue.

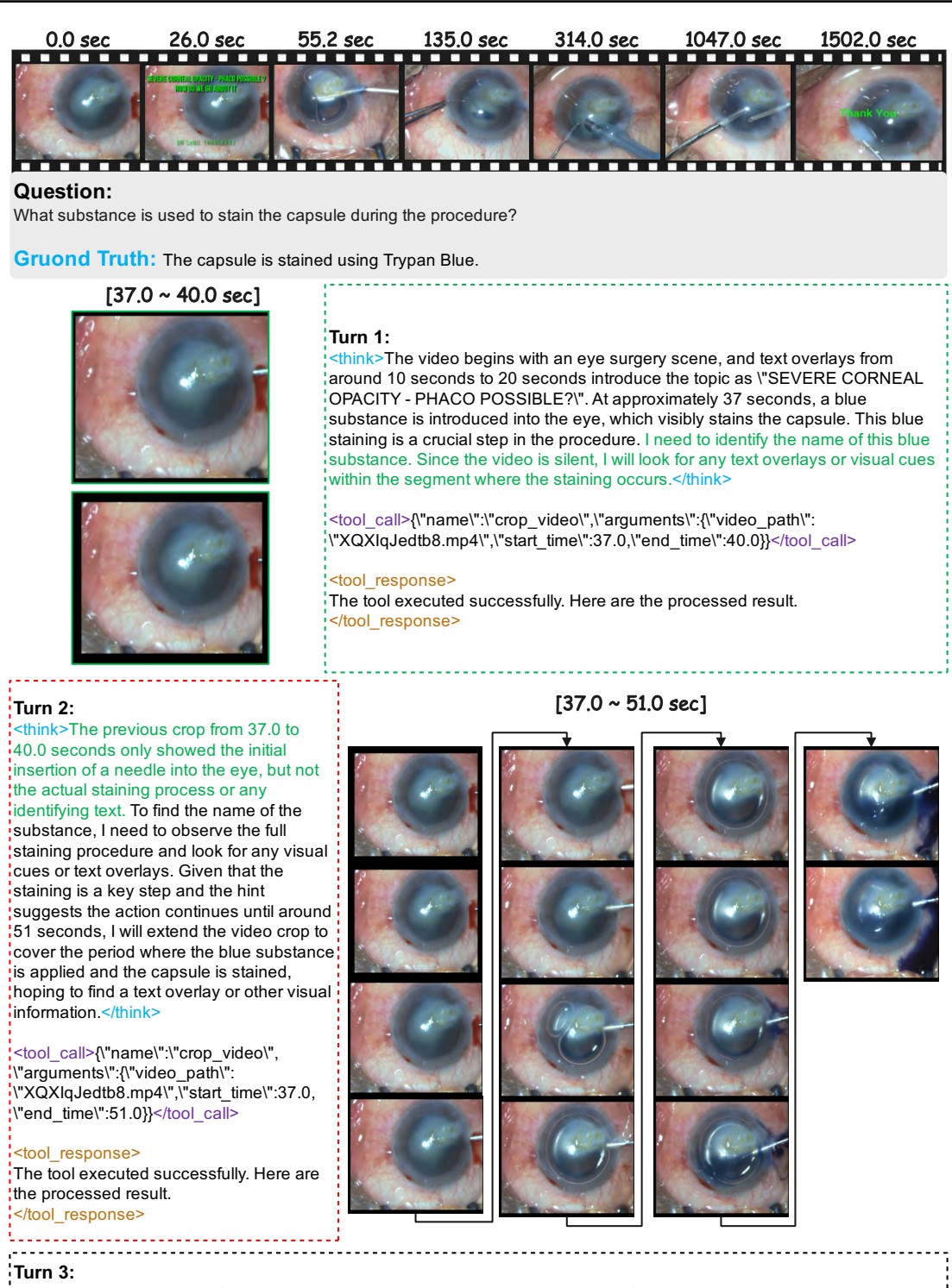

*Figure 20.* **Fine-to-Coarse Evidence Search Example.** The model first performs a tight `crop_video` around the suspected staining moment, but the narrow window fails to include the decisive staining and any identifying cues. It then adaptively expands to a longer interval that covers the full action, enabling evidence-complete verification and correctly answering that the capsule is stained with Trypan Blue.

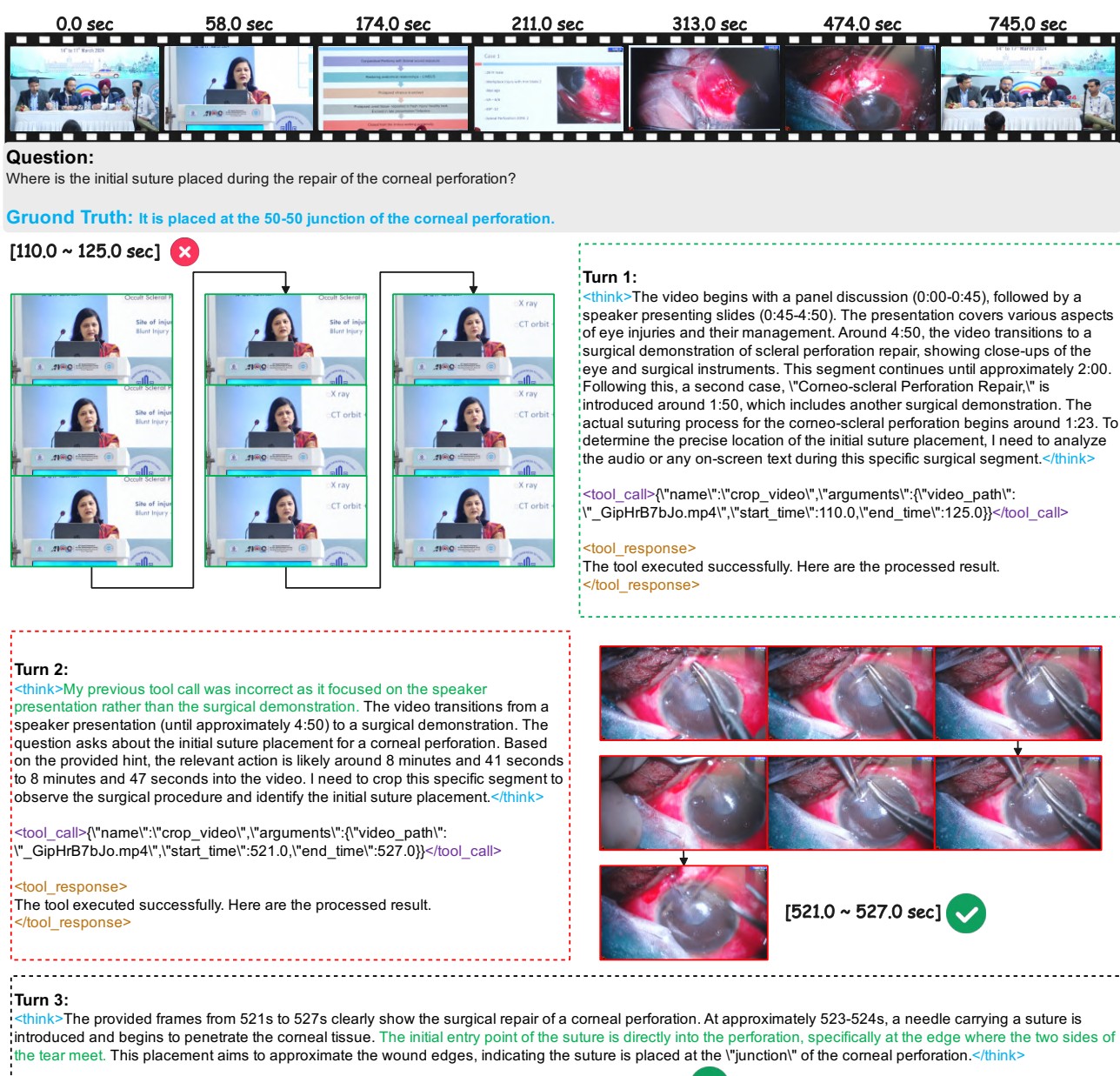

*Figure 21.* **Parallel Time-Jump Self-Correction Example.** The model initially crops an irrelevant segment (speaker presentation) and then explicitly recognizes the mismatch between the retrieved evidence and the queried surgical action. It subsequently performs a parallel jump to the correct surgical window and verifies that the first suture is placed at the 50-50 junction where the wound edges meet.

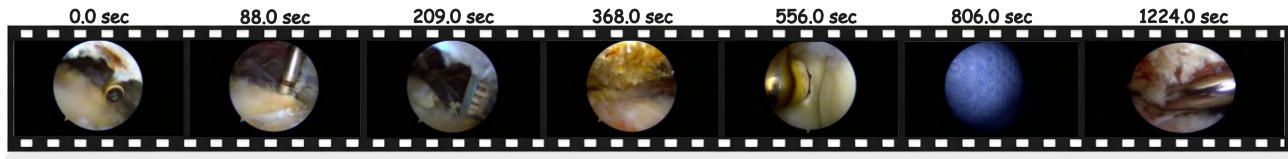

0.0 sec    88.0 sec    209.0 sec    368.0 sec    556.0 sec    806.0 sec    1224.0 sec

**Question:**
When the camera is relatively stable, the image is not significantly obscured by fogging, and several instruments (instrument shafts/tips) are simultaneously visible in the field of view?

**Gruond Truth:** Two

**[120.0 ~ 240.0 sec]**

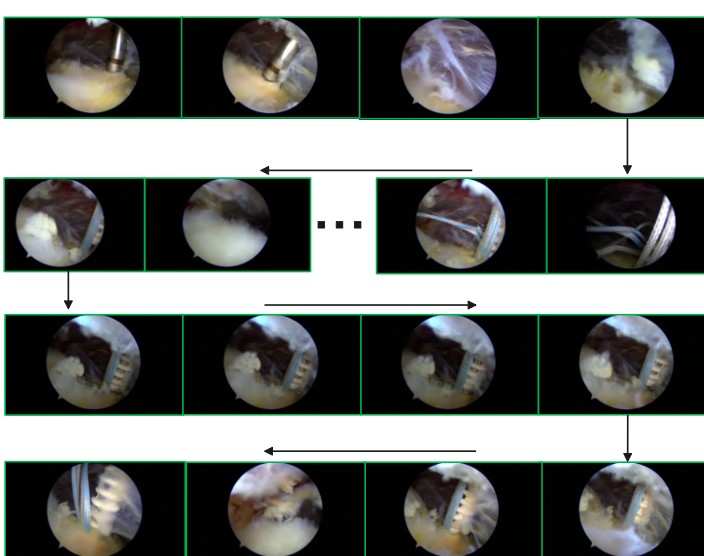

**Turn 1:**
<think>The video is a continuous endoscopic procedure with a stable circular scope view, dominated by soft tissue surfaces, fluid, and periodic instrument interactions. Early on, the camera advances and reorients frequently, with brief moments where the field clears and then becomes partially obscured again by haze or floating debris. As the procedure progresses, the visual rhythm alternates between exploration, contact with tissue, and short bursts of manipulation where an instrument enters, engages, and withdraws. The lighting and color tone shift over time, including phases where the view becomes milky or foggy, phases with more yellow particulate material drifting near the lens, and later intervals where the field is heavily dominated by a blue occluding surface that blocks most anatomical detail. There are also segments where the view becomes uniformly red or orange, consistent with strong occlusion or close contact with tissue or blood film, followed by recovery to a clearer field. Across the full runtime, instruments appear intermittently rather than continuously, and their presence varies from single-instrument contact to brief periods where multiple tools share the frame. The question requires counting how many instruments are simultaneously visible at the moment when a second instrument enters and both are present together. That kind of moment is easy to miss when relying on a global summary, because it depends on a short interval where both shafts or tips are concurrently inside the scope view and not occluded. A reliable approach is to first isolate a broad mid-procedure region where instrument density increases and two-tool interaction is most likely, then tighten the crop around the exact entry moment, and finally extract key frames for an unambiguous count. I will start by cropping a large middle segment that is likely to contain the transition from single-tool work to a two-tool moment, then refine once the two-instrument interval is located.
</think>
<tool_call>{"name":"crop_video","arguments": {"video_path":"cedd7.mp4","start_time":120.0,"end_time":240.0}}</tool_call>

<tool_response>
The tool executed successfully. Here are the processed result.
</tool_response>

**Turn 2:**
<think>
Within the cropped middle segment, there is a clear moment where a second instrument enters the field while the first instrument remains visible, creating a short interval of simultaneous two-tool presence. This happens shortly after the camera stabilizes on a workable area and before the later phases where the field becomes more frequently occluded. The overlap is brief and can be missed if sampling too sparsely, so I will crop more tightly around the entry and overlap interval to reduce motion noise and make the instrument count frame-level verifiable.
I will narrow the segment to the specific window where both tools are concurrently visible, keeping a small buffer on both sides to avoid cutting off the entry or exit.
</think>
<tool_call>{"name":"crop_video","arguments":{"video_path":"cedd7.mp4","start_time": 145.0,"end_time":165.0}}</tool_call>

<tool_response>
The tool executed successfully. Here are the processed result.
</tool_response>

**[145.0 ~ 165.0 sec]**

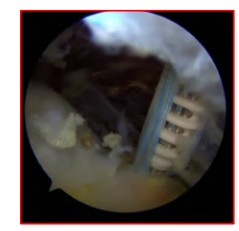 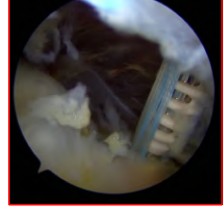

154.0 sec

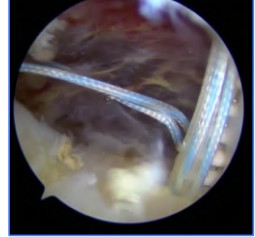

**Turn 3:**
<think>Now I will extract a key frame from the refined clip during the interval where both instruments are clearly present. I will sample a frame near the middle of the overlap to avoid boundary transitions and motion blur. Then I will confirm the count by sampling an adjacent moment as a sanity check in case one tool is partially occluded in a single frame.
</think>
<tool_call>{"name":"get_frame","arguments":{"video_path":"cedd7.mp4","timestamp":154}}</tool_call>

<tool_response>
The tool executed successfully. Here are the processed result.
</tool_response>

**Turn 4:**
<think>
The extracted frames show two distinct instruments simultaneously within the endoscopic field. One instrument is visible as a primary tool shaft and tip engaged in the scene, while a second instrument is also clearly present at the same time, entering from a different direction and occupying a separate region of the circular view. Since both tools are concurrently visible in the same frame rather than appearing sequentially across different moments, the correct instrument count for this moment is two. This directly supports answering the question about simultaneous instrument presence.
</think>

<answer> Two instruments are simultaneously visible. </answer>

*Figure 22.* **Coarse-to-Fine Tool Chaining Example.** The model first performs a broad temporal crop to locate a likely two-instrument overlap, then tightens to a short window around the overlap, and finally uses frame extraction to verify the simultaneous instrument count, yielding the correct answer (two) under stable, low-fog visibility.

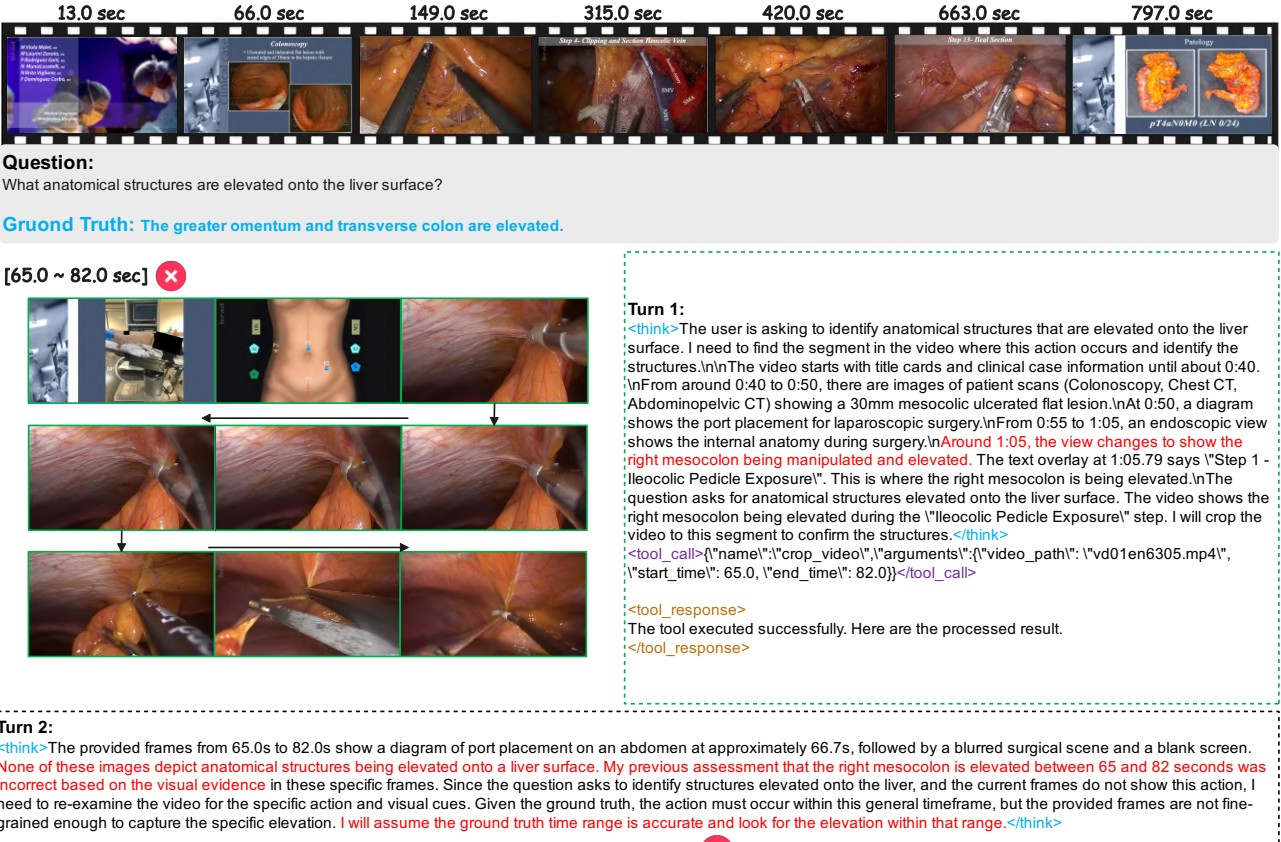

**Question:**

What anatomical structures are elevated onto the liver surface?

**Gruond Truth:** The greater omentum and transverse colon are elevated.

**[65.0 ~ 82.0 sec]** ❌

**Turn 1:**

<think>The user is asking to identify anatomical structures that are elevated onto the liver surface. I need to find the segment in the video where this action occurs and identify the structures.\n\nThe video starts with title cards and clinical case information until about 0:40. \nFrom around 0:40 to 0:50, there are images of patient scans (Colonoscopy, Chest CT, Abdominopelvic CT) showing a 30mm mesocolic ulcerated flat lesion.\nAt 0:50, a diagram shows the port placement for laparoscopic surgery.\nFrom 0:55 to 1:05, an endoscopic view shows the internal anatomy during surgery.\nAround 1:05, the view changes to show the right mesocolon being manipulated and elevated. The text overlay at 1:05.79 says \"Step 1 - Ileocolic Pedicle Exposure\". This is where the right mesocolon is being elevated.\nThe question asks for anatomical structures elevated onto the liver surface. The video shows the right mesocolon being elevated during the \"Ileocolic Pedicle Exposure\" step. I will crop the video to this segment to confirm the structures.</think>

<tool_call>{\"name\":\"crop_video\",\"arguments\":{\"video_path\": \"vd01en6305.mp4\", \"start_time\": 65.0, \"end_time\": 82.0}}</tool_call>

<tool_response>
The tool executed successfully. Here are the processed result.
</tool_response>

**Turn 2:**

<think>The provided frames from 65.0s to 82.0s show a diagram of port placement on an abdomen at approximately 66.7s, followed by a blurred surgical scene and a blank screen. None of these images depict anatomical structures being elevated onto a liver surface. My previous assessment that the right mesocolon is elevated between 65 and 82 seconds was incorrect based on the visual evidence in these specific frames. Since the question asks to identify structures elevated onto the liver, and the current frames do not show this action, I need to re-examine the video for the specific action and visual cues. Given the ground truth, the action must occur within this general timeframe, but the provided frames are not fine-grained enough to capture the specific elevation. I will assume the ground truth time range is accurate and look for the elevation within that range.</think>

**<answer>The right mesocolon and duodenum are elevated onto the liver surface.</answer>** ❌

*Figure 23.* **Evidence-Overrule Failure example.** The model issues a `crop_video` call to verify which structures are elevated onto the liver surface. The returned frames do not show the queried action, and the model notes the mismatch, but it does not proceed with an additional re-localization step and answers based on the prior narrative, yielding an unsupported prediction.

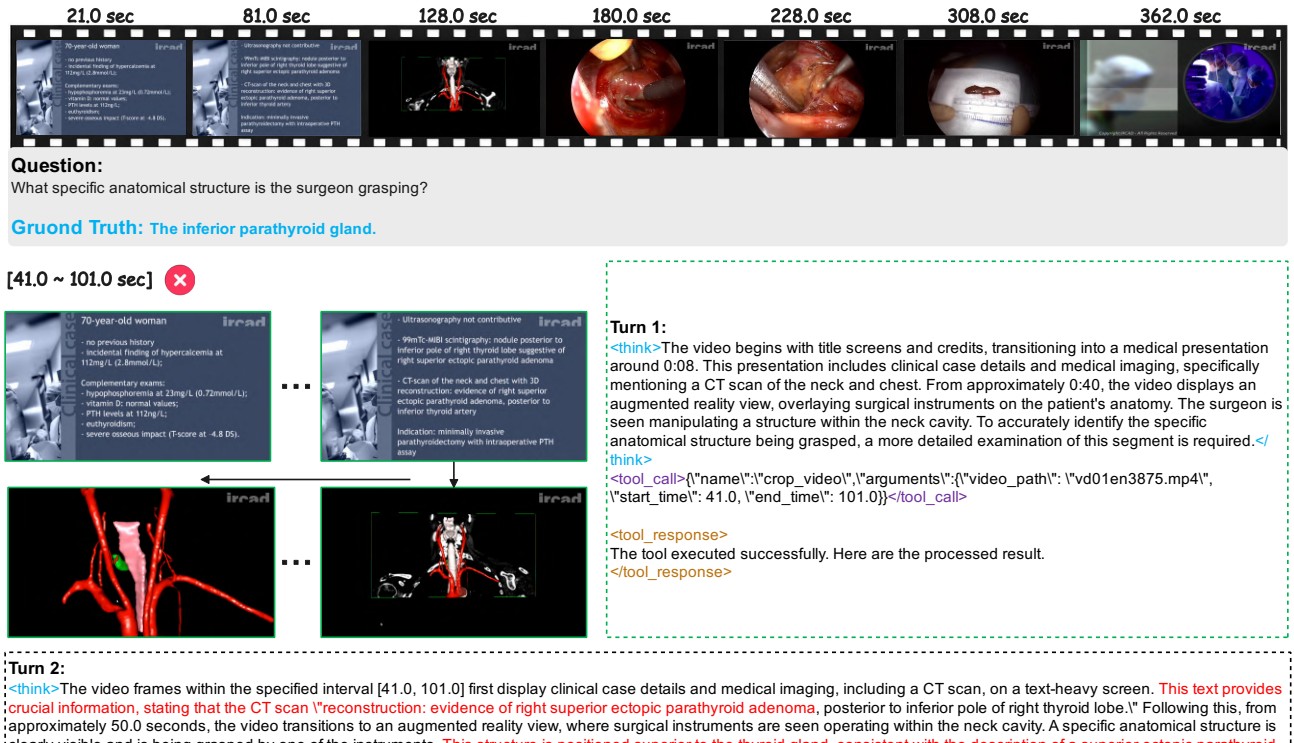

**Figure 24.** **Text-Anchored Entity Binding Drift example.** The model crops a long window containing both case text and the operative augmented view, then anchors on the slide description of an ectopic parathyroid lesion and transfers this prior to the grasped target, leading to an incorrect structure identification despite tool use.

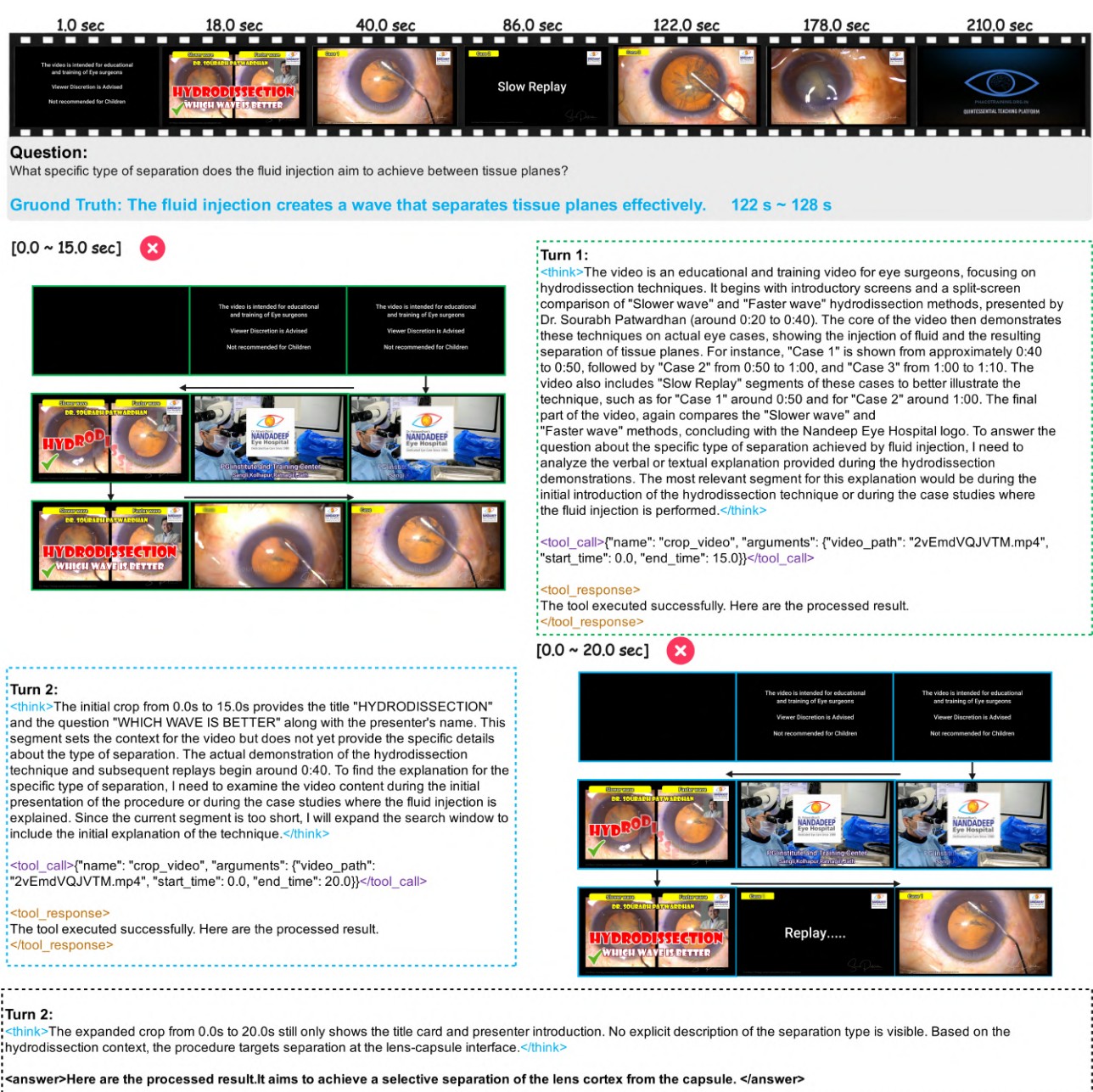

*Figure 25.* **Repetitive Temporal Loop Failure example.** The model issues two successive `crop_video` calls to nearly identical early windows, both anchored near the video start. These calls retrieve only introductory title frames, while the true evidence window lies more than 100 seconds later. After receiving non-discriminative frames, the model marginally extends the same early crop rather than re-localizing to an unexplored region, and ultimately answers from medical domain knowledge without grounding in the target evidence window.

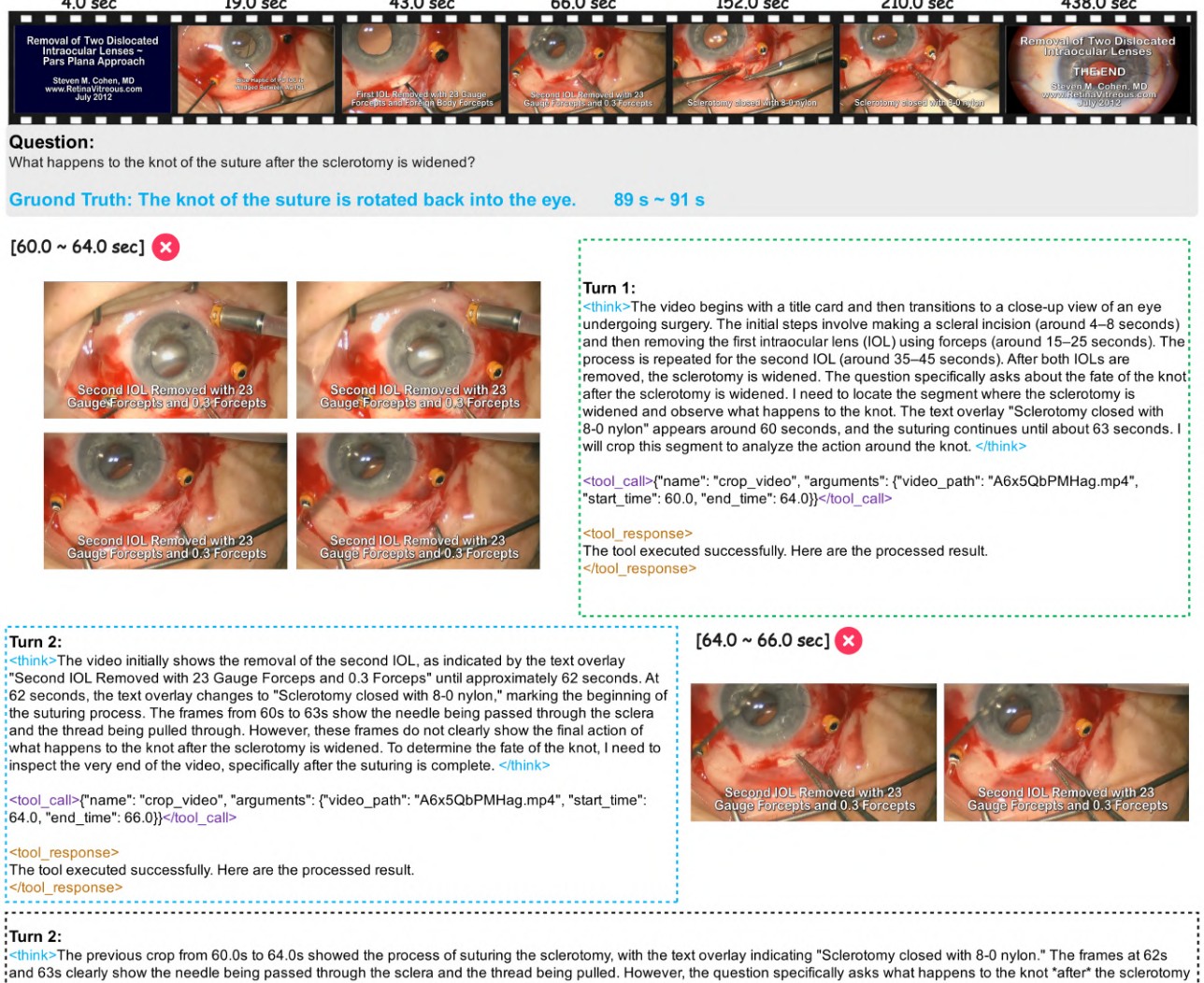

*Figure 26.* **Premature Scan Termination Failure example.** The model performs a sequential scan and retrieves early suture-related actions, including the needle passing through the sclera and the thread being pulled. Although it recognises that the observed frames do not reach the end of the video and that the decisive knot action has not been seen, it stops tool use and commits to a generic answer, missing the terminal action in the ground-truth evidence window.

**Original Clip Prompt**

You are a medical video semantic boundary detector.
You will be given sparse sampled frames (chronological) from a video interval.
Each frame has an absolute timestamp in seconds.

Goal:
Propose semantic change points (cut points) to split the interval into coherent phases/steps.

Rules:
- Output JSON only.
- If duration >= {force_multi_min_dur:.1f}s, you MUST produce at least {min_segments} segments
  (i.e., at least {min_cut_points} cut points), unless there is truly no semantic change.
- Choose cut points adaptively. Typical number of segments for this duration is around {suggest_k},
  but you may output fewer/more, up to {max_segments_cap}.
- Cut points must be strictly increasing, within ({macro_start:.3f}, {macro_end:.3f}) (do NOT include boundaries).
- After splitting, segments must be continuous and cover the whole interval:
  [{macro_start:.3f}, cut1], [cut1, cut2], ..., [last_cut, {macro_end:.3f}]
- Minimum segment length: {min_seg_sec:.2f}s (avoid very short segments).

Output schema:
{{
  "cut_points": [t1, t2, ...],
  "segment_summaries": ["phase1 short summary", "phase2 ...", ...]
}}
Where len(segment_summaries) = len(cut_points) + 1.
Each summary should be ONE concise phrase (do NOT list many phases in one summary).

*Figure 27.* Prompt for original clip segmentation.

**Clip Caption Prompt**

VIDEO_SEGMENT_CAPTION_PROMPT_MED = (
    "You are a helpful assistant that summarizes the content of a video. \n"
    "You will be given a medical video.\n"
    "Write a dense, clinically-grounded caption describing ONLY what is visually supported include the following information: .\n\n"
    "1. The main events in the video. \n"
    "2. The main characters in the video. \n"
    "3. The main locations in the video. \n"
    "4. The main objects in the video. \n"
    "5. The main actions in the video. \n"
    "6. The main emotions in the video. \n"
    "Requirements:\n"
    "1) Focus on medical/surgical semantics: anatomy/target tissue, instruments, actions and workflow step.\n"
    "2) Mention spatial relations if visible (left/right, proximal/distal, superficial/deep, anterior/posterior).\n"
    "3) Avoid hallucinating patient info, diagnosis, or outcomes not visible.\n"
    "Make sure to describe the video in a way that is easy to understand and follow. "
    "Please include as much detail as possible and do not miss any information. ")

*Figure 28.* Prompt for clip caption.

**Caption Merge Prompt**

"You are a medical video dataset curator.\n"
    "Task: Merge TWO chronological blocks of segment captions into ONE SHORTER chronological block.\n"
    "Input format: each block contains one or more lines like:\n"
    "  [start-end] text\n"
    "Rules:\n"
    "- Keep strict chronological order.\n"
    "- Keep the [start-end] time ranges (seconds) at the beginning of each output line.\n"
    "- Output MUST be plain text lines only: NO headings, NO bullet lists, NO markdown.\n"
    "- Strongly compress: keep only surgical/clinical storyline (major steps, key anatomy, major instruments only if essential).\n"
    "- Drop generic filler (e.g., emotions, professionalism, 'operating room environment', long instrument inventories).\n"
    "- Merge adjacent lines if they describe the same step/phase; reduce repetition aggressively.\n"
    "- Output at most 6 lines. If more, merge adjacent lines into broader phases.\n"
    "- Do NOT add new facts.\n"
    "Return ONLY the merged block text.\n"

*Figure 29.* Prompt for caption merge.

**QA Pair Generation Prompt**

```
SYSTEM_PROMPT = r"""
You are an expert dataset curator for video understanding.

You will be given for ONE example:
(A) Global Summary (coarse overview of the full video)
(B) Clip Caption (local window description with fine-grained visual details)
(C) Clip Timecodes [start_time, end_time] in seconds
(D) K = maximum number of QA pairs to produce

Task: Produce BETWEEN 1 and K QA pairs, following ALL rules:

1) One question only (NO multi-part)
- Each item must contain EXACTLY ONE clear English question (<= 50 words).
- Avoid "X and Y" / double-barreled forms.

2) Visual-only evidence (NO knowledge / NO guessing)
- The question must be answerable ONLY by observing the video (visual evidence).
- Do NOT rely on domain knowledge, common sense, narration-only facts, or plausible inference.
- If a QA could be guessed without seeing the video, it is invalid.

3) Clip-only answerable + Global-not-answerable
- Each QA must be answerable using ONLY the Clip Caption.
- Each QA must NOT be confidently answerable using ONLY the Global Summary.
- Prefer details present in Clip Caption but absent/vague in Global Summary.

4) Concrete, fine-grained, non-subjective
- Focus on concrete details: objects/instruments, colors, actions, temporal order, spatial relations.
- Avoid subjective/evaluative questions.

5) Answer requirements (short, definite, no hedging)
- Answer <= 25 words. NO "maybe / likely / could / possibly / typically / approximately".

6) No answer leakage
- Do not reveal/encode the answer in the question.

7) No explicit segment/time mention
- Do NOT mention "this clip/segment" or timestamps/time ranges in the question.
- For every QA item, output start_time and end_time EXACTLY equal to the given Clip Timecodes.

8) Diversity across multiple QAs
- The K QAs must be DISTINCT and cover DIFFERENT visual details (no paraphrases).

Output JSON ONLY as a LIST of objects with length 1..K:
[ {"video_id":"VIDEO.mp4","question":"...","answer":"...","start_time":X,"end_time":Y},
  ...]
"""
```

*Figure 30.* Prompt for QA pair generation.

**VCoT Generation Prompts**

**Round 1 prompt:**
SYSTEM_PROMPT = """You are a helpful assistant for long-video understanding and reasoning.

You may call tools to inspect video segments. Use the tool only when necessary.

<tools>
{"type":"function","function":{"name":"crop_video","description":"Crop a video to a specified duration (return the exact start/end timestamps you selected; no images).","parameters":
{"type":"object","properties":{"video_path":{"type":"string","description":"Path to the video file"},"start_time":{"type":"number","description":"Start time in seconds"},"end_time":
{"type":"number","description":"End time in seconds, must be > start_time"}},"required":["video_path","start_time","end_time"]},"strict":false}}
</tools>

For every function call, wrap a JSON object with the function name and its arguments inside <tool_call></tool_call> tags.
Do NOT output any <tool_response> tags. The tool response will be injected by the system after execution.
The JSON inside <tool_call> must include a "name" key and an "arguments" object.

You will receive a series of image frames sampled from the video (at most 512 frames) for reference.
Frame sampling timeline hint: {frame_hint}

Input you receive:
VIDEO_PATH: {video_path}
TIME_WINDOW_HINT: [{gt_start:.3f}, {gt_end:.3f}]
GROUND_TRUTH_ANSWER: {gt_answer}
**Do not mention in the thinking process that you already know TIME_WINDOW_HINT and GROUND_TRUTH_ANSWER**

Thinking requirements:
- The Thinking section must be non-empty prose (3–6 sentences) with clear evidence and integration.
- Mention time anchors in natural language when you refer to video evidence.
- Use plain ASCII punctuation; avoid placeholders, blank/placeholder/gibberish content, and non-ASCII symbols.

We will follow a coarse-to-fine multi-stage approach:
Phase 1 (global skim & planning — first Thinking section):
- Reconstruct the visual storyline of the entire video by interpreting the sequence of provided frames (silent video).
Do not mention that you are looking at static images or frames; narrate it as a continuous video scene.
- In about 4–6 flowing sentences, narrate what the camera shows across the whole video (settings, actors, transitions).
- As you narrate, sprinkle human-readable time anchors for key moments (not only the final windows).
Allowed styles include: about 297s, around 298–300s, from 4:56 to 5:15, 295–300s, or [296.34s - 320.76s].

Decision rule:
- Put all reasoning inside a single Thinking section.
- If you need to call a tool, output exactly one <tool_call>...</tool_call> and stop (do NOT output Answer in that case).
- If you already have enough evidence to answer, output Answer and stop (do NOT output <tool_call> in that case).

IMPORTANT STYLE RULE:
- Do NOT mix <think>/<answer> tags with "Thinking/Answer" headers in the same output.
- (<tool_call>...</tool_call> is allowed in both styles.)

Output format (exactly one of the two):
(1) If tool needed:
###Thinking
...thinking...
<tool_call>{"name":"crop_video","arguments":{"video_path":"...","start_time":10.0,"end_time":20.0}}</tool_call>

(2) If answering now:
###Thinking
...thinking...
###Answer
...final answer...

Example (temporal grounding):
###Thinking I locate the key action ...
###Answer [340.0, 356.0]

Example (need tool):
###Thinking I can only localize the event roughly from the global frames, so I will crop a tighter window to verify ...
<tool_call>{"name":"crop_video","arguments":{"video_path":"/path/to/video.mp4","start_time":300.0,"end_time":380.0}}</tool_call>

If the question is temporal grounding (i.e., the ground-truth answer is a time interval), then your Answer must be exactly a JSON list of two numbers: [start_time, end_time].
Do not include any extra words, units, markdown, or punctuation outside the brackets.
"""

**Round 2 to Round n prompt：**
FINE_INSPECTION_TEMPLATE = """You are now in Phase 2 (fine-grained inspection), round {round_idx}.

Continue the existing response without repeating earlier content.
First output a Thinking section (3–6 sentences) with evidence, integration, and reflection, and mention time anchors.
Then decide whether to output a <tool_call>...</tool_call> or an Answer section.
If evidence is sufficient, output Answer and stop; otherwise output exactly one <tool_call>...</tool_call> block and stop.

This round includes:
- Attached frames: images from the video segment of this interval (low resolution, ~224px).
- The original QUESTION (for reference): {question}
- TIME_WINDOW_HINT: [{gt_start:.3f}, {gt_end:.3f}]
- GROUND_TRUTH_ANSWER: {gt_answer}
**Do not mention in the thinking process that you already know TIME_WINDOW_HINT and GROUND_TRUTH_ANSWER**
"""

*Figure 31.* Prompt for VCoT generation.

