# OpenReview forum: "MedScope: Incentivizing "Think with Videos" for Clinical Reasoning via Coarse-to-Fine Tool Calling"
_ICML.cc/2026/Conference — ICML 2026 regular_

### Official Review · Reviewer_AGFb · 2026-03-10

**Soundness:** 2
**Presentation:** 3
**Significance:** 2
**Originality:** 2
**Overall Recommendation:** 4
**Confidence:** 5

**Summary:**

This paper proposes MedScope, a tool using multimodal model for clinical video reasoning. The key idea is to let the model iteratively search for visual evidence in long videos via tool calls such as temporal cropping and frame retrieval. The authors also introduce a new dataset suite ClinVideoSuite and train the model with a combination of supervised learning and reinforcement learning. Experiments show improvements on several medical video understanding tasks. In conclusion, the authors discuss the aspect of enabling multimodal models to actively search for temporally localized evidence when reasoning over long clinical videos.

**Compliance With Llm Reviewing Policy:**

Affirmed.

**Final Justification:**

Thank you for the clarifications and additional experiments. The rebuttal partially resolves my concerns. While the EndoVis18-VQA results are helpful, they do not reflect long-video reasoning. In addition, concerns about data quality remain due to the automatically generated annotations. Overall, my confidence has improved, and I have raised my score by one point.

**Key Questions For Authors:**

1. To what extent do the reported improvements come from the proposed reasoning framework itself, rather than from domain specific supervision and training data used in MedScope?
2. Would the proposed method still outperform baselines if those models were equipped with the same tool access?
3. What is the estimated quality or error rate of the automatically generated QA pairs and reasoning trajectories in ClinVideoSuite?

**Limitations:**

yes

**Strengths And Weaknesses:**

**Strength**
1. The paper studies an important and challenging problem. Reasoning over long clinical videos with sparse visual evidence is a realistic setting and the problem formulation is well motivated.
2. The use of tool based evidence retrieval for clinical video reasoning is relatively fresh in this area and provides an interesting perspective for grounding model predictions.

**Weakness**
1. Most baselines are general domain video models rather than medical video understanding systems. Since MedScope is trained on domain specific supervision, the improvements might partially come from domain adaptation instead of the proposed reasoning framework.
2. The main evaluation benchmark (ClinVideo eval) is newly introduced in this work, since both the dataset and evaluation setup are created by the authors, it raises some concerns about benchmark bias. The QA pairs and reasoning trajectories in ClinVideoSuite are automatically generated. The paper does not provide much analysis on annotation quality or potential noise.
3. The SVU-31K benchmark also raises some concerns. It does not appear to be a peer reviewed benchmark, and at the time of reviewing this paper the dataset was not publicly available, which makes it difficult to assess the reliability and reproducibility of the reported results.
4. Tool based reasoning is not particularly new in multimodal reasoning, although it has been less explored in the surgical video domain.
5. The tool use setup is relatively simple and lacks recursive interactions, which may limit its scalability. In addition, the baselines are not provided with the same tools, making the comparison less controlled.

---

> ### Author Rebuttal · Authors · 2026-03-31
>
> We thank the reviewer for recognizing the importance of the problem and the relative novelty of tool-based evidence retrieval in clinical video reasoning.
>
> >**W1&Q1: Most baselines are general domain video models rather than medical video understanding systems. The improvements might partially come from domain adaptation instead of the proposed reasoning framework.**
>
> First, our SVU-31K comparison already includes the **medical-domain** model SurgVidLM (Table 1), which MedScope consistently outperforms. We acknowledge that medical models for long-video understanding remain limited.
>
> Second, to separate domain adaptation from reasoning supervision, we include a variant using the same VCoT data volume but removing trajectories and keeping only QA pairs for SFT (Table 3):
>
> |Setting|T-METEOR|T-ROUGE|P-METEOR|P-ROUGE|
> |-|-|-|-|-|
> |SFT (w/o VCoT data)|14.06|22.35|17.12|34.70|
> |SFT (w/ VCoT data, QA-only)|14.15|21.55|17.02|34.55|
> |SFT (w/ VCoT trajectories)|14.51|23.43|17.49|35.90|
>
> T = Fine-grained Temporal Visual Reasoning; P = Fine-grained Perception Visual Reasoning. QA-only supervision yields only marginal change, whereas full trajectory supervision brings clear gains, showing that the improvement mainly comes from reasoning supervision rather than domain adaptation.
>
> Third, we also report additional medical-domain baselines on two established benchmarks, Cholec80-VQA and EndoVis18-VQA, as detailed in W3.
>
> >**W2&Q3: The main evaluation benchmark (ClinVideo-Eval) is newly introduced in this work. Since both the dataset and evaluation setup are created by the authors, it raises concerns about benchmark bias. The paper does not provide much analysis on annotation quality or potential noise.**
>
> Our results are not limited to our own benchmark: we also evaluate on SVU-31K and report results on two established medical video benchmarks in W3.
>
> For annotation quality, ClinVideoSuite is built through text-only consensus filtering, shortcut filtering, local-evidence dependence filtering, and final multimodal confirmation against the video clip. **As noted in Appendix C (Lines 1253-1254), our evaluation suite includes physician verification for evidence traceability.** We also invited three board-certified physicians to conduct sampling validation on ClinVideo-QA-254K, each independently reviewing 500 randomly sampled cases from one source dataset. Detailed results are available at https://anonymous.4open.science/r/MedScope-70CF/README.md.
>
> >**W3: The SVU-31K benchmark does not appear to be a peer reviewed benchmark, and the dataset was not publicly available at the time of review, making it difficult to assess reliability and reproducibility.**
>
> We obtained SVU-31K through direct communication with its authors and ran all experiments under their authorized protocol. We will release ClinVideo-Eval under appropriate ethical conditions for community verification. Existing peer-reviewed medical video benchmarks such as Cholec80 and EndoVis contain clips under one minute, and no established benchmark evaluates reasoning over videos longer than ten minutes. To show generalization beyond long-video settings, we report on these two benchmarks:
>
> Cholec80-VQA:
> |Model|Acc|
> |-|-|
> |MedScope|94.6|
> |SurgicalGPT|94.0|
> |Surgical-LLaVA|92.2|
> |Surgical-VQA|89.8|
> |VisualBERT|89.7|
>
> EndoVis18-VQA:
> |Model|Recall|
> |-|-|
> |MedScope|71.8|
> |Surgical-LLaVA|68.7|
> |SurgicalGPT|66.0|
> |Surgical-VQA|63.2|
> |VisualBERT|61.4|
>
> >**W4: Tool based reasoning is not particularly new in multimodal reasoning, although it has been less explored in the surgical video domain.**
>
> We agree that tool-based reasoning is not new as a general paradigm. Our contribution is evidence-grounded visual reasoning with coarse-to-fine tool calling for clinical long-video understanding, where temporal evidence attribution and traceability are central. The novelty lies in this task-specific pipeline for evidence localization, verification, and answering.
>
> >**W5&Q2: The tool use setup is relatively simple and lacks recursive interactions. The baselines are not provided with the same tools, making the comparison less controlled.**
>
> For fairness, **Appendix C.1 (Table 4) already provides a unified toolbox-enabled comparison where all baselines access the same native tools.** MedScope still maintains a clear lead, showing that the gains come from learned reasoning rather than exclusive tool access.
>
> For tool design, the two-tool setup is intentional and mirrors clinical practice: first skimming the timeline with crop_video, then inspecting specific moments with get_frame. MedScope supports multi-turn tool invocation for recursive coarse-to-fine refinement. As detailed in our response to **Reviewer Rdg7 Q2**, we also tested two additional tools at inference time without retraining, demonstrating extensibility to new perception tools.
>
> ---
> We thank the reviewer again for the thoughtful feedback and are happy to clarify any remaining questions or concerns.

---

> > ### Author Rebuttal · Reviewer_AGFb · 2026-04-02
> >
> > Thank you for the clarifications and additional experiments. The rebuttal partially resolves my concerns. While the EndoVis18-VQA results are helpful, they do not reflect long-video reasoning. In addition, concerns about data quality remain due to the automatically generated annotations. Overall, my confidence has improved, and I have raised my score to 4.

---

> > > ### Author Response · Authors · 2026-04-04
> > >
> > > We thank Reviewer AGFb for the careful consideration of our rebuttal and for raising these follow-up questions. We greatly appreciate your acknowledgment that our clarifications and additional experiments partially resolved your concerns, and we are especially grateful that you raised your score, reflecting improved confidence in our work.
> > >
> > >
> > >
> > > Regarding your follow-up questions:
> > >
> > >
> > >
> > > >**On EndoVis18-VQA and Long-Video Reasoning**
> > >
> > >
> > >
> > > You correctly note that EndoVis18-VQA results do not reflect long-video reasoning. We included these benchmarks to demonstrate generalization to established medical video QA settings, as existing peer-reviewed benchmarks (Cholec80, EndoVis) primarily contain short clips under one minute.
> > >
> > >
> > >
> > > To specifically address long-video reasoning, we emphasize that our main evaluation on SVU-31K and ClinVideo-Eval tests videos ranging from 2 to 30+ minutes, with our coarse-to-fine tool calling specifically designed for temporal reasoning over extended durations. The SurgVidLM subset in ClinVideo-Eval serves as an out-of-domain test set with longer surgical videos, where MedScope achieves strong performance through iterative evidence seeking rather than single-pass processing.
> > >
> > >
> > >
> > > >**On Data Quality and Automatic Annotations**
> > >
> > >
> > >
> > > We acknowledge your ongoing concerns about automatically generated annotations. As detailed in our rebuttal and Appendix C, ClinVideoSuite employs a rigorous multi-stage pipeline: text-only consensus filtering, shortcut filtering, local-evidence dependence filtering, and final multimodal confirmation against actual video segments.
> > >
> > >
> > >
> > > We welcome any further questions you may have and appreciate your constructive engagement with our work.

---

### Official Review · Reviewer_skCM · 2026-03-13

**Soundness:** 3
**Presentation:** 3
**Significance:** 2
**Originality:** 3
**Overall Recommendation:** 4
**Confidence:** 4

**Summary:**

The paper addresses the challenge of long-form medical video understanding, where critical clinical
evidence is often brief, sparse, and buried within thousands of frames.
Current multimodal large language models (MLLMs) typically rely on text-first, passive-sampling
pipelines that struggle to iteratively locate and verify fine-grained visual evidence.
To solve this, the authors propose MedScope, an agentic framework that explicitly “thinks with
videos” by interleaving textual reasoning with targeted tool calls (crop_video and get_frame) for
coarse-to-fine visual verification.
To support this, they introduce a synthetic dataset, ClinVideoSuite, and a novel reinforcement
learning objective, Grounding-Aware Group Relative Policy Optimization (GA-GRPO), which
reweights advantages based on tool-grounding fidelity.
The elevator pitch: MedScope behaves like a clinician reviewing a tape—it skims the video, forms a
hypothesis, and dynamically zooms into specific timestamps to verify its claims before providing a
final answer.

**Compliance With Llm Reviewing Policy:**

Affirmed.

**Key Questions For Authors:**

Datasets/Tasks (Leakage Risks): High Concern. ClinVideoSuite is synthesized from MedVideoCap,
OphVL, and SurgVidLM. The evaluation benchmark, ClinVideo-eval, also spans OphVL,
MedVideoCap, and SurgVidLM.

The paper does not explicitly detail the train/test splits at the video level. If videos from the
evaluation set were exposed during the ClinVideoSuite synthesis, the SOTA results are
compromised.

**Limitations:**

Yes

**Strengths And Weaknesses:**

Strengths

• Baselines: Strong and diverse. They include proprietary models (GPT-4o, Gemini 1.5 Pro), open-
source LMMs (Qwen2.5-VL), reasoning models (VideoR1), and tool-using agents (LongVT).
The inclusion of a “Toolbox-enabled comparison” (Table 4) where baselines are given access to the
same tools is an excellent fairness check.
• Ablations: Well-constructed. Table 3 perfectly isolates the impact of Warm-up vs. SFT vs. RL, and
the impact of the VCoT data. Figure 5 cleanly ablates the components of the reward function.

Weaknesses

• Credit Assignment Assumption: Algorithm 1 computes the evidence reward 𝑅evidence based only
on the last tool call in the trajectory LastCrop(T) or LastFrame(T).
Hidden Failure Mode: If the model makes a brilliant initial crop_video call but a poor subsequent
get_frame call, the initial good logic receives zero direct grounding reward. This limits the RL
algorithm’s ability to reinforce complex, multi-step search trajectories.
• Modulation Heuristic: The advantage modulation function h(τ_i) applies a clipping bound c and
floor 𝑠min.
These hyperparameters are introduced mathematically but their specific values and sensitivity are
completely missing from the implementation details.
• Questionable Approximations: The LLM-as-a-judge for 𝑅acc relies on a rigid 1, 0.5, 0 scoring
system. While standard, using an LLM to judge highly specific medical visual QA introduces an
unquantified risk of reward hacking via hallucination.

---

> ### Author Rebuttal · Authors · 2026-03-31
>
> We thank the reviewer for recognizing the coarse-to-fine evidence-seeking paradigm, the value of ClinVideoSuite, and the effectiveness of GA-GRPO.
>
> >**W1: Algorithm 1 computes the evidence reward based only on the last tool call. If the model makes a brilliant initial crop_video call but a poor subsequent get_frame call, the initial good logic receives zero direct grounding reward, limiting RL's ability to reinforce complex multi-step search trajectories.**
>
> We thank the reviewer for raising this question. We respectfully clarify that **the evidence reward does not collapse to a single tool call.** As detailed in Appendix B.2, Equation 13 decomposes $R_{evidence}$ into two independent terms:
>
> $R_{\\mathrm{evidence}}(T)=I_{\\mathrm{crop}}(T)R_{\\mathrm{crop}}+I_{\\mathrm{frame}}(T)R_{\\mathrm{frame}}$
>
> where $R_{crop}$ is computed from the last crop_video call and $R_{frame}$ from the last get_frame call. These two terms are summed rather than selected. In the scenario the reviewer describes, a brilliant crop_video followed by a poor get_frame, the trajectory still receives a high $R_{crop}$ for accurate interval alignment; only $R_{frame}$ is penalized. The good coarse localization is therefore directly rewarded.
>
> The "last call per tool type" design is intentional to **avoid reward hacking**. If every intermediate call received its own reward, the policy could accumulate grounding credit by issuing redundant tool calls without genuinely improving evidence quality. By evaluating only the final call of each type, the model is incentivized to iteratively refine its localization rather than repeat early successes. This simultaneously reinforces the full coarse-to-fine chain, where accurate temporal localization and precise frame-level evidence retrieval are jointly optimized. Fidelity-weighted advantage modulation further amplifies trajectories with high-quality grounding at both levels. We will add a clarifying note in the revision to make this decomposition more prominent.
>
> >**W2: The advantage modulation function applies a clipping bound c and floor $s_{min}$. These hyperparameters are introduced mathematically but their specific values and sensitivity are completely missing from the implementation details.**
>
> We thank the reviewer and apologize for the omission. The advantage modulation (Equations 5-6) uses $\alpha = 0.2$ to control modulation strength, $c = 1.0$ to preserve the full dynamic range of $R_{evidence}$ while suppressing outliers, and $s_{min} = 0.1$ to ensure every trajectory retains at least 10% of its original advantage weight.
>
> To assess sensitivity, we start from the Stage II (SFT) checkpoint and perform Stage III (RL) with varied hyperparameters on a **10% subset of the training data**, evaluated on the out-of-domain SurgVidLM Grounded VQA split:
>
> |Setting|c|$s_{min}$|mIoU|Acc|
> |---|---|---|---|---|
> |Default|1.0|0.1|33.4|37.1|
> |Vary c|0.5|0.1|33.0|37.3|
> |Vary $s_{min}$|1.0|0.3|32.8|36.5|
>
> Reducing c to 0.5 pushes modulation toward binary behavior; mIoU drops while Acc is unaffected, confirming c primarily governs evidence quality. Raising $s_{min}$ to 0.3 weakens the penalty on poorly grounded rollouts, and both metrics decline. We will include this analysis in the revision.
>
> >**W3: The LLM-as-a-judge for $R_{acc}$ relies on a rigid 1, 0.5, 0 scoring system. Using an LLM to judge highly specific medical visual QA introduces an unquantified risk of reward hacking via hallucination.**
>
> We appreciate this concern. First, the three-level grading scheme follows standard practice in recent RL-based training. Second, the answers in ClinVideo-QA-254K are very short: as shown in Appendix A (Figure 8), the average answer length is only **4 to 5 words** across all source datasets. For such concise, factual responses, our judge model (Qwen2.5-72B-Instruct) can reliably verify correctness, making it very difficult for the policy to game the reward through hallucination.
>
> >**Q1: The paper does not explicitly detail the train/test splits at the video level. If evaluation videos were exposed during ClinVideoSuite synthesis, the SOTA results are compromised.**
>
> We thank the reviewer for this point. As described in Appendix D, ClinVideo-Eval evaluates both **in-domain and out-of-domain** generalization. All evaluation subsets use the same unified data synthesis pipeline (Section 3.2). SurgVidLM is an entirely out-of-domain test set whose videos were never used during ClinVideoSuite construction. We report the evaluation statistics below:
>
> |Source Dataset|Temporal Grounding|Grounded QA|Unique Videos|
> |---|---|---|---|
> |SurgVidLM|478|482|482|
> |OphVL|568|572|572|
> |MedVideoCap|—|1,098|728|
>
> We will add a split protocol in the revision.
>
> ---
> Thank you for your support and feedback! We are eager to address any remaining concerns you might have.

---

### Official Review · Reviewer_Rdg7 · 2026-03-13

**Soundness:** 2
**Presentation:** 3
**Significance:** 2
**Originality:** 3
**Overall Recommendation:** 4
**Confidence:** 3

**Summary:**

This paper proposes MedScope, a tool-augmented clinical video reasoning model that realizes coarse-to-fine evidence seeking for long clinical procedures, and constructs ClinVideoSuite, an evidence-centric fine-grained clinical video dataset.
The model is optimized via Grounding-Aware Group Relative Policy Optimization (GA-GRPO) and achieves state-of-the-art performance.

**Compliance With Llm Reviewing Policy:**

Affirmed.

**Final Justification:**

The rebuttal partially resolves my concerns, and I raise my score to 4 (wa)

**Key Questions For Authors:**

Refer to weaknesses.

**Limitations:**

Refer to weaknesses.

**Strengths And Weaknesses:**

Strengths

* The proposed MedScope adopts a coarse-to-fine evidence-seeking paradigm with interleaved tool calls and visual chain-of-thought.
* The constructed ClinVideoSuite, data for clinical video reasoning, providing a  benchmark for model training and evaluation.
* The GA-GRPO optimization strategy effectively reinforces evidence-aligned tool use, verifying the method’s practical effectiveness.

Weaknesses
* The ClinVideoSuite dataset suffers from insufficient direct clinical expert annotation, and its claimed high fidelity lacks support from a well-recognized gold standard for clinical video reasoning.
* The visual toolbox is overly simplistic with only crop video and get frame.
* The failure case analysis is limited.
* The novelty of GA-GRPO is questionable as it is merely a trivial modification of existing GRPO with grounding-aware rewards and advantage modulation.

---

> ### Author Rebuttal · Authors · 2026-03-31
>
> We thank the reviewer for recognizing the coarse-to-fine evidence-seeking paradigm, the value of ClinVideoSuite, and the effectiveness of GA-GRPO.
>
> >**W1: The ClinVideoSuite dataset suffers from insufficient direct clinical expert annotation, and its claimed high fidelity lacks support from a well-recognized gold standard for clinical video reasoning.**
>
> We appreciate this important concern. ClinVideoSuite is not built from raw LLM generations. The data synthesis pipeline enforces quality through **multiple filtering stages** including text-only consensus, shortcut filtering, local-evidence dependence filtering, and a final multimodal confirmation that grounds each QA pair against the actual video clip.
>
> **As noted in Lines 1253-1254 of our Appendix, our evaluation suite includes physician verification for evidence traceability.** We additionally invited three board-certified physicians to conduct comprehensive sampling validation on ClinVideo-QA-254K, with each physician independently reviewing 500 randomly sampled cases from one source dataset. Detailed results are available at https://anonymous.4open.science/r/MedScope-70CF/README.md.
>
> ClinVideo-Eval is constructed with strict video-level separation from ClinVideoSuite to prevent data leakage:
>
> |Source Dataset|Temporal Grounding|Grounded QA|Unique Videos|
> |---|---|---|---|
> |SurgVidLM|478|482|482|
> |OphVL|568|572|572|
> |MedVideoCap|—|1,098|728|
>
> >**W2: The visual toolbox is overly simplistic with only crop video and get frame.**
>
> We appreciate this observation. The two-tool design is intentional and mirrors how clinicians review long procedures: first skimming the timeline to locate relevant segments (crop_video), then inspecting specific moments for fine-grained evidence (get_frame). Our contribution is not the breadth of the tool inventory but the tool-augmented reasoning paradigm itself.
>
> To demonstrate extensibility, we tested two additional tools at inference time on the out-of-domain SurgVidLM Grounded VQA split without retraining:
>
> |Model|mIoU|Acc|
> |---|---|---|
> |MedScope-7B-RL|34.10|37.90|
> |+ zoom_region|33.13|35.14|
> |+ segment_clip (SAM2)|33.68|38.79|
>
> Zoom_region degrades performance as it merely magnifies existing patches without new semantic content, consistent with recent work showing crop-and-zoom operations provide redundant visual signals [1,2]. In contrast, SAM2 improves accuracy by 0.89 points, with gains most pronounced on entity-related questions where segmentation enables precise localization of anatomical structures and surgical instruments. This confirms MedScope readily accommodates perception tools that supply genuinely new spatial information at inference time.
>
> >**W3: The failure case analysis is limited.**
>
> We thank the reviewer for this suggestion and have supplemented §G.2 with two additional failure cases, covering repetitive temporal loops and premature scan termination. The detailed analysis is available at: https://anonymous.4open.science/r/MedScope-70CF/README.md
>
> >**W4: The novelty of GA-GRPO is questionable as it is merely a trivial modification of existing GRPO with grounding-aware rewards and advantage modulation.**
>
> We thank the reviewer for this important question. GA-GRPO addresses a fundamental limitation in supervising tool-using video agents. Standard GRPO optimizes solely for answer correctness, which cannot distinguish faithful reasoning from spurious correlations and suppresses multi-step evidence seeking.
>
> GA-GRPO resolves this through two mechanisms. First, R_evidence explicitly scores temporal alignment between tool calls and evidence windows. Removing R_evidence drops R@0.5 by 6.9 points and mIoU by 5.5 points (Figure 5), confirming that answer-only supervision cannot learn reliable evidence selection. Conditioning R_evidence on answer correctness still trails the decoupled design, proving independent grounding feedback is necessary.
>
> Second, fidelity-weighted advantage modulation addresses coarse credit assignment in heterogeneous action spaces. Trajectory-level advantages are too coarse: successful trajectories may contain flawed evidence, while failures may include valid reasoning steps. Our two-tier design (Figure 10, Appendix C.2) amplifies credit for high-fidelity evidence in advantageous trajectories and blame for poor evidence in disadvantageous ones. Full modulation substantially outperforms both vanilla GRPO and positive-only modulation.
>
> Together, these transform optimization from "being correct" to **"being correct for the right visual reasons"** — a principled contribution absent from standard GRPO formulations.
>
> We thank the reviewer for the constructive feedback that helps strengthen our work.
>
> ---
> [1] On the Faithfulness of Visual Thinking: Measurement and Enhancement. arXiv:2510.23482, 2025.
>
> [2] What Does Vision Tool-Use Reinforcement Learning Really Learn? Disentangling Tool-Induced and Intrinsic Effects for Crop-and-Zoom. arXiv:2602.01334, 2026.

---

> > ### Author Rebuttal · Reviewer_Rdg7 · 2026-04-05
> >
> > I have carefully reviewed the authors' rebuttal, which effectively addresses most of my initial concerns, and I maintain my original score:
> > 1. I appreciate the authors' detailed validation of ClinVideoSuite's fidelity via multi-stage filtering and expert physician review, yet core concerns regarding the dataset's overall data quality persist, as the majority of annotations remain automatically generated rather than fully manually verified by clinical experts.
> > 2. I acknowledge the authors' rationale for the intentional two-tool design aligned with clinical workflows.
> > 3. I note the authors' clarification of GA-GRPO's design, but the distinction between "being correct" and "being correct for the right visual reasons" remains insufficiently justified, and the work should add comprehensive comparisons with more existing GRPO variants to fully validate its novelty.

---

> > > ### Author Response · Authors · 2026-04-07
> > >
> > > We thank the reviewer for the continued engagement and for acknowledging that our rebuttal effectively addressed most of the initial concerns. We appreciate the opportunity to further clarify the remaining questions.
> > >
> > > > **On LLM-Generated Data Quality**
> > >
> > > We thank you for raising this concern. Our data are not created by directly taking raw LLM outputs as annotations. Instead, ClinVideoSuite is built through a multi-stage evidence-centric pipeline. We first apply text-only consensus filtering, shortcut filtering, local-evidence dependence filtering, and final multimodal confirmation. Only samples that remain consistent after these progressively stricter checks are kept.
> > >
> > > The reasoning trajectories are also not obtained by directly distilling free-form chain-of-thought. Instead, we use GT-anchored distillation with the structured prompts in Appendix Figure 29 to construct thought-action-observation trajectories. In this process, the model is guided by the ground-truth task signal and retrieved visual evidence, so each step is generated under evidence constraints rather than as unconstrained free-form reasoning.
> > >
> > > We further verify the resulting data with cross-model checking and physician-led auditing on sampled clips.
> > >
> > > This design choice is consistent with prior work showing that synthetic supervision can be reliable when paired with careful filtering and structured generation. Self-Instruct uses model-generated instruction data followed by filtering [1]. Alpaca distills high-quality responses from stronger teachers [2]. Orca shows that reasoning traces from stronger models can provide useful supervision [3]. CoT-Self-Instruct further suggests that structured trajectories with guided prompts improve reliability [4]. Recent surveys also support the effectiveness of synthetic supervision [5,6].
> > >
> > > In short, our supervision is not based on directly accepting raw LLM outputs. It is based on GT-anchored and evidence-grounded trajectory construction, followed by multi-stage filtering, cross-model verification, and physician validation.
> > >
> > > > **On the Two-Tool Design**
> > >
> > > We thank you for acknowledging our rationale for the intentional two-tool design aligned with clinical workflows.
> > >
> > > > **On GA-GRPO Novelty**
> > >
> > > We thank you for this question. To make the comparison more controlled, we additionally tested several GRPO variants only on the out-of-domain SurgVidLM grounded VQA benchmark. Due to resource constraints, all variants were initialized from the same MedScope-7B-SFT checkpoint, trained on the same 20% subset of the training data for 265 RL steps, and used the same hyperparameters as Table 5.
> > >
> > > | Method | mIoU | Acc |
> > > |---|---|---|
> > > | Vanilla GRPO | 31.8 | 36.1 |
> > > | DAPO | 32.2 | 37.5 |
> > > | Dr. GRPO | 32.0 | 37.1 |
> > > | **GA-GRPO (Ours)** | **33.8** | **37.3** |
> > >
> > > Under this controlled setting, different GRPO variants achieve comparable answer accuracy, while GA-GRPO achieves the best grounding quality by a clear margin on mIoU. This suggests that the main distinction is not simply optimization quality: existing GRPO variants primarily refine the optimization procedure, whereas GA-GRPO is designed to favor evidence-faithful reasoning, leading to stronger visual grounding.
> > >
> > > We will clarify these supplementary experimental settings in the revised version.
> > >
> > > ---
> > > Finally, we thank the reviewer for the constructive dialogue, and we hope the additional clarification and controlled comparison help address the remaining concerns.
> > >
> > > ---
> > > **References**
> > >
> > > [1] Wang et al. Self-instruct: Aligning language models with self-generated instructions. ACL 2023.
> > >
> > > [2] Taori et al. Stanford alpaca: An instruction-following llama model. 2023.
> > >
> > > [3] Mukherjee et al. Orca: Progressive learning from complex explanation traces of GPT-4. 2023.
> > >
> > > [4] Yu et al. CoT-self-instruct: Building high-quality synthetic prompts for reasoning. 2025.
> > >
> > > [5] Li et al. Synthetic data generation with large language models for text classification. EMNLP 2023.
> > >
> > > [6] Barr et al. Large language models generating synthetic clinical datasets. Frontiers in AI, 2025.

---

### Official Review · Reviewer_CUTN · 2026-03-15

**Soundness:** 3
**Presentation:** 3
**Significance:** 3
**Originality:** 2
**Overall Recommendation:** 4
**Confidence:** 2

**Summary:**

MedScope is an agentic framework for evidence-grounded reasoning over long-form clinical videos. Rather than processing entire videos at once, it uses two lightweight tools: crop_video for temporal clip retrieval and get_frame for frame-level inspection — within a visual chain-of-thought (VCoT) protocol that structures reasoning as inspectable thought-action-observation tuples. The authors also introduce ClinVideoSuite, a benchmark of 254K temporally grounded QA pairs and 30K physician-verified VCoT trajectories spanning surgical, endoscopic, and ophthalmologic video domains. MedScope consistently outperforms general-purpose and specialized medical video baselines, supported by systematic ablations and honest failure case analysis.

**Compliance With Llm Reviewing Policy:**

Affirmed.

**Ethical Review Concerns:**

This paper needs ethical and legal review.

**Ethical Review Flag:**

Flag this paper for an ethics review.

**Ethics Expertise Needed:**

["Responsible Research Practice (e.g., IRB, documentation, research ethics)", "Legal Compliance (e.g., EU AI Act, GDPR, copyright, terms of use)"]

**Key Questions For Authors:**

I don't have questions

**Limitations:**

The authors discuss limitations such as 2D video modalities and llm generated responses.

**Strengths And Weaknesses:**

The paper is technically sound. the framework is formally defined, the benchmark pipeline is principled, and ablations credibly isolate each component's contribution. The main concern is that QA pairs and reasoning trajectories are largely LLM-generated, risking annotation bias, and results are evaluated primarily on the authors' own benchmark. Presentation is clear and well-illustrated, though many implementation details are buried in the appendix and the related work could better distinguish MedScope from existing tool-augmented video QA systems. Long-form clinical video understanding is important and underserved, and the benchmark alone is a valuable community contribution. Originality is solid but incremental: the novelty is in applying tool-augmented reasoning to clinical video with physician-verifiable trajectories, rather than introducing a fundamentally new method.

---

> ### Author Rebuttal · Authors · 2026-03-31
>
> Thank you for your detailed review for highlighting so many strengths in our work for recognizing the **technical soundness of our framework, the credible ablations, and the value of our benchmark**.
>
> >**W1: QA pairs and reasoning trajectories are largely LLM-generated, risking annotation bias.**
>
> We thank the reviewer for this important question. We clarify that ClinVideoSuite is not built from raw LLM generations but through a multi-stage pipeline including text-only consensus filtering, shortcut filtering, local-evidence dependence filtering, and a final multimodal confirmation. The confirmation directly locates specific clip segments and verifies QA pairs against visual evidence at a fine-grained level, retaining only samples with strong agreement.
>
> To reduce bias, we employ two distinct model families: Qwen for captioning and filtering, Gemini for segmentation and multimodal confirmation. Cross-checking corrects shared blind spots. **As noted in Lines 1253-1254 of our Appendix, our evaluation suite includes physician verification for evidence traceability. We additionally invited three board-certified physicians to conduct comprehensive sampling validation on ClinVideo-QA-254K, with each physician independently reviewing 500 randomly sampled cases from one source dataset.** Each physician watched the clip and judged whether the answer is factually correct and visually grounded. Detailed results are available at https://anonymous.4open.science/r/MedScope-70CF/README.md.
>
> >**W2: Results are evaluated primarily on the authors' own benchmark.**
>
> We thank the reviewer for this question. **Our results are not limited to our own benchmark.** We also evaluate on SVU-31K, an independently developed benchmark covering Multi-grained Video Description and Fine-grained Visual Reasoning. **However，existing benchmarks like Cholec80 and EndoVis contain clips under one minute, and no benchmark evaluates reasoning over videos longer than ten minutes.** This motivated ClinVideo-Eval, verified by physicians as in W1. We additionally report on two established short-clip benchmarks:
>
> Cholec80-VQA:
> |Model|Acc|
> |---|---|
> |**MedScope**|**94.6**|
> |SurgicalGPT|94.0|
> |Surgical-LLaVA|92.2|
> |Surgical-VQA|89.8|
> |VisualBERT|89.7|
>
> EndoVis18-VQA:
> |Model|Recall|
> |---|---|
> |**MedScope**|**71.8**|
> |Surgical-LLaVA|68.7|
> |SurgicalGPT|66.0|
> |Surgical-VQA|63.2|
> |VisualBERT|61.4|
>
> >**W3: Many implementation details are buried in the appendix.**
>
> We agree. Due to space constraints, details were placed in the appendix. We will add clearer forward references in the main text to improve readability and reproducibility.
>
> >**W4: The related work could better distinguish MedScope from existing tool-augmented video QA systems.**
>
> We agree and will strengthen the positioning in the revision. The key distinction is that MedScope does not simply equip a general video model with tools. It targets clinical long videos where evidence traceability is essential, introduces coarse-to-fine retrieval and verification as a training objective rather than a prompting strategy, and uses GA-GRPO to optimize temporal grounding quality alongside answer correctness. Unlike existing methods, MedScope mirrors clinical workflow: skimming to hypothesize, then revisiting to verify, producing physician-verifiable trajectories.
>
> >**W5: Originality is solid but incremental: the novelty is in applying tool-augmented reasoning to clinical video with physician-verifiable trajectories, rather than introducing a fundamentally new method.**
>
> As the reviewer acknowledges, **"long-form clinical video understanding is important and underserved, and the benchmark alone is a valuable community contribution."** We build on this to clarify the depth of our contribution. To the best of our knowledge, MedScope is the first work to introduce an agent-based visual chain-of-thought paradigm for long-form medical video understanding, where the model actively seeks and verifies visual evidence through iterative tool calling rather than passively consuming pre-sampled frames. Our contributions span three levels: the skim-then-verify reasoning paradigm mirroring clinical practice; evidence-centric supervision via ClinVideoSuite creating a trainable closed loop among tool calls, evidence windows, and answers; and GA-GRPO incorporating temporal grounding quality into RL optimization. Together these form a systematic methodological contribution to evidence-grounded clinical video reasoning.
>
> >**W6: This paper needs ethical and legal review.**
>
> We appreciate this important concern. All source datasets, namely MedVideoCap, OphVL, and SurgVidLM, are publicly available datasets released for research purposes. Data were processed under institutional ethics review. MedScope is developed for research, not clinical deployment. We are happy to provide additional ethics documentation upon request.
>
> Thanks so much for your support! We are eager to address any remaining concerns you might have.

---

> > ### Author Rebuttal · Reviewer_CUTN · 2026-04-04
> >
> > The authors response for LLM generations part is not convincing. The authors have addressed my other concerns. I am keeping my original score.

---

> > > ### Author Response · Authors · 2026-04-04
> > >
> > > We thank the reviewer for the careful consideration of our rebuttal and for acknowledging that our responses addressed the other concerns. We appreciate the continued feedback regarding LLM generated data and the opportunity to further clarify this important point.
> > >
> > > > **On Data Quality and Automatic Annotations**
> > >
> > > We thank the reviewer for raising concerns about potential annotation bias from LLM generated data. ClinVideoSuite is not constructed from raw LLM outputs but through a multi stage evidence centric pipeline including text only consensus filtering, shortcut filtering, local evidence dependence filtering, and final multimodal confirmation against localized video clips. Only samples that remain consistent under progressively stricter evidence checks are retained, ensuring that supervision is grounded in visual evidence rather than free form generation.
> > >
> > > For reasoning trajectories, generation is further constrained through GT anchored distillation with structured prompts provided in the Appendix. Trajectories are synthesized as environment interactive thought action observation sequences conditioned on retrieved visual evidence instead of unconstrained chain of thought generation. This design enforces evidence grounded reasoning and reduces hallucinated intermediate steps.
> > >
> > > We further mitigate bias through cross model verification across different model families and perform physician led manual auditing on randomly sampled clips to verify factual correctness and visual grounding.
> > >
> > > Finally, we note that LLM generated data with filtering and distillation has become widely adopted in recent literature. Prior work such as Self Instruct demonstrates that models can generate instruction data followed by filtering to improve supervision quality [1]. Subsequent works such as Alpaca further extend this paradigm by distilling high quality instruction responses from stronger teacher models [2]. Orca further shows that reasoning traces distilled from stronger models can provide reliable supervision for complex reasoning tasks [3]. More recent work such as CoT Self Instruct demonstrates that structured reasoning trajectories generated with guided prompts can improve training reliability [4]. In addition, recent surveys and domain specific studies highlight the effectiveness of synthetic supervision pipelines with filtering and distillation, including general synthetic data surveys and clinical synthetic data generation studies [5,6]. These studies collectively support the reliability of synthetic supervision when combined with filtering and distillation.
> > >
> > > Together, multi stage filtering, GT anchored trajectory distillation, cross model verification, and human validation substantially mitigate annotation bias and ensure the reliability of ClinVideoSuite.
> > >
> > >
> > > References
> > >
> > > [1] Wang, Yizhong, et al. "Self-instruct: Aligning language models with self-generated instructions." Proceedings of the 61st annual meeting of the association for computational linguistics (volume 1: long papers). 2023.
> > >
> > > [2] Taori, Rohan, et al. "Stanford alpaca: An instruction-following llama model." 10 Mar. 2023,
> > >
> > > [3] Mukherjee, Subhabrata, et al. "Orca: Progressive learning from complex explanation traces of gpt-4." arXiv preprint arXiv:2306.02707 (2023).
> > >
> > > [4] Yu, Ping, et al. "Cot-self-instruct: Building high-quality synthetic prompts for reasoning and non-reasoning tasks." arXiv preprint arXiv:2507.23751 (2025).
> > >
> > > [5] Li, Zhuoyan, et al. "Synthetic data generation with large language models for text classification: Potential and limitations." Proceedings of the 2023 conference on empirical methods in natural language processing. 2023.
> > >
> > > [6] Barr, Austin A., et al. "Large language models generating synthetic clinical datasets: a feasibility and comparative analysis with real-world perioperative data." Frontiers in artificial intelligence 8 (2025): 1533508.

---

### Review · Ethics_Reviewer_8Rgr · 2026-04-01

**Recommendation:** Remediation action needed

**Ethics Issue:**

This is a strong paper, but the ethics discussion is still somewhat too thin given that the application domain is medicine and therefore high-risk. The manuscript does contain a basic responsible-research statement: it says the clinical data were obtained under institutional ethical review and approval, processed under applicable regulations and data-governance requirements, and subject to privacy protection and de-identification where required; it also states that the system is not intended for standalone clinical use and should be used with human oversight.

The paper however currently says little about legal and governance compliance in a narrower sense. In particular, it does not clearly address GDPR-specific obligations, copyright or licensing constraints, terms of use of the underlying datasets, or how these considerations relate to the planned release of “code, models, and data.” Given that this work is built on clinical video data aggregated from several sources, that omission is significant. The authors should clarify what exactly will be released, under what access conditions, and how release is reconciled with privacy, institutional approval, and source-dataset restrictions. I think this should be a straightforward fix for the authors.

A second concern is over-trust. The paper rightly emphasizes evidence-grounded reasoning and localized visual support, but this can create an impression of reliability that exceeds what benchmark performance alone establishes. The authors should state more plainly that evidence localization does not by itself guarantee correctness, robustness, or clinical safety, especially under distribution shift or tool failure. This point is partly acknowledged in the impact statement, but it would be better stated in the main paper as well. This could be as short as a sentence but it would be useful.

Overall, I do not see a major ethical flaw that would block the work, but I do think the paper needs a bit more concrete discussion of responsible research practice and legal compliance. The strongest revision would be to expand the current ethics statement beyond general IRB/privacy language and address data release, licensing, terms of use, and the limits of clinical readiness more directly.

**Remediation Action:**

See above in the description of issues what should be changed.

---

### Review · Ethics_Reviewer_GrsW · 2026-04-02

**Recommendation:** Remediation action needed

**Ethics Issue:**

Thank you for your submission. Currently, the contribution is very interesting, but more discussion around dataset ethics is needed.

1) The dataset relies on real surgical recordings from real procedures performed by real surgeons. While the impact statement mentions IRB approval for the dataset, construction, there is no discussion of whether the institutions, patients, or surgeons whose work underlies the dataset were asked for consent, or have any rights, attribution, or benefit-sharing arrangements. Similarly, more discussion around copyright and potential ethical implications stemming from the original data is needed.

2) Concerning the data governance aspects of the dataset, more discussion is needed on how the dataset complies with the various privacy regulations. The authors' claim that data was "obtained under institutional ethical review" reveals little about the adequacy of de-identification, particularly given that the paper's own figures show on-screen patient clinical histories and imaging findings. Further concern arises from the data synthesis pipeline, in which patient video clips are passed to proprietary third-party models for captioning and filtering.

---

### Decision · Program_Chairs · 2026-04-30

**Decision:**

Accept (regular)

**Comment:**

This paper addresses an important problem in long-form clinical video reasoning and makes a technically solid contribution through a tool augmented evidence seeking framework and a new benchmark. Reviewers were generally positive about the motivation, empirical results, and overall contribution.

The main concerns were about the reliance on automatically generated annotations, benchmark bias, and the need for stronger discussion of data governance, release conditions, and limits of clinical readiness. After considering the reviews, rebuttal, and ethics feedback, I find the paper strong enough for accept. The final version should explicitly address the ethics remediation points and clarify the limitations of the benchmark and the claims around clinical reliability.